# Forward-KL Convergence of Time-Inhomogeneous Langevin Diffusions

**Andreas Habring** [1]  **Martin Zach** [2]

## Abstract

Many practical samplers rely on time-dependent drifts—often induced by annealing or tempering schedules—to improve exploration and stability. This motivates a unified non-asymptotic analysis of the corresponding Langevin diffusions and their discretizations. We provide a convergence analysis that includes non-asymptotic bounds for the continuous-time diffusion and its Euler–Maruyama discretization in the forward-Kullback–Leibler divergence under a single set of abstract conditions on the time-dependent drift. The results apply to many practically-relevant annealing schemes, including geometric tempering and annealed Langevin sampling. In addition, we provide numerical experiments comparing the annealing schemes covered by our theory in low- and high-dimensional settings.

## 1. Introduction

We consider the problem of sampling from a Gibbs probability distribution $\pi$ on $\mathbb{R}^d$ which admits a density (with respect to the Lebesgue measure $\lambda$) of the form

$$p(x) \coloneqq \frac{\mathrm{d}\pi}{\mathrm{d}\lambda}(x) = \frac{1}{Z} \exp(-U(x)) \tag{1}$$

for some *potential* $U : \mathbb{R}^d \to \mathbb{R}$, where $Z = \int \exp(-U(x))\mathrm{d}x$ is the *partition function*. Sampling from such distributions is an omnipresent task within modern machine learning (LeCun et al., 2007; Welling & Teh, 2011), Bayesian inverse problems (Stuart, 2010; Dashti & Stuart, 2017), generative modeling (Song & Ermon, 2019; Nijkamp et al., 2019), and related fields.

---

[1]Institute of Visual Computing, Graz University of Technology, Graz, Austria [2]Biomedical Imaging Group, École Polytechnique Fédérale de Lausanne, Lausanne, Switzerland and CIBM Center for Biomedical Imaging, École Polytechnique Fédérale de Lausanne, Lausanne, Switzerland. Correspondence to: Andreas Habring <andreas.habring@tugraz.at>.

*Proceedings of the 43rd International Conference on Machine Learning*, Seoul, South Korea. PMLR 306, 2026. Copyright 2026 by the author(s).

In most relevant settings in high dimensions, it is typically infeasible to generate independent exact samples from $\pi$ (*e.g.*, via inversion or rejection sampling), and instead methods based on Markov chains (MCs) have become popular (Durmus & Moulines, 2017; Falk et al., 2025; Zach et al., 2023; Chehab et al., 2025; Habring et al., 2025b; Narnhofer et al., 2024; Vempala & Wibisono, 2019). Such approaches rely on an ergodic MC $(X_k)_k$ which admits $\pi$ (or, sometimes, an approximation thereof) as its stationary distribution so that sampling may be performed by simulating the chain for a sufficiently large number of iterations. A subclass of those MC-based methods is derived from discretizations of stochastic differential equations (SDEs). Among these SDEs, the *Langevin diffusion*

$$\mathrm{d}X_t = -\nabla U(X_t)\mathrm{d}t + \sqrt{2}\mathrm{d}W_t \tag{2}$$

for $t > 0$, where $(W_t)_t$ denotes Brownian motion, is particularly popular. Indeed, under appropriate assumptions on $U$ (*e.g.*, strong convexity and Lipschitz-continuity of the gradient), (2) admits a unique strong solution for all time whose distribution converges to $\pi$ as $t \to \infty$ (Dalalyan, 2017; Roberts & Tweedie, 1996; Durmus & Moulines, 2019). While more elaborate methods exist (Pereyra, 2016; Pereyra et al., 2020), (2) is typically discretized with an Euler–Maruyama scheme, which results in the popular *unadjusted Langevin algorithm (ULA)*

$$X_{k+1} = X_k - h\nabla U(X_k) + \sqrt{2h}Z_k \tag{3}$$

for $k = 0, 1, \ldots$, where $Z_k \sim \mathcal{N}(0, I_d)$. Though the MC $(X_k)_k$ is ergodic, its stationary distribution $\pi_h$ depends on the stepsize $h > 0$, is in general not identical to $\pi$, and only converges to $\pi$ as $h \to 0$ (Dalalyan, 2017; Durmus & Moulines, 2017).

In practice, ULA suffers from several drawbacks. In particular, distributions of interest in the context of modern machine-learning applications are typically high-dimensional, multimodal, and nonsmooth. In such cases, ULA exhibits poor convergence; samples frequently get trapped in a single mode and excessively many iterations are necessary to cover multiple modes of $\pi$. Approaches that build on the *underdamped* Langevin diffusion (Falk et al., 2025; Cheng et al., 2018) aim to accelerate the diffusion process and its discretization. In practice, however, methods based on *annealing* or *tempering*, such as annealed

Langevin sampling (Song & Ermon, 2019) or geometric tempering (Chehab et al., 2025), often perform better. Such approaches make use of a *path* of distributions $(\pi_\tau)_{\tau \geq 0}$ that is deliberately constructed such that $\pi_0$ coincides with $\pi$, the map $\tau \mapsto \pi_\tau$ is continuous (in some appropriate sense), and, ideally, it is easier to sample $\pi_\tau$ the larger $\tau$ is. We discuss the most popular ways to construct $(\pi_\tau)_{\tau \geq 0}$ and how they map into our framework in Section 4.3. Sampling using $(\pi_\tau)_\tau$ is performed by successively guiding the samples along the path $\pi_\tau$ as $\tau \to 0$. In order to do so, we consider the *time-inhomogeneous* Langevin diffusion

$$\mathrm{d}X_t = -\nabla U_{\tau(t)}(X_t)\mathrm{d}t + \sqrt{2}\mathrm{d}W_t, \qquad (4)$$

where $U_{\tau(t)} = -\log p_{\tau(t)} + c$ and $p_{\tau(t)} = \frac{\mathrm{d}\pi_{\tau(t)}}{\mathrm{d}\lambda}$.[1] In other words, we consider the Langevin diffusion for the *moving target* $\pi_{\tau(t)}$. The function $\tau : \mathbb{R}_{\geq 0} \to \mathbb{R}_{\geq 0}$ satisfies $\tau(t) \to 0$ as $t \to \infty$ and is an *annealing schedule* which allows us to control how quickly the target changes and, in particular, how quickly it moves from an easy-to-sample distribution to the actual target $\pi$. We denote the law of $X_t$ that satisfies (4) as $\mu_t$.

Despite their empirical success, the theoretical analysis of the time-inhomogeneous Langevin diffusion remains fragmented. Existing guarantees typically specialize to a particular path (*e.g.*, geometric tempering (Guo et al., 2025; Chehab et al., 2025) or convolutional paths (Cordero-Encinar et al., 2025)) and require smoothness assumptions that implicitly assume strong solutions of the Fokker–Planck equation. In addition, contrary to the time-homogeneous Langevin dynamics, most existing guarantees in the time-inhomogeneous setting control the backward-Kullback–Leibler divergence $\mathrm{KL}(\pi \,|\, \mu_t)$ instead of the forward-Kullback–Leibler divergence $\mathrm{KL}(\mu_t \,|\, \pi)$ (Guo et al., 2025; Cordero-Encinar et al., 2025; Cattiaux et al., 2025; Young et al., 2026). However, the latter is the natural notion of convergence for the Langevin diffusion since it is the Lyapunov functional of the Fokker–Planck equation and directly controls entropy dissipation. As a result, practitioners lack a single theorem that (a) certifies forward-KL convergence for broad families of annealing paths and (b) exposes how schedule choice trades off mixing and tracking error. We provide such a theorem for (4) and its Euler–Maruyama discretization in this article.

**Contributions**  We provide the following contributions:

1. We show that the time-inhomogeneous Langevin diffusion (4) and its Euler–Maruyama discretization converge to the target $\pi$ in the forward-Kullback–Leibler divergence under abstract assumptions on the path $(\pi_\tau)_\tau$ (Theorems 4.4 and 4.7). Contrary to existing

---

[1]The potential $U_{\tau(t)}$ needs to be specified only up to an additive constant as it only enters via its gradient.

analyses, we carefully address potential nondifferentiabilities (due to only weak solutions of the Fokker–Planck equation in general) and provide results about bias-free convergence of the schemes for vanishing step sizes (square-summable but not summable).

2. We show under which assumptions popular paths are covered by our analysis. In particular, we consider geometric tempering, dilation, paths based on the Moreau envelope of the potential, and convolutional paths used in diffusion models (Propositions 4.11 to 4.13 and 4.16). We also show that paths of the posteriors of Bayesian Gaussian-linear inverse problems inherit the guarantees from the paths of the prior (Proposition 4.17).

3. We derive practical guidelines for designing rapidly-converging paths $(p_\tau)_\tau$ from the convergence results (Remark 4.10).

4. We compare annealing schemes numerically on multimodal densities. The results confirm our theoretical analysis and provide insights into use cases and benefits of the considered time-inhomogeneous sampling schemes.

**Organization of the Article**  The remainder of the article is organized as follows. In Section 2, we discuss related work. In Section 3, we introduce some notation and relevant preliminary results and present the necessary assumptions on the path $(\pi_\tau)_\tau$. Section 4 is devoted to our main contributions: We prove convergence of the continuous-time diffusion (4) to the target $\pi$ in Section 4.1, prove convergence of the corresponding Euler–Maruyama discretization in Section 4.2, and cast various popular paths into our framework in Section 4.3. Finally, in Section 5, we provide numerical results that compare these different annealing families empirically and conclude the article with future research directions in Section 6.

## 2. Related Work

### 2.1. Time-Inhomogeneous Langevin Diffusion

The convergence of the time-homogeneous Langevin diffusion (2), its Euler–Maruyama discretization (3), and particular extensions have been studied extensively in recent years and convergence results under various assumptions on the potential can be found in (Roberts & Tweedie, 1996; Durmus & Moulines, 2017; Durmus et al., 2019; Habring et al., 2024; Dalalyan, 2017). The theoretical analysis of the time-inhomogeneous Langevin diffusion (4) is more recent and, so far, all works are specialized for particular paths. Cordero-Encinar et al. (2025) analyzed *convolutional paths* (defined in Section 4.3.4), which are closely related to diffusion models that were introduced by Song et al. (2021). They proved convergence of the corresponding sampling

scheme by bounding the backward-Kullback–Leibler divergence between target and sample distribution using the Wasserstein metric derivative along the path of distribution. Guo et al. (2025) and Chehab et al. (2025) considered the *geometric-tempering path* (defined in Section 4.3.1). Like the analysis by Cordero-Encinar et al. (2025), the analysis by Guo et al. (2025) is with respect to the backward-Kullback–Leibler divergence and builds on bounding the Wasserstein metric derivative. The analysis by Chehab et al. (2025) is based on the descent of the forward-Kullback–Leibler divergence along the Fokker–Planck equation and is closer to the one in this manuscript. However, the present manuscript provides several additional contributions as mentioned above, most notably the consideration of potential non-smoothness arising from the weak Fokker–Planck equation and the bias-free convergence for vanishing step sizes. Moreover, all of the works mentioned above focus on a specific annealing scheme, whereas we provide a unifying analysis for annealing schemes based on abstract assumptions on the path $(\pi_\tau)_\tau$.

## 2.2. Related Schemes

A (discrete-time) sequence $(\pi_k)_{k=0}^K$ of distributions with $\pi_0 = \pi$ is also the basis of sequential Monte-Carlo (SMC) sampling (Dai et al., 2022; Lee & Santana-Gijzen, 2024; Mathews & Schmidler, 2024). The iteration rule in SMC alternates between proposal steps of drawing from a transition kernel with importance sampling and re-sampling steps. SMC places no restrictions on the sequence $(\pi_k)_{k=0}^K$ and the geometric path, which is also frequently used in time-inhomogeneous Langevin sampling, is a popular choice (Dai et al., 2022). Convergence results under the assumption of a uniform bound on the ratios $\pi_k/\pi_{k-1}$ can be found in (Lee & Santana-Gijzen, 2024; Mathews & Schmidler, 2024).

The time-inhomogeneous Langevin diffusion is also closely related to diffusion models (Song et al., 2021). Consider the SDE

$$\mathrm{d}X_t = f(X_t, t)\mathrm{d}t + g(X_t, t)\mathrm{d}W_t, \tag{5}$$

where $X_0 \sim \pi$. Under suitable choices for $f$ and $g$, the forward process admits a limiting marginal $X_\infty$ as $t \to \infty$, and one could sample from $\pi_0$ by simulating the SDE (5) in reverse with initial condition $X_\infty$: By Anderson's theorem (Anderson, 1982), the reverse SDE that reproduces the forward marginals satisfies

$$\mathrm{d}X_t = \big(f(X_t, t) - g^2(t)\nabla \log p_t(X_t)\big)\mathrm{d}t + g(t)\mathrm{d}W_t, \tag{6}$$

where $p_t$ denotes the density of $X_t$ defined by the forward process, and $\mathrm{d}t$ is negative. Alternatively, one could sample from $\pi_0$ through the time-inhomogeneous Langevin diffusion (4), where the path traces out the density of $X_t$ defined by the forward process in reverse, and it is a natural question whether one should be preferred in certain situations.

Indeed, both variants have been considered: Methods that rely on discretizations of (6) are typically called "diffusion models", whereas the time-inhomogeneous Langevin diffusion on the forward-process densities is called "annealed Langevin sampling" (Song & Ermon, 2019). Hybrid variants have also been considered: The predictor–corrector algorithm due to Song et al. (2021, Algorithm 2) effectively alternates between reverse-SDE steps and Langevin sampling of a fixed density. In this context, our results are relevant because they endow annealed Langevin sampling and related schemes with convergence results.

## 3. Notation, Preliminaries, and Assumptions

For $x, y \in \mathbb{R}^d$, we denote the Euclidean scalar product as $\langle x, y \rangle$ and the Euclidean norm as $|x|$. We equip $\mathbb{R}^d$ with the standard Borel $\sigma$-algebra. We denote the Lebesgue measure on $\mathbb{R}^d$ as $\lambda$. Probability measures on $\mathbb{R}^d$ are denoted using greek letters, mainly $\mu, \nu$, and $\pi$, and probability density functions (with respect to the Lebesgue measure) using roman letters, mainly $p$ and $q$. The set of all probability measures on $\mathbb{R}^d$ is denoted as $\mathcal{P}(\mathbb{R}^d)$. For $\mu, \nu$ such that $\mu$ is absolutely continuous with respect to $\nu$ (denoted $\mu \ll \nu$), we denote the Radon–Nikodým derivative of $\mu$ with respect to $\nu$ as $\frac{\mathrm{d}\mu}{\mathrm{d}\nu}$. We define the Kullback–Leibler divergence as $\mathrm{KL} : \mathcal{P}(\mathbb{R}^d) \times \mathcal{P}(\mathbb{R}^d) \to [0, \infty]$,

$$\mathrm{KL}(\mu|\nu) = \begin{cases} \int \log\left(\frac{\mathrm{d}\mu}{\mathrm{d}\nu}\right) \mathrm{d}\mu, & \text{if } \mu \ll \nu, \\ \infty, & \text{otherwise.} \end{cases} \tag{7}$$

We denote with $C_c^\infty(\Omega)$ the set of (infinitely) smooth functions with compact support in $\Omega$, with $B_R(c) = \{x \in \mathbb{R}^d \mid |x - c| < R\}$ the open ball with radius $R > 0$ centered at $c \in \mathbb{R}^d$, and with $\mathbb{1}_A$ the characteristic function of the set $A$. We say that a function $f : \mathbb{R}^d \to \mathbb{R}^n$ is *Lipschitz-continuous* if there exists some $L > 0$ such that

$$|f(x) - f(y)| \leq L|x - y| \tag{8}$$

holds for all $x, y \in \mathbb{R}^d$. We say that a vector field $v : \mathbb{R}^d \to \mathbb{R}^d$ is *dissipative* if there exist $a, b, R > 0$ such that

$$\langle x, v(x) \rangle \geq a|x|^2 - b\mathbb{1}_{B_R(0)}(x) \tag{9}$$

holds for any $x \in \mathbb{R}^d$. We call $a, b, R > 0$ (in that order) the *dissipativity constants* of $v$. We say that a function $f : \mathbb{R}^d \to \mathbb{R}$ is *convex* if $f(\alpha x + (1 - \alpha)y) \leq \alpha f(x) + (1 - \alpha)f(y)$ for all $x, y \in \mathbb{R}^d$ and $\alpha \in [0, 1]$. A function $f$ is $\alpha$-*weakly convex* if $f + \frac{\alpha}{2}|\cdot|^2$ is convex. A function $f$ is *convex outside a ball* if there exists a compact set $K$ such that

$$f(\lambda x + (1 - \lambda)y) \leq \lambda f(x) + (1 - \lambda)f(y) \tag{10}$$

for all $\lambda \in (0, 1)$ and $x, y \in \mathbb{R}^d \setminus K$. For some $f : \mathbb{R}^d \to \mathbb{R}$, we define the *Moreau envelope with Moreau parameter* $m$,

denoted $M_f^m$, as

$$M_f^m(x) = \inf_{y \in \mathbb{R}^d} \left( f(y) + \frac{1}{2m}|x-y|^2 \right). \qquad (11)$$

Our theoretical analysis relies on the log-Sobolev inequality (LSI), which we now define.

**Definition 3.1** (Log-Sobolev inequality). We say that a probability measure $\nu$ satisfies the *log-Sobolev inequality* if there exists some $C > 0$ such that for every $f \in C_c^\infty(\mathbb{R}^d)$ it holds that

$$\int f^2 \log\left( \frac{f^2}{\overline{f^2}} \right) \mathrm{d}\nu \le C \int |\nabla f|^2 \mathrm{d}\nu, \qquad \text{(LSI)}$$

where $\overline{f^2} = \int f^2 \mathrm{d}\nu$. We call the smallest $C$ such that (LSI) holds the *log-Sobolev constant* of $\nu$, denoted $C_{\mathrm{LSI}}$.

We denote the path of *potentials* corresponding to the path of densities $(p_\tau)_\tau$ as $U_\tau := -\log p_\tau$.

Our theoretical analysis of (4) and its Euler–Maruyama discretization heavily relies on the following assumptions on $(p_\tau)_\tau$.

**Assumption 3.2.** The path $p_\tau$ satisfies the following:

- *At most quadratic growth of the time derivative.* There exists a constant $c > 0$ such that $|\partial_\tau U_\tau(x)| \le c(|x|^2 + 1)$ for any $x \in \mathbb{R}^d$ and $\tau \ge 0$.
- *Smoothness.* The potential $U_\tau$ is differentiable and its gradient $\nabla U_\tau$ is $L_\tau$-Lipschitz-continuous.
- *Dissipativity.* For any $\tau > 0$, $\nabla U_\tau$ is dissipative with dissipativity constants $a_\tau, b_\tau, R_\tau > 0$.
- Smoothness and dissipativity hold uniformly in $\tau$, that is,

$$L := \sup_{\tau > 0} L_\tau < \infty, \quad a := \inf_{\tau > 0} a_\tau > 0,$$
$$b := \sup_{\tau > 0} b_\tau < \infty, \quad R := \sup_{\tau > 0} R_\tau < \infty. \qquad (12)$$

- *Log-Sobolev inequality.* $\pi_\tau$ admits an LSI with constant $C_{\mathrm{LSI}}(\tau)$ and $\sup_{\tau \ge 0} C_{\mathrm{LSI}}(\tau) \le \overline{C_{\mathrm{LSI}}} < \infty$.

The LSI assumption is redundant if $U_\tau$ is *twice* continuously differentiable and satisfies the other assumptions, as we show in Lemma B.6. The addition of it does, however, allow us to extend our results to potentials that are only *once* continuously differentiable. In a slight abuse of notation we denote $C_{\mathrm{LSI}}(t) = C_{\mathrm{LSI}}(\tau(t))$.

*Remark* 3.3 (Nonsmooth potentials). Our results only apply to smooth potentials. One may, however, use our results to obtain convergence to appropriate smooth surrogates of any nonsmooth target potentials.

## 4. Theoretical Analysis

A prerequisite for the convergence is the existence of a unique (strong) solution of the Langevin diffusion (4), which

we now formally guarantee. This result is standard, but we include it here to emphasize that the associated densities only satisfy the associated Fokker–Planck equation in the weak sense in general.

**Theorem 4.1** (Existence of strong solutions of the time-inhomogeneous Langevin diffusion). *The Langevin diffusion* (4) *admits a unique strong solution* $(X_t)_t$ *for all time. More-over, the solution admits a density* $(q_t)_t$ *with respect to the Lebesgue measure which satisfies the Fokker–Planck equation*

$$\partial_t q_t = \mathrm{div}(q_t \nabla U_{\tau(t)}) + \Delta q_t \qquad (13)$$

*in the weak sense, that is, for all* $\varphi \in C_c^\infty(\mathbb{R}^d \times (0, \infty))$ *it holds*

$$0 = \int_{(0,\infty)} \int_{\mathbb{R}^d} \left( \partial_t \varphi(x,t) - \left\langle \nabla\varphi(x,t), \nabla U_{\tau(t)}(x) \right\rangle \right.$$
$$\left. + \Delta\varphi(x,t) \right) q_t(x) \mathrm{d}x \mathrm{d}t \qquad (14)$$

*Proof.* See Appendix B.1.1. □

### 4.1. Continuous-Time Diffusion

We will now state our results regarding the continuous-time Langevin diffusion. Our main result, the convergence in forward-Kullback–Leibler divergence in Theorem 4.4, will be based on a particular expression of the time-derivative of the forward-Kullback–Leibler divergence that we give in the following lemma.

**Lemma 4.2.** *Let* $\mu_t$ *be the distribution of* $X_t$ *satisfying* (4) *and* $q_t$ *its Lebesgue density. Then,*

$$\frac{\mathrm{d}}{\mathrm{d}t} \mathrm{KL}(\mu_t | \pi_{\tau(t)}) =$$
$$- \int_{\mathbb{R}^d} q_t \left| \nabla \log \frac{q_t}{p_{\tau(t)}} \right|^2 - q_t \partial_t \log p_{\tau(t)} \, \mathrm{d}x. \qquad (15)$$

*Proof.* See Appendix B.1.2. □

The expression of the time-derivative of the forward-Kullback–Leibler divergence in (15) can also be found in (Chehab et al., 2025; Vempala & Wibisono, 2019). To the best of our knowledge, however, the cited works implicitly assume sufficient regularity of the involved functions which are only known to be weak solutions of the Fokker–Planck equation in general. In contrast, our proof is based on careful mollification techniques and holds also if the density $q_t$ is only weakly differentiable and satisfies the Fokker–Planck equation in the weak sense.

**Lemma 4.3.** *There exists some $c > 0$ such that[2]*

$$\mathrm{KL}(\mu_t|\pi) \leq c\tau(t)$$

$$+ \exp\left(-\int_0^t \frac{4}{C_{\mathrm{LSI}}(s)}\mathrm{d}s\right)\mathrm{KL}(\mu_0|\pi_{\tau(0)})$$

$$+ c\int_0^t \exp\left(-\int_s^t \frac{4}{C_{\mathrm{LSI}}(\sigma)}\mathrm{d}\sigma\right)|\dot{\tau}(s)|\mathrm{d}s. \tag{16}$$

*Proof.* See Appendix B.1.3. □

Though it is possible to derive the appearing constants explicitly, we leave this for future work and only provide qualitative convergence results within this article.

We are now in a position to state our main result about the convergence of the time-inhomogeneous Langevin diffusion in forward-Kullback–Leibler divergence.

**Theorem 4.4** (Forward-KL convergence of the time-inhomogeneous Langevin diffusion). *Let $\tau : [0, \infty) \to [0, \infty)$ be such that $\tau(t) \to 0$ as $t \to \infty$ and $\dot{\tau} \leq 0$. Then,*

$$\lim_{t\to\infty} \mathrm{KL}(\mu_t|\pi) = 0. \tag{17}$$

*Moreover, as a worst case complexity bound in order to reach accuracy $\mathrm{KL}(\mu_t|\pi) \leq \varepsilon$, ignoring constants, we require that $t = \mathcal{O}(t_1 + \log(\varepsilon^{-1}))$ with $t_1 > 0$ such that $\tau(t_1) = \mathcal{O}(\varepsilon)$.*

*Proof.* See Appendix B.1.4. □

### 4.2. Discrete-Time Diffusion

We consider the Euler–Maruyama discretization of the time-inhomogeneous Langevin diffusion (4), which results in the update rule

$$X_{k+1} = X_k - h_k \nabla U_{\tau(t_k)}(X_k) + \sqrt{2h_k}Z_k \tag{18}$$

for $k = 0, 1, \ldots$, where $(Z_k)_k$ is a sequence of i.i.d. vectors $Z_k \sim \mathcal{N}(0, I_d)$, $(h_k)_k$ are the step sizes, and $t_k = \sum_{i=0}^{k-1} h_i$. We denote the law of $X_k$ that tracks (18) as $\hat{\mu}_k$. The structure of our results is analogous to the continuous case described in Section 4.1: We first relate the Kullback–Leibler divergences between the sample distributions of two consecutive iterates and the corresponding targets to each other (Lemma 4.5), then relate the Kullback–Leibler divergences between the sample distribution of an arbitrary iterate to the target (Lemma 4.6), and then show that this expression can be made arbitrarily small (Theorem 4.7). The following result is mostly taken from Lemma 3 in (Vempala & Wibisono, 2019). We provide a proof in the appendix

---

[2]For simplicity we use the single symbol $c$ for all appearing constants.

nonetheless for the sake of completeness and since the exposition in (Vempala & Wibisono, 2019) omits some details about exchanging integrals and derivatives.

**Lemma 4.5.** *Assume that the initial distribution admits a finite third moment, and let $h_k < \frac{a_{\tau(t_k)}}{L^2_{\tau(t_k)}}$. Then there exists some $c > 0$ such that for any $k \in \mathbb{N}$*

$$\mathrm{KL}(\hat{\mu}_{k+1}|\pi_{\tau(t_k)})$$

$$\leq \mathrm{KL}(\hat{\mu}_k|\pi_{\tau(t_k)})\exp\left(-\frac{2h_k}{C_{\mathrm{LSI}}(t_k)}\right) + ch_k^2. \tag{19}$$

*Proof.* See Appendix B.2.1 □

**Lemma 4.6.** *Under the assumptions of Lemma 4.5 the discrete scheme (18) satisfies for some $c > 0$*

$$\mathrm{KL}(\hat{\mu}_k|\pi) \leq \mathrm{KL}(\hat{\mu}_0|\pi_{\tau(t_0)})\exp\left(-\sum_{i=1}^k \frac{2h_i}{C_{\mathrm{LSI}}(t_i)}\right)$$

$$+ c\sum_{i=0}^{k-1}\exp\left(-\sum_{j=0}^i \frac{2h_{k-j}}{C_{\mathrm{LSI}}(t_{k-j})}\right)|\tau(t_{k-i}) - \tau(t_{k-1-i})|$$

$$+ c\sum_{i=0}^{k-1}h_{k-i}^2\exp\left(-\sum_{j=0}^{i-1} \frac{2h_{k-j}}{C_{\mathrm{LSI}}(t_{k-j})}\right) + c\tau(t_k) \tag{20}$$

*Proof.* See Appendix B.2.2. □

**Theorem 4.7** (Forward-KL convergence of the Euler—Maruyama discretization of the time-inhomogeneous Langevin diffusion). *Let $\tau : [0, \infty) \to [0, \infty)$ be such that $\tau(t) \to 0$ as $t \to \infty$ and $\dot{\tau} \leq 0$. Let the assumptions of Lemma 4.5 be satisfied and assume that $\sum_k h_k = \infty$ and $\sum_k h_k^2 < \infty$. Then*

$$\lim_{k\to\infty} \mathrm{KL}(\hat{\mu}_k|\pi) = 0. \tag{21}$$

*Moreover, choosing the step size constant, i.e., $h_k = h$ for all $k$ as a worst case complexity in order to obtain $\mathrm{KL}(\hat{\mu}_{k+1}|\pi)$ we require $h = \mathcal{O}(\varepsilon)$ and $k = \mathcal{O}(T_1\varepsilon^{-1} + \log(\varepsilon^{-1})h^{-1})$ with $T_1$ such that $\tau(T_1) = \mathcal{O}(\varepsilon)$.*

*Proof.* See Appendix B.2.3. □

*Remark* 4.8 (Feasibility of step sizes). The step-size conditions can be satisfied: Indeed, since $a_\tau$ and $L_\tau$ are uniformly bounded, the quantity $a_\tau/L^2_\tau$ is uniformly bounded away from zero. Therefore, it is possible to choose step sizes $(h_k)_k$ such that $\sum_k h_k = \infty$. The square-summability-condition is satisfied if the step sizes decay sufficiently fast.

*Remark* 4.9 (Constant step sizes). For the constant-step-size setting in Theorem 4.7, we obtain that $\lim_{h\to 0}\limsup_{k\to\infty}\mathrm{KL}(\hat{\mu}_k|\pi) = 0$. However, as expected, in general a nonzero bias remains for any fixed $h > 0$.

*Remark* 4.10 (Path-design guidelines). Our theoretical results imply the following practical consequences:

- [Lemma 4.6](#) provides design guidelines for the path $(p_\tau)_\tau$ and the annealing schedule $\tau$: The two main tools that lead to a small error term are small log-Sobolev constants and a fast-decreasing $\tau$. Of course, for $\tau = 0$, the log-Sobolev constant is pre-determined by the target $\pi$. Therefore, we find two contradicting incentives which need to be balanced. On the one hand, $\tau$ should remain *large* as long as possible to benefit from smaller log-Sobolev constants.[3] On the other hand, $\tau$ should converge to zero quickly to reach our actual target density. We will show in the numerical experiments that in well-designed annealing schemes (*e.g.*, convolutional paths and paths based on the Moreau envelope of the potential), the family $p_\tau$ is so well-conditioned initially that annealing leads to significant benefits even when $\tau$ decreases quickly.
- For the design of the path $(p_\tau)_\tau$, [Lemma B.6](#) implies that one should aim for paths that result in small values of $b_\tau, R_\tau$, and $L_\tau$, and large values of $a_\tau$. In addition to improved log-Sobolev constants and, thus, also fast convergence in the continuous-time setting, based on [Lemma 4.5](#) this also allows for large step sizes. The numerical experiments confirm that this is indeed a significant advantage of certain paths.

### 4.3. Paths Covered by our Framework

We now show how several popular annealing schemes are covered by the presented analysis.

#### 4.3.1. GEOMETRIC-TEMPERING PATH

The geometric-tempering path is defined for all $\tau \in [0, 1]$ as

$$p_\tau \propto p_1^\tau p_0^{1-\tau}, \qquad (22)$$

where $p_0$ is the target density and $p_1$ any (simple) initial density. It has the convenient property that the potential of $p_\tau$ is a linear combination of the potentials of $p_1$ and $p_0$, which makes it easy to implement. The convergence of annealing based on this path is covered in ([Chehab et al., 2025](#)) under the assumption that $U_0$ and $U_1$ are differentiable, and that $\nabla U_0$ and $\nabla U_1$ are Lipschitz-continuous and dissipative.

**Proposition 4.11.** *If $U_0$ and $U_1$ have Lipschitz-continuous and dissipative gradients, then the geometric path $(p_\tau)_{\tau \in [0,1]}$ satisfies [Assumption 3.2](#).*

*Proof.* Since $U_\tau = (1 - \tau)U_0 + \tau U_1 + c$, $U_\tau$ inherits Lipschitzness and dissipativity of its gradient uniformly in $\tau$. In

turn, Lipschitzness implies that $\partial_\tau U_\tau = U_1 - U_0$ satisfies the quadratic growth bound. $\square$

#### 4.3.2. DILATION

[Chehab & Korba](#) ([2024](#)) introduced the *dilation* path, which is defined for $\tau \in [0, 1)$ as

$$p_\tau = \frac{1}{(1-\tau)^{d/2}} p\left(\frac{\cdot}{\sqrt{1-\tau}}\right). \qquad (23)$$

Their work shows empirically that this path may be preferred in certain configurations, but they do not verify convergence of the scheme. In the following proposition, we verify that the dilation path conforms to our assumptions and, in turn, leads to a convergent scheme under the assumption that $U$ is differentiable, has a Lipschitz-continuous and dissipative gradient.

**Proposition 4.12.** *If $U$ has a Lipschitz-continuous and dissipative gradient, then the dilation path $(p_\tau)_{\tau \in [0,\overline{\tau}]}$ ([23](#)) satisfies [Assumption 3.2](#) for any $\overline{\tau} \in (0, 1)$.*

*Proof.* Since $U_\tau = U(\cdot / \sqrt{1-\tau}) + \frac{d}{2} \log(1-\tau) + c$,

$$\nabla U_\tau = \frac{1}{\sqrt{1-\tau}} \nabla U\left(\frac{\cdot}{\sqrt{1-\tau}}\right), \qquad (24)$$

and for $\tau \leq \overline{\tau}$ we obtain a uniform Lipschitz bound $|\nabla U_\tau(x) - \nabla U_\tau(y)| \leq \frac{L}{1-\overline{\tau}} |x - y|$. Dissipativity follows similarly as

$$\begin{aligned}
\langle x, \nabla U_\tau(x) \rangle &= \left\langle \frac{x}{\sqrt{1-\tau}}, \nabla U\left(\frac{x}{\sqrt{1-\tau}}\right) \right\rangle \\
&\geq \frac{a_\tau}{1-\tau} |x|^2 - b_\tau \mathbb{1}_{B_{R_\tau}(0)}\left(\frac{x}{\sqrt{1-\tau}}\right) \\
&\geq a_\tau |x|^2 - b_\tau \mathbb{1}_{B_{R_\tau}(0)}(x)
\end{aligned} \qquad (25)$$

Finally, since

$$\partial_\tau U_\tau(x) = \frac{1}{2(1-\tau)^{3/2}} \left\langle \nabla U\left(\frac{x}{\sqrt{1-\tau}}\right), x \right\rangle - \frac{d}{2(1-\tau)} \qquad (26)$$

the growth bound on $[0, \overline{\tau}]$ follows from Lipschitzness. $\square$

#### 4.3.3. DIFFUSION AT ABSOLUTE ZERO

[Habring et al.](#) ([2026](#)) consider a path of distributions that is defined via the Moreau envelope of the potential, with $\tau$ being the Moreau parameter:

$$p_\tau = \frac{\exp\left(-M_U^\tau(\cdot)\right)}{\int \exp\left(-M_U^\tau(y)\right) \mathrm{d}y}. \qquad (27)$$

They termed their method diffusion at absolute zero (DAZ) since this potential can be seen as the zero-temperature (in

---

[3] Consequently, annealing is useless whenever the family $p_\tau$ admits worse log-Sobolev constants for $\tau > 0$ than for $\tau = 0$, *cf.* ([Chehab et al., 2025](#)).

the Boltzmann sense) limit of the corresponding diffusion potentials (Habring et al., 2026, Section 4.3).

They analyze this path under the assumptions that the potential $U$ is differentiable, has a Lipschitz-continuous and dissipative gradient, and is $\alpha$-weakly convex and convex outside a ball. They show that, under these assumptions, for $\tau < \left(\frac{1}{\alpha} \wedge \frac{1}{L}\right)$, the path $p_\tau$ satisfies Assumption 3.2 so that the presented results hold.

**Proposition 4.13.** *If $U$ is $\alpha$-weakly convex and differentiable, $\nabla U$ is $L$-Lipschitz, dissipative, and $U$ is convex outside a ball, then the path $(p_\tau)_{\tau \in [0,\overline{\tau}]}$ (27) satisfies Assumption 3.2 on $[0,\overline{\tau}]$ for any $\overline{\tau} \in (0, 1/\alpha \wedge 1/L)$.*

*Proof.* For $\tau < (1/\alpha \wedge 1/L)$, the minimizing argument in (11) is unique, the Moreau envelope is differentiable and its gradient is globally Lipschitz with constant bounded by $L/(1 - \tau L)$ (Habring et al., 2026, Remark 4.17). Dissipativity is also maintained (Habring et al., 2026, Lemma 4.13). Since the Moreau envelope satisfies the Hamilton–Jacobi equation $\partial_\tau M_U^\tau = -\frac{1}{2}|\nabla M_U^\tau|^2$, the quadratic growth bound is implied by Lipschitzness. $\square$

This result significantly extends the analysis of Habring et al. (2025a): They only showed the convergence of the intermediate sampling distributions to the corresponding intermediate target, whereas this result shows that the annealing procedure *as a whole* converges to the ultimate target.

### 4.3.4. CONVOLUTIONAL PATH

Cordero-Encinar et al. (2025) consider the *variance preserving* convolutional path

$$p_\tau = \frac{p\left(\frac{\cdot}{\sqrt{1-\tau}}\right)}{(1-\tau)^{d/2}} * \frac{p_1\left(\frac{\cdot}{\sqrt{\tau}}\right)}{\tau^{d/2}}, \quad (28)$$

where $p_1 = \mathcal{N}(0, I_d)$ is a standard Gaussian and $p = p_0$ is the target. This path is used in annealed Langevin sampling (Song & Ermon, 2019) and the vast literature on diffusion models. With $X \sim p_0$ and $Z \sim p_1$ independent, its distribution corresponds to the law of the random variable

$$X_\tau = \sqrt{1-\tau} X + \sqrt{\tau} Z. \quad (29)$$

Cordero-Encinar et al. (2025) show (Lemma 3.2) that if $U$ is twice continuously differentiable, admits a Lipschitz-continuous gradient, and is strongly convex outside a ball, then $U_\tau$ admits a uniformly Lipschitz-continuous and dissipative gradient. In the following we will prove that the remaining assumptions of this paper are satisfied. We begin by deriving the SDE characterizing $(X_\tau)_\tau$.

**Lemma 4.14.** *The density $p_\tau$ from (28) is precisely the density of the solution of the SDE*

$$\mathrm{d}X_\tau = -\frac{1}{2(1-\tau)} X_\tau \mathrm{d}\tau + \frac{1}{\sqrt{1-\tau}} \mathrm{d}W_\tau, \quad (30)$$

*with initial condition $X_0 \sim \pi_0$.*

*Proof.* See Appendix B.3.1. $\square$

Based on this result, the following lemma formally shows that the quadratic growth bounds are satisfied.

**Lemma 4.15.** *There exists $c > 0$ such that the density (28) satisfies $|\partial_\tau U_\tau(x)| \le c(|x|^2 + 1)$ for $\tau \in [0, 1]$.*

*Proof.* See Appendix B.3.2. $\square$

This result, paired with the results by Cordero-Encinar et al. (2025), immediately implies that this path is covered by our analysis.

**Proposition 4.16.** *If $U$ is twice continuously differentiable, has a Lipschitz gradient and is strongly convex outside a ball, then the path $(p_\tau)_{\tau \in [0,1]}$ (28) satisfies Assumption 3.2.*

*Proof.* By (Cordero-Encinar et al., 2025, Lemma 3.2), all potentials in the path have uniformly Lipschitz-continuous and dissipative gradients. By Lemma 4.15, the growth bound on $\partial_\tau \log p_\tau$ required in Assumption 3.2 follows. $\square$

### 4.3.5. POSTERIORS IN BAYESIAN INVERSE PROBLEMS

As an important special case, we consider the posterior distributions that arise in the context of Bayesian inverse problems. The analysis in this section is different to that in Sections 4.3.1 to 4.3.4: Instead of showing that a particular path is valid, we show that such posteriors inherit the properties of a valid path if the prior is such a valid path. In this context annealing schemes have also been applied in the literature, *cf.* (Xun et al., 2025; Habring et al., 2026; Blumenthal et al., 2025).

In Bayesian inverse problems, one is interested in sampling from posterior distributions

$$p(x) := p_{X|Y}(x|y) \propto p_{Y|X}(y|x) p_X(x) \quad (31)$$

where $y$ is some known data. In many relevant settings, such as medical imaging, microscopy, and astronomy, the relationship between the data $y$ and the data-generating and unknown signal $x$ can be modeled as $y = Ax + \varepsilon$ where $\varepsilon \sim \mathcal{N}(0, \sigma^2)$ with $\sigma^2$ the variance of the additive Gaussian noise. In this case, the *likelihood* admits the density

$$p_{Y|X}(y|x) = \frac{1}{(2\pi\sigma^2)^{d/2}} \exp\left(-\frac{|Ax - y|^2}{2\sigma^2}\right) \quad (32)$$

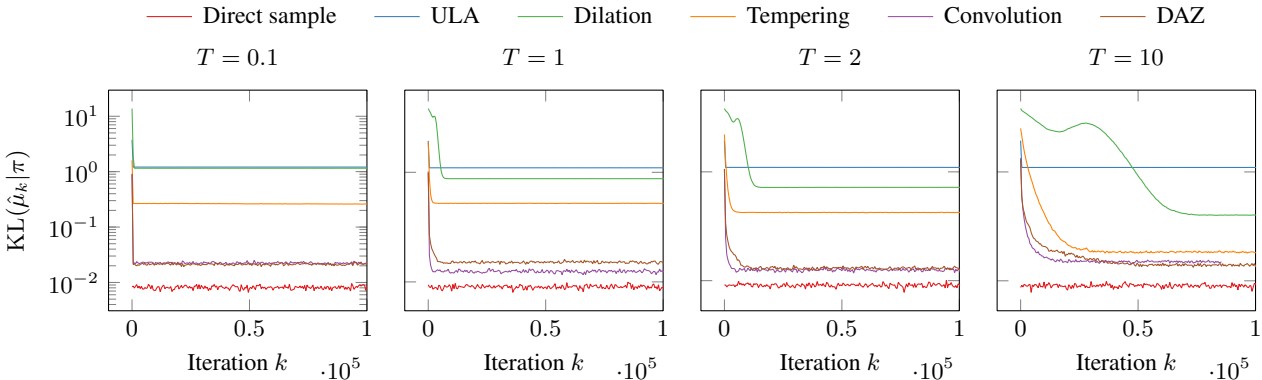

*Figure 1.* Forward-Kullback–Leiber divergence between the sample distribution and the one-dimensional Gaussian-mixture target in terms of the iteration for different paths and traversal speeds controlled by the parameter $T$. *Direct sample*: A sample of the same size as used in the other schemes, directly drawn from the Gaussian mixture.

and we will show in the following proposition that it is possible to prescribe a path $(p_X^\tau)_\tau$ for the *prior* and that

$$p_\tau(x) := \frac{p_{Y|X}(y|x)p_X^\tau(x)}{p_Y(y)} \quad (33)$$

satisfies Assumption 3.2 as long as $p_X^\tau$ satisfies it.

**Proposition 4.17.** *Assume the prior path $(p_X^\tau)_{\tau\in[0,1]}$ satisfies Assumption 3.2 with potentials $U_X^\tau = -\log p_X^\tau$. Define for fixed $y$ the posterior path $(p_\tau)_\tau$ as $p_\tau(x) \propto p_{Y|X}(y|x)p_X^\tau(x)$ and let $U_\tau = -\log p_\tau$ be its potential. Then $(p_\tau)_{\tau\in[0,1]}$ satisfies Assumption 3.2. Moreover, the Lipschitz constant of $\nabla U_\tau$ increases by at most $\sigma^{-2}\|A^\top A\|$.*

*Proof.* We have $U_\tau(x) = \frac{1}{2\sigma^2}|Ax - y|^2 + U_X^\tau(x) + \text{const}$, hence

$$\nabla U_\tau(x) = \frac{1}{\sigma^2}A^\top(Ax - y) + \nabla U_X^\tau(x). \quad (34)$$

The first term is globally Lipschitz with constant $\sigma^{-2}\|A^\top A\|$, so $\nabla U_\tau$ is uniformly Lipschitz if $\nabla U_X^\tau$ is. Dissipativity is preserved (and typically strengthened) since $\langle x, A^\top Ax \rangle \geq 0$ and cross terms are at most linear in $|x|$. Finally, $\partial_\tau U_\tau = \partial_\tau U_X^\tau$ because the likelihood term does not depend on $\tau$, so the growth condition transfers directly. $\square$

## 5. Numerical Experiments

We now confirm the presented convergence results and compare the efficacy of the different paths discussed in Section 4.3 numerically. Across all experiments we consider the problem of sampling from the Gaussian mixture

$$\pi = \sum_{i=1}^{n_c} \alpha_i \mathcal{N}(m_i, \Sigma_i) \quad (35)$$

with $n_c$ components that have the means $(m_i)_{i=1}^{n_c} \subset \mathbb{R}^d$, covariances $(\Sigma_i)_{i=1}^{n_c} \subset \mathbb{R}^{d\times d}$, and relative weights $(\alpha_i)_{i=1}^{n_c} \subset$

$\mathbb{R}$. The choice of the Gaussian mixture is primarily motivated by the fact that it admits analytical expressions for all intermediate distributions (and, in particular, the gradient of the potential of their densities) in all paths: The distributions of the convolutional path from Section 4.3.4 can be computed explicitly for all $\tau \in [0, 1]$ as

$$\pi_\tau = \sum_{i=1}^{n_c} \alpha_i \mathcal{N}(m_i(\tau), \Sigma_i(\tau)) \quad (36)$$

with means $m_i(\tau) = m_i\sqrt{1 - \tau}$ and covariance $\Sigma_i(\tau) = (1 - \tau)\Sigma_i + \tau I_d$. Therefore, $\nabla U_\tau$ of the convolutional path remains easy to compute without prior training. This allows us to directly compare the different annealing schemes without interference of potential training inaccuracies.

For all experiments, we choose the annealing schedule[4] $\tau(t) = \exp\left(-\frac{t}{T}\right)$ and compare different values for $T$. We always pick the largest possible step size that the theoretical results allow for each method: $h_k = a_{\tau(t_k)}/L_{\tau(t_k)}^2$. We estimated these parameters for the target through the largest and smallest eigenvalue of any of the covariance matrices. For geometric tempering and the dilation path, this suffices to directly compute these parameters for all $k$. For the convolutional path, we estimated them at each iteration after an appropriate addition of a multiple of the identity matrix. For DAZ, we use the estimates by Habring et al. (2026). In all experiments, we simulate $N = 5000$ independent chains to estimate the KL divergence.

### 5.1. Results

Here, we only present the results for $d = 1$. Additional results for $d \in \{2, 10\}$ as well as for an imaging experiment can be found in Appendix A. The parameters of the target

---

[4]The exponential schedule was not chosen for a particular reason other than that it constitutes a very natural way to model decay and its velocity via the half-life period $T$.

distribution (35) are set to $n_c = 3$, $(\alpha_i)_i = (0.3, 0.4, 0.3)$, $(m_i)_i = (-2, 0, 2)$, $(\sigma_i)_i = (0.2, 0.1, 0.3)$ where in this case $\sigma_i$ is the standard deviation of the $i$-th mixture component.

We show the KL divergence for the different annealing schemes over the iterations in Figure 1. Some corresponding representative empirical histograms (after 1000, 5000, and 40 000 iterations) are shown in Figure 2 in Appendix A. Overall, the convolutional path and DAZ perform best. For those paths, the annealing provides significant benefits in terms of quantitative convergence as well as mode coverage. Tempering, dilation, and ULA trail in that order. This is in line with the theoretical results. In order to obtain small error terms in (16) and its discrete counterpart (20), two contradicting conditions are relevant, *cf.* Remark 4.10: (i) Small log-Sobolev constants $C_{\mathrm{LSI}}(\tau)$, which is only possible for $\tau \gg 0$, since for $\tau = 0$, the log-Sobolev constant is pre-determined by $\pi$ and (ii) fast convergence $\tau \to 0$. The convolutional path and DAZ exhibit better log-Sobolev constants for $\tau \gg 0$ as they yield better constants for $L, a, b, R$, *cf.* Lemma B.6. Moreover, these paths decrease $L_\tau$ leading to larger step sizes (Habring et al., 2026, Remark 4.17). In contrast, dilation and tempering do not provide a similar regularizing behavior. This, in particular, leads to these methods struggling to cover all modes of the target, which is evident in the empirical histograms that we show in Figure 2 in Appendix A. The convergence of these methods becomes worse as $T$ decreases which indicates that these methods require more time at larger $\tau$ to effectively benefit from the annealing paths. Dilation in fact increases the Lipschitz constant, leading to the smallest allowed step sizes which is in line with its performance being worst among the compared methods.

## 6. Conclusion

In this article, we proved the convergence of time-inhomogeneous Langevin diffusions and their discretizations in the forward-Kullback–Leibler divergence. The proofs hold under an abstract set of assumptions on the drift of the diffusion and cover many popular annealing schemes, such as geometric tempering and annealed Langevin sampling. Compared to the literature, the presented proofs rigorously take care of potential non-smoothness arising from the fact that solutions of the Fokker–Planck equation are generally guaranteed only in the weak sense.

The convergence results as well as the numerical experiments clearly suggest that annealing paths should be designed in a way that ensures that initially the target density is mostly convex and well-conditioned (*i.e.*, small log-Sobolev constant, small Lipschitz constant of the gradient of the potential). In particular, in such cases we find a clear benefit of the time-inhomogeneous diffusion over basic ULA in terms

of convergence speed and mode coverage.

**Limitations** A more concrete quantification and ideal choice of the schedule $\tau(t)$ with the current analysis would rely on estimates of the log-Sobolev constants which are in general hard to obtain.

**Future work** For future work, we plan to derive explicit bounds on the log-Sobolev constants of probability paths $(p_\tau)_\tau$ in order to estimate optimal annealing schemes $\tau$ as well as optimal convergence rates.

## Impact statement

This paper presents work whose goal is to advance the field of machine learning. There are many potential societal consequences of our work, none of which we feel must be specifically highlighted here.

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

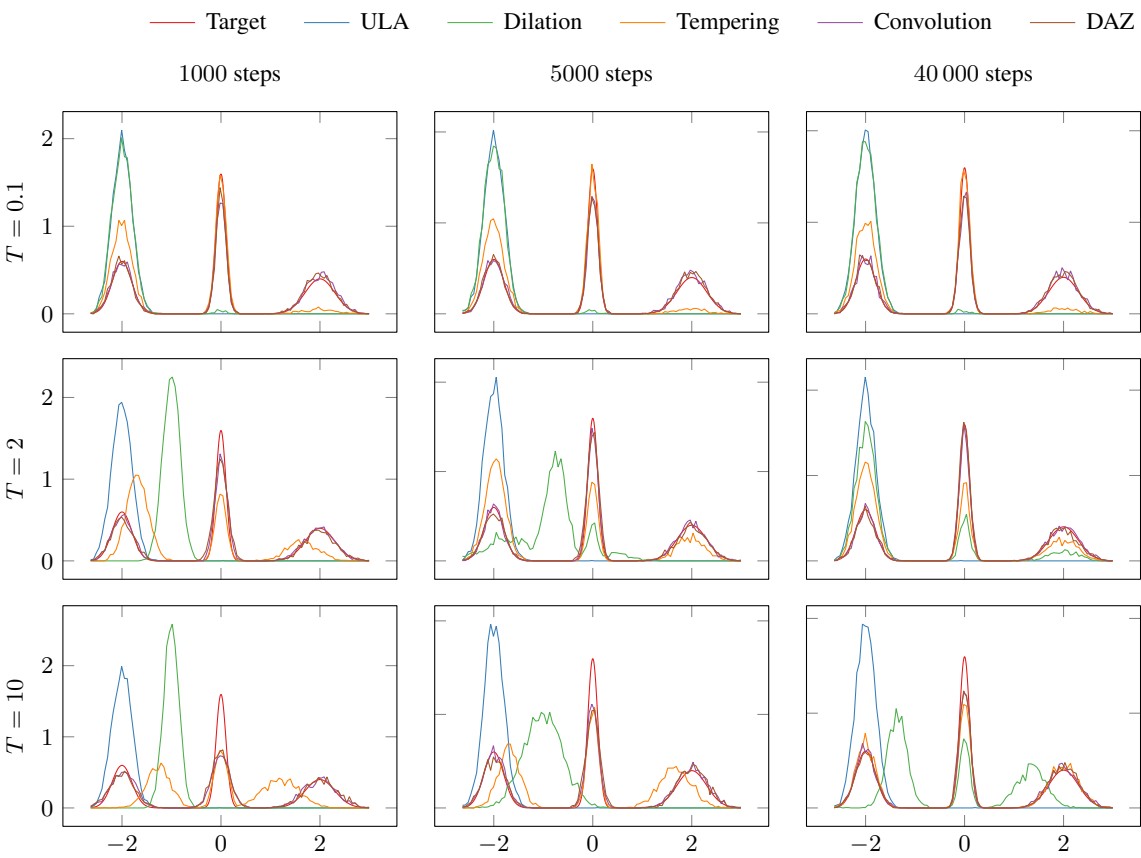

*Figure 2.* Histogram of the sample distributions compared to the one-dimensional Gaussian-mixture target. Rows show different values of the *annealing speed* controlled by $T$. Columns show the empirical histograms after a different number of steps of the discretization.

## A. Implementation Details Additional Numerical Results

### A.1. Implementation Details

As mentioned in the main part of the manuscript, we set the step sizes for all methods as $h_k = a_{\tau(t_k)}/L^2_{\tau(t_k)}$. In order to do so, we need to estimate $a_{\tau(t_k)}$ and $L^2_{\tau(t_k)}$, which we do as follows.

**ULA** For ULA, we have that $p_\tau = p_0 = \sum_{i=1}^{n_c} \alpha_i \mathcal{N}(m_i, \Sigma_i)$ for all $\tau$. We estimate $L = L_0$ and $a = a_0$ as

$$L_0 = \max_i \frac{1}{\lambda_{\min}(\Sigma_i)} \tag{37}$$

and

$$a_0 = \min_i \frac{1}{\lambda_{\max}(\Sigma_i)} \tag{38}$$

where $\lambda_{\min}$ and $\lambda_{\max}$ denote the smallest and largest eigenvalues, respectively.

**Dilation** We show in Section 4.3.2 that the time-dependent constants satisfy $a_\tau = \frac{a_0}{1-\tau}$ and $L_\tau = \frac{L_0}{1-\tau}$, so that the values for $\tau > 0$ can be estimated using $a_0$ and $L_0$ from ULA.

**Tempering** In the case of tempering where $U_\tau = (1-\tau)U_0 + \tau U_1 + c$ it follows that $L_\tau = (1-\tau)L_0 + \tau L_1$, $a_\tau = (1-\tau)a_0 + \tau a_1$. In our experiments we set $U_1(x) = |x|^2/2$ to be a standard Gaussian so that the parameters $a_1 = L_1 = 1$. The parameters for $\tau = 0$ are again estimated like for ULA.

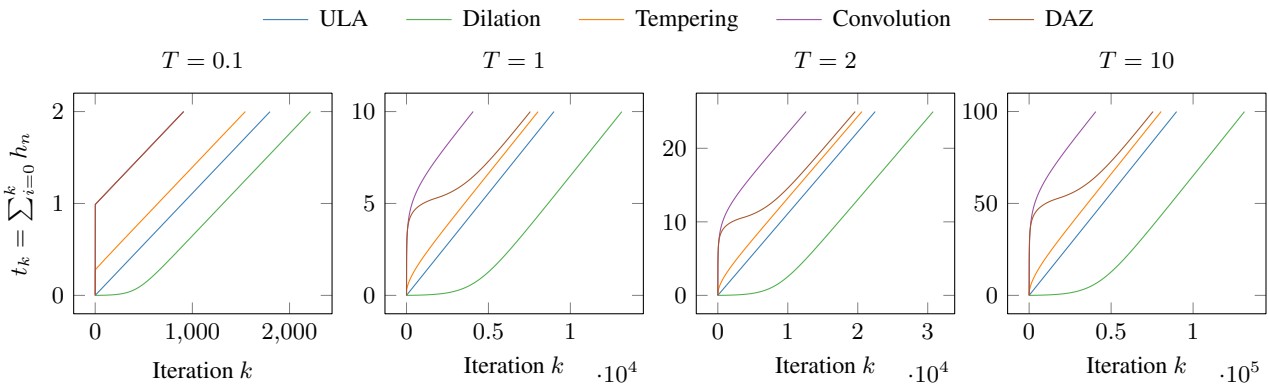

*Figure 3.* Comparison of step sizes for different annealing schemes. On the $x$ axis we plot the iteration and on the $y$-axis the physical time $t_k = \sum_{n=1}^{k} h_n$ achieved at iteration $k$ for the different annealing paths with the maximum allowed step size $h_n = a_{\tau(t_n)}/L^2_{\tau(t_n)}$. One can observe that especially the convolutional and the DAZ path admit larger step-sizes initially aiding the faster convergence.

**Convolution** In the convolutional path, $p_\tau$ stays a Gaussian mixture model (GMM) with component covariance matrices $\Sigma_i(\tau) = (1 - \tau)\Sigma_i + \tau I_d$. Thus, for each $\tau$ we may estimate the parameters as

$$L_\tau = \max_i \frac{1}{(1-\tau)\lambda_{\min}(\Sigma_i) + \tau} \tag{39}$$

and

$$a_\tau = \min_i \frac{1}{(1-\tau)\lambda_{\max}(\Sigma_i) + \tau}. \tag{40}$$

**DAZ** Based on (Habring et al., 2026, Remark 4.17), we set

$$L_\tau = \min\left\{\frac{1}{\tau}, \max\left\{0, \frac{L_0}{1 - L_0\tau}\right\}\right\} \tag{41}$$

and based on the definition of the Moreau envelope, we set $a_\tau = \min\left\{a_0, \frac{1}{\tau}\right\}$.

In Figure 3 we plot the physical time $t_k = \sum_{n=0}^{k} h_k$ for the different annealing paths. There one can observe that paths, which lead to a more well-posed distribution for $\tau > 0$ (especially DAZ and convolutional) naturally lead to larger step sizes meaning we can travel more physical time in the same number of iterations. An interesting aspect is that for DAZ after initially allowing for very large step sizes, there is a phase where step sizes are rather small which is related to the regime change in (39).

## A.2. Additional Numerical Results

We provide some additional experimental results in higher dimensional settings ($d = 2, 10$). In both cases, we again use as a target the GMM from (35).

## A.3. One-dimensional GMM

In Figure 4 we provide an additional plot for the 1D GMM experiment, where we use the exact same step sizes for all paths. The purpose of this experiment is to investigate whether the benefits of the annealing schemes are mostly caused due to larger possible step sizes or if, in fact, the well-posedness of $(p_\tau)_\tau$ plays a crucial role. In this experiment we use a fixed step size $h_k = h = a_0/L_0^2$ for all paths and all iterations where $a_0$ and $L_0$ are the parameters from Assumption 3.2 for $\tau = 0$. While the convergence is slower compared to the adaptive step size results in Figure 1, indeed, we find qualitatively a similar behavior, as in Figure 1. As a main difference we find that for very slow annealing schedules ($T = 10$) geometric tempering performs significantly better with smaller step sizes.

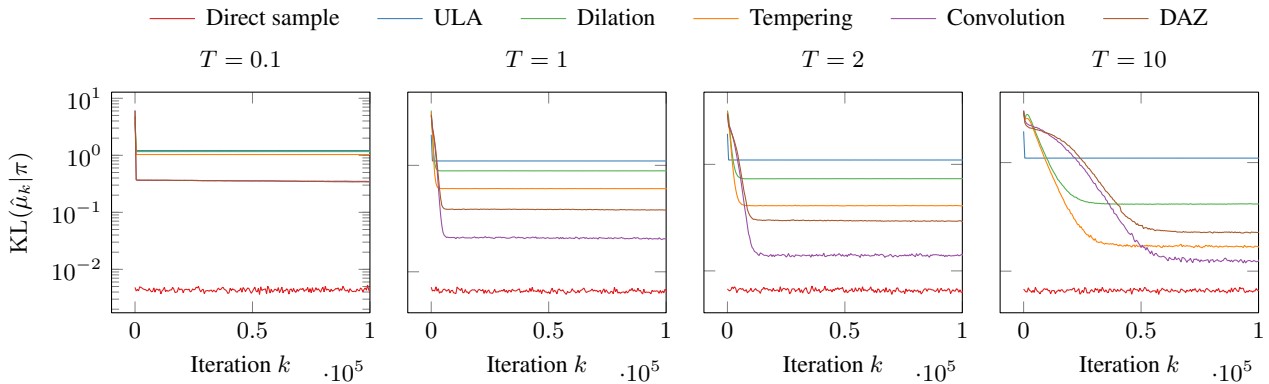

*Figure 4.* Forward-Kullback–Leiber divergence between the sample distribution and the one-dimensional Gaussian-mixture target in terms of the iteration for different paths and traversal speeds controlled by the parameter $T$. *Direct sample*: A sample of the same size as used in the other schemes, directly drawn from the Gaussian mixture.In this experiment all methods use the same, constant step size $h_k = h = a_0/L_0^2$ with $a_0, L_0$ the parameters according to Assumption 3.2 for $\tau = 0$.

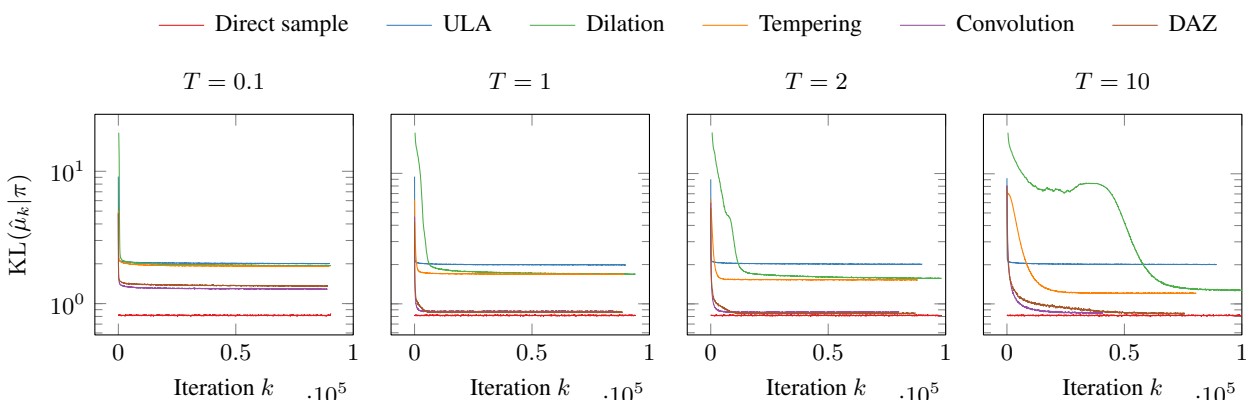

*Figure 5.* 2D Gaussian mixture. Convergence of different annealing schemes in KL. Different plots show results for different annealing speeds controlled by the parameter $T$. *Direct sample* refers to the KL divergence obtained for a sample directly drawn from the Gaussian-mixture model and of the same sample size $N$ as simulated using the annealing schemes.

### A.4. Two-dimensional GMM

In the 2D case we choose the parameters of the GMM as $n_c = 4$, $(x_i)_i = ([0., 0], [2, 0], [0, 2], [2, 2])$, and $(\alpha_i)_i = (0.2, 0.4, 0.2, .2)$. Regarding the covariance, we choose each $\Sigma_i$ as a diagonal matrix with entries $(\sigma_{i,j})_j$ which are chosen as $(\sigma_{1,j})_j = (0.2, 0.2)$, $(\sigma_{2,j})_j = (0.1, 0.2)$, $(\sigma_{3,j})_j = (0.3, 0.1)$, $(\sigma_{4,j})_j = (0.1, 0.1)$. We initialize the sampling at $x = (-1, -1)$. This way, it will be especially difficult for the methods to cover all modes, as the samplers may easily get trapped in the mode at $(0, 0)$.

In Figure 5 as in the 1D case we show convergence plots in KL distance and in Figure 6 empirical histograms after $1000$ and $40\,000$ iterations in comparison to the ground truth density. Overall, we can observe the same behavior as in the 1D case with best performance by DAZ and the convolutional path followed by tempering, dilation, and lastly conventional ULA. In Figure 5, we find the same behavior as in the 1D case. If the annealing *speed* is too high (*i.e.*, $T$ is small), the methods cannot benefit from favorable properties of $p_\tau$ for $\tau \gg 0$. When increasing $T$ on the other hand, the methods reach lower final errors. In this case, annealing helps covering all modes of the target. This is also confirmed by the empirical histograms in Figure 6. There, for $T = 0.1$ all methods except DAZ and the convolutional path fail to recover additional modes except for the one at $(0, 0)$. For $T = 2$ dilation and tempering cover three of the four modes, while still only DAZ and the convolutional path seem to reach all modes to an appropriate degree. The behavior of the dilation path for $1000$ iterations can be explained by the fact that for dilation $\nabla p_\tau(x) \approx 0$ for $x$ significantly far away from the origin so that the sampler cannot make meaningful progress until $\tau$ decreases sufficiently.

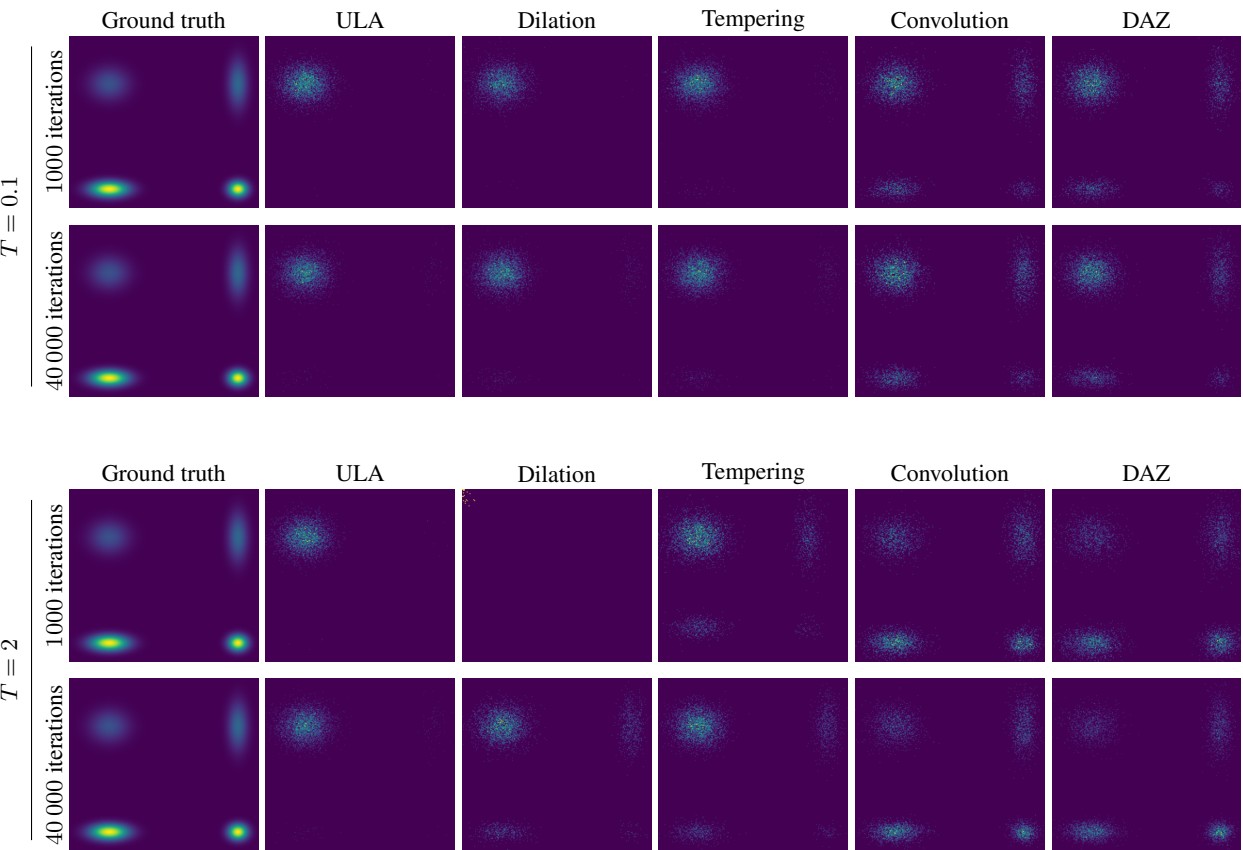

*Figure 6.* 2D Gaussian mixture. Empirical distributions compared to the true density. Rows show different values of the *annealing speed* controlled by $T$ and different numbers of steps of the discretization. Columns show different paths $(p_\tau)_\tau$.

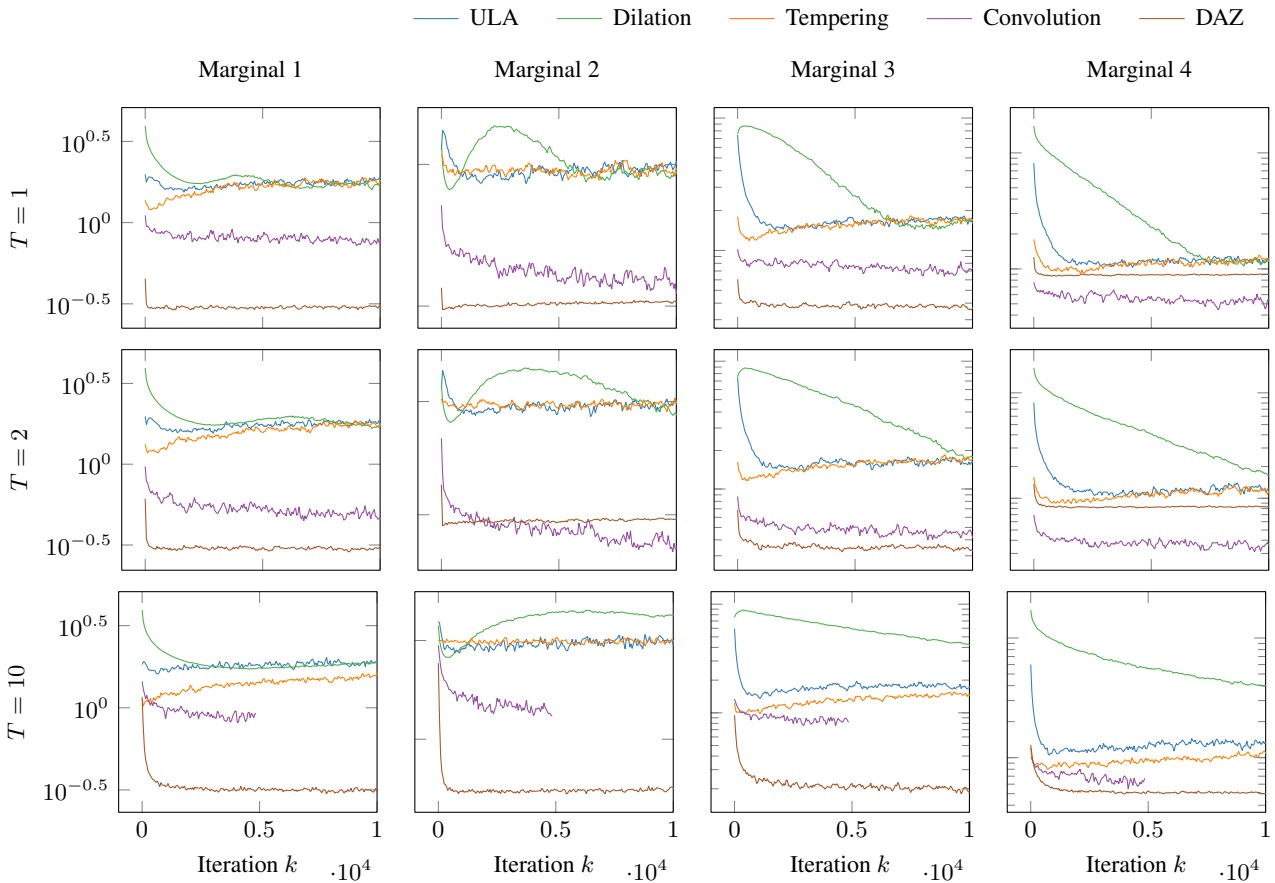

*Figure 7.* 10D Gaussian mixture. Convergence of different annealing schemes in KL. We plot the KL divergence $\mathrm{KL}(\Pi_{x_i}\hat\mu_k|\Pi_{x_i}\pi)$ for $i = 1, \ldots, 4$ where for $\nu \in \mathcal{P}(\mathbb{R}^d)$, $\Pi_{x_i}\nu$ denotes the $x_t$ marginal of $\nu$. Rows show results for different annealing speeds controlled by the parameter $T$. Columns show the KL divergence of the marginals $x_i$ for $i = 1, \ldots, 4$.

## A.5. 10-dimensional GMM

Lastly we consider the case $d = 10$. The GMM is chosen to have four modes again with weights $(\alpha_i)_i = (0.2, 0.4, 0.2, .2)$. The means as well as the variances of the components are initialized randomly with the means drawn from a standard normal distribution and the values $\sigma_{i,j}$ uniformly in the range $[0.1, 0.4]$. Since in high dimensions evaluating the full KL divergence is increasingly difficult, as a proxy we compute the KL divergence of the individual marginals. More precisely, let $\Pi_i : \mathcal{P}(\mathbb{R}^d) \to \mathcal{P}(\mathbb{R})$ be defined for any $\nu \in \mathcal{P}(\mathbb{R}^d)$ via

$$\Pi_i\nu(A) = \nu(\mathbb{R}^{i-1} \times A \times \mathbb{R}^{d-i}), \quad A \subset \mathbb{R}, \text{ measurable.} \tag{42}$$

Then as a measure of convergence we evaluate $\mathrm{KL}(\Pi_i\hat\mu_k|\Pi_i\pi)$. The results for $i = 1, \ldots, 4$ are shown in Figure 7. Again we find best convergence for the convolutional path and DAZ. In particular, DAZ performs best across all settings. In this experiment the difference to the other methods is even more significant. The reason for the graph showing the convergence of the convolutional path ending sooner at $T = 10$ is that we simulate all sampling algorithms for a fixed maximum time $t$. Since for each method the step sizes $h_k$ are chosen as the largest allowed step sizes and, thus, $h_k$ differs from method to method, it may happen that some methods reach the maximum time with fewer iterations then others.

## A.6. Image Denoising

In Figures 8 and 9 we show results for annealing used for image denoising. More specifically, in this example the target density is defined as in Section 4.3.5 as

$$p(x) = p_{X|Y} \propto p_{Y|X}(y|x)p_X(x) \tag{43}$$

where

$$p_{Y|X}(y|x) = \frac{1}{(2\pi\sigma^2)^{d/2}} \exp\left(-\frac{|x-y|^2}{2\sigma^2}\right) \tag{44}$$

and the annealing path is applied only to the prior, that is,

$$p_\tau(x) := \frac{p_{Y|X}(y|x)p_X^\tau(x)}{p_Y(y)} \tag{45}$$

with $p_X^\tau(x)$ following the usual annealing schemes from Section 4.3. As the data $y$ we use a clean image $x^\dagger$ from the CelebA dataset (Liu et al., 2015) with size $d = 256 \times 256$ which is corrupted by additive Gaussian noise, *i.e.*, $y = x^\dagger + \sigma z$ with $\sigma = 0.5$ and $z \sim \mathcal{N}(0, I_d)$. As a prior $p_X$ we use the DiffUNet model provided by the DeepInverse library (Tachella et al., 2025). We compare all methods except for DAZ in this experiment as the estimation of the proximal operator for a diffusion model[5] is highly unlikely to be accurate. For all methods we fix a uniform step size of $h = h_k = 10^{-3}$ and initialize the chains at the noisy image $y$.

In Figure 8 we see that for ULA the estimated posterior mean contains visible artifacts from the noisy initialization in the depicted face whereas the annealing schemes provide a clean denoised image indicating that those methods depend less strongly on the initialization. Naturally, if $T$ is too small, these effects are diminished (*cf.* the convolutional path for $T = 0.01$). In Figure 9 we provide quantitative results by plotting the mean squared error of the posterior mean to a reference estimate. As a reference we use the posterior mean obtained with ULA after $30\,000$ iterations. Again, we find that annealing can provide accelerated convergence. If $T$ is chosen too small, the benefits of annealing diminish, however, the convergence speed is still at least as fast as with plain ULA. On the other hand, if $T$ is chosen too large, naturally, convergence becomes slower.

## B. Postponed Proofs

Especially in proofs heavily relying on integration by parts, will sometimes use Einstein's sum convention, that is, whenever any index appears twice within a single term, we sum over said index, *e.g.*, $\partial_{x_i}\partial_{x_i}f(x) = \Delta f(x)$ and $\partial_{x_i}f(x)\partial_{x_i}g(x) = \langle\nabla f(x), \nabla g(x)\rangle$. We denote the upper bound of the log-Sobolev constants as

$$\overline{C_{\mathrm{LSI}}} = \sup_{\tau \geq 0} C_{\mathrm{LSI}}(\tau). \tag{46}$$

The $p$-th moment of a probability distribution $\mu \in \mathcal{P}(\mathbb{R}^d)$ is defined as the quantity $\int |x|^p \mathrm{d}\mu(x) < \infty$.

### B.1. Continuous-time Diffusion

#### B.1.1. Proof of Theorem 4.1

While the proof relies mostly on standard results we nonetheless provide the necessary references and the rigorous arguments.

*Proof.* The existence of a unique strong solution for all time is implied by the Lipschitz continuity of the drift (Karatzas & Shreve, 1998, Section 5.2, Theorem 2.9). Then, Itô's lemma implies that

$$\mathrm{d}\varphi(X_t, t) = \left[\partial_t\varphi(X_t, t) - \langle\nabla\varphi(X_t, t), \nabla U_{\tau(t)}(X_t)\rangle + \Delta\varphi(X_t, t)\right]\mathrm{d}t + \sqrt{2}\nabla\varphi(X_t, t)^\top \mathrm{d}W_t \tag{47}$$

for any $\varphi \in C_c^\infty(\mathbb{R}^d \times (0, \infty))$. Consequently, $\mu_t$ satisfies

$$\mathbb{E}[\varphi(X_T, T) - \varphi(X_0, 0)] = \mathbb{E}[\varphi(X_T, T)] = \int_0^T \int \left(\partial_t\varphi(x, t) - \langle\nabla\varphi(x, t), \nabla U_{\tau(t)}(x)\rangle + \Delta\varphi(x, t)\right) \mathrm{d}\mu_t(x)\mathrm{d}t. \tag{48}$$

The compact support of $\varphi$ then implies that

$$\lim_{T\to\infty} \mathbb{E}[\varphi(X_T, T)] = 0 = \int_0^\infty \int \left(\partial_t\varphi(x, t) - \langle\nabla\varphi(x, t), \nabla U_{\tau(t)}(x)\rangle + \Delta\varphi(x, t)\right) \mathrm{d}\mu_t(x)\mathrm{d}t \tag{49}$$

---

[5]which is not even guaranteed to be the gradient of any potential in general

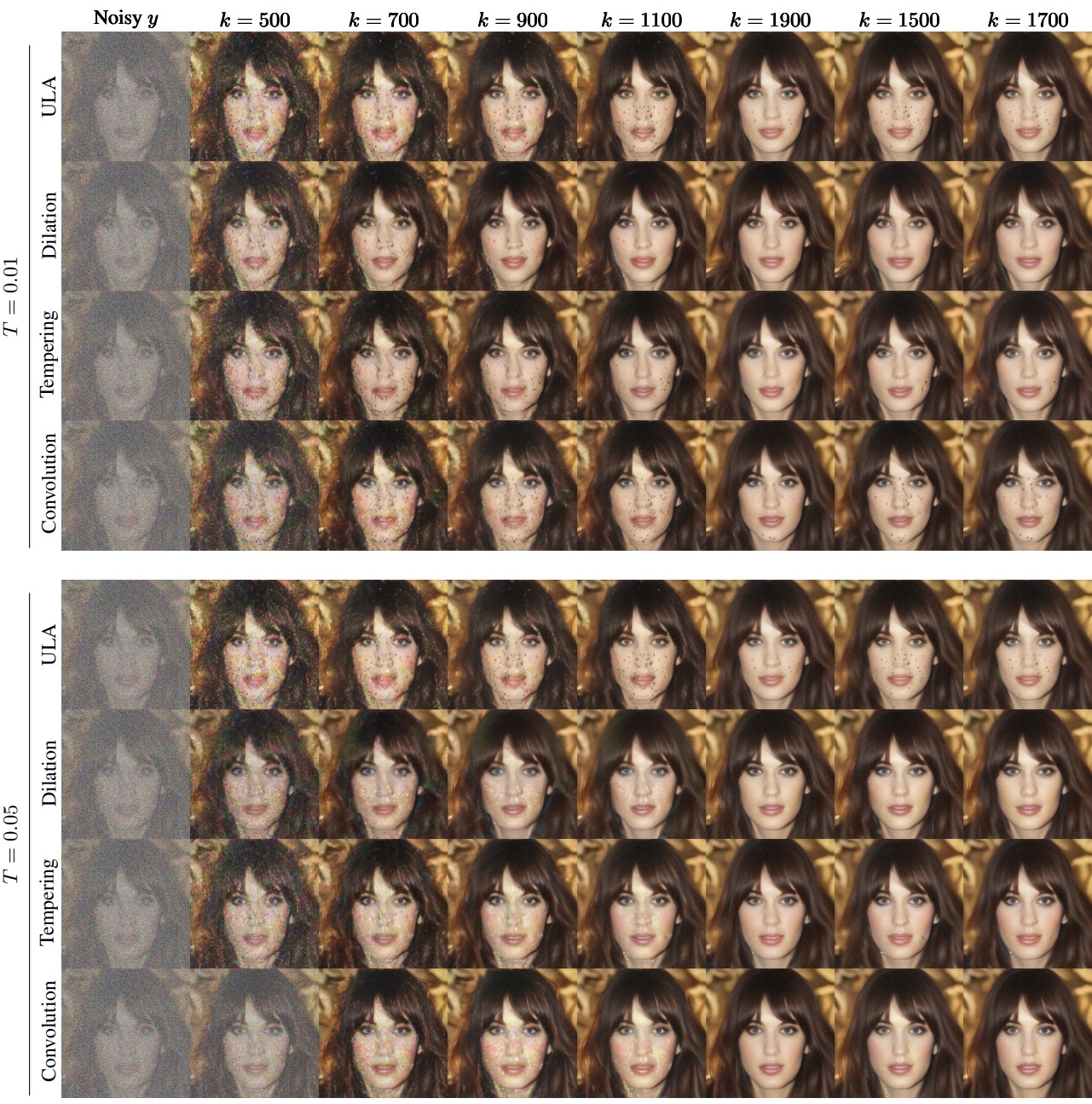

*Figure 8.* Image denoising using different annealing schemes. For each sampling method we compute the Monte-Carlo estimate of the expected value across different iterations. More precisely, if $(x_k)_k$ denotes the realization of a Markov chain with one of the methods, we plot $\bar{x}_k = \frac{1}{n} \sum_{i=1}^{n} x_{k-i*s}$ for different values of the iterations $k$ where $n$ is the sample size for the Monte Carlo estimate and $s$ a thinning parameter. We set $n = 50$ and $s = 10$. One can observe that with ULA after 1700 iterations there are still clearly visible artifacts in the face whereas with the annealing schemes (especially for $T = 0.05$) this is not the case.

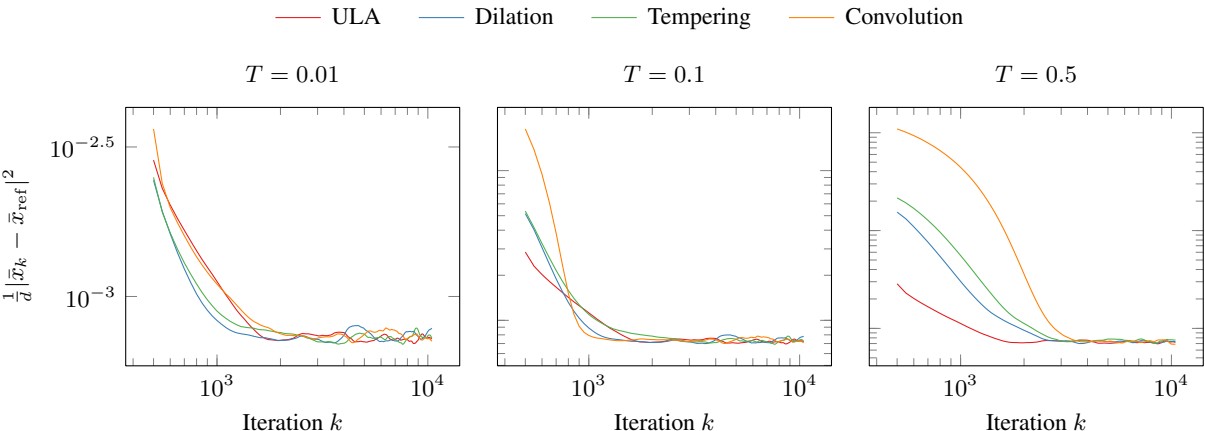

*Figure 9.* MSE convergence of the mean for image denoising. We plot the errors $\frac{1}{d}|\bar{x}_k - \bar{x}_{\text{ref}}|^2$ where we refer to the caption of Figure 8 for the definition of the Monte-Carlo estimate $\bar{x}$. As a reference $\bar{x}_{\text{ref}}$ we use the Monte-Carlo estimate $\bar{x}_k$ obtained with ULA after $k = 30000$ iterations. For $T = 0.1$ the benefits of annealing are most prominent. For $T = 0.01$ the convolutional path already closely resembles that of ULA whereas for $T = 0.5$ the annealing is too slow. Note, moreover, the steep slope of the error with the convolutional path due to favorable log-Sobolev constants. We want to emphasize, however, that the used error metric provides only a very crude quantification of accuracy.

or, equivalently, that $\mu_t$ satisfies the partial differential equation (PDE)

$$\partial_t \mu_t = \operatorname{div}(\mu_t \nabla U_{\tau(t)}) + \Delta \mu_t \tag{50}$$

in the weak sense. Corollary 6.3.2. due to Bogachev et al. (2022) guarantees that there exists some $q_t \in L^1_{\text{loc}}(\mathbb{R}^d \times (0, \infty))$ such that $\frac{d\mu_t}{d\lambda} = q_t$. Moreover, $q_t$ is continuous and strictly positive on $\mathbb{R}^d \times (0, \infty)$ and for a.e. $t$, $q_t \in W^{1,p}_{\text{loc}}(\mathbb{R}^d)$ for all $p > d + 2$ (Bogachev et al., 2002, Section 9.4). $\qquad\square$

### B.1.2. PROOF OF LEMMA 4.2

We first show a formal proof sketch under the assumption of sufficient regularity to provide some intuition and proceed with a rigorous argument below. Swapping integration and differentiation we find

$$\frac{d}{dt} \operatorname{KL}(\mu_t, \pi_{\tau(t)}) = \frac{d}{dt} \int q_t \log \frac{q_t}{p_{\tau(t)}} d\lambda = \int \partial_t q_t \log \frac{q_t}{p_{\tau(t)}} + q_t \partial_t \log \frac{q_t}{p_{\tau(t)}} d\lambda. \tag{51}$$

Now, since $q_t$ satisfies the Fokker–Planck equation (13), we find that

$$\begin{aligned}
\frac{d}{dt} \operatorname{KL}(\mu_t, \pi_{\tau(t)}) &= \int \left( \partial_{x_i} \left( -\partial_{x_i} \log p_{\tau(t)} q_t \right) + \partial_{x_i} \partial_{x_i} q_t \right) \log \frac{q_t}{p_{\tau(t)}} + \partial_t q_t - q_t \frac{\partial_t p_{\tau(t)}}{p_{\tau(t)}} d\lambda \\
&= -\int \left( -\partial_{x_i} \log p_{\tau(t)} q_t + \partial_{x_i} q_t \right) \partial_{x_i} \log \frac{q_t}{p_{\tau(t)}} + \partial_t q_t - q_t \frac{\partial_t p_{\tau(t)}}{p_{\tau(t)}} d\lambda \\
&= -\int q_t \left| \nabla \log \frac{q_t}{p_{\tau(t)}} \right|^2 - q_t \partial_t \log p_{\tau(t)} d\lambda
\end{aligned} \tag{52}$$

where we used in the last equality that $\int \partial_t q_t d\lambda = \frac{d}{dt} \int q_t d\lambda = \frac{d}{dt} 1 = 0$.

The formal arguments above are not valid in general as $q_t$ is only a weak solution of the Fokker–Planck equation. Therefore, a classical time-derivative $\partial q_t$ need not exist and even if it does, it is not clear that we may swap integration with respect to $x$ and differentiation with respect to $t$. We now proceed with the rigorous proof, which relies on mollification techniques.

*Proof.* First note that by (Bogachev et al., 2022, 7.3.10. Example, 8.2.4. Example, 8.2.5. Proposition) we have that for any $0 < t_1 < t_2 < \infty$ there exist $c_1, c_2 > 0$ such that

$$\exp\left(-c_1(|x|^2 + 1)\right) \leq q_t(x) \leq \exp\left(-c_2|x|^2\right), \quad t \in [t_1, t_2], \ x \in \mathbb{R}^d. \tag{53}$$

Let $\varepsilon_0 > 0$ such that $t_1 > \varepsilon_0$. Moreover, let $\varphi \in C_c^\infty(\mathbb{R}^d \times \mathbb{R})$ be a mollifier with, in particular, $\varphi(x,t) = 0$ for $|t| \geq 1$, denote for $0 < \varepsilon < \varepsilon_0$, $\varphi^\varepsilon(x,t) = \frac{1}{\varepsilon^{d+1}}\varphi((x,t)/\varepsilon)$ and $q_t^\varepsilon = \tilde{q} * \varphi^\varepsilon(x,t)$ with the convolution in $x$ and $t$ where $\tilde{q}$ is the zero for $t \notin [t_1 - \varepsilon_0, t_2 + \varepsilon_0]$ and identical to $q$ otherwise. It follows that $q_t^\varepsilon$ is $C^\infty$ and satisfies

$$\partial_t q_t^\varepsilon = -(q_t \partial_{x_i} \log p_{\tau(t)}) * \partial_{x_i} \varphi^\varepsilon + q_t * \partial_{x_i} \partial_{x_i} \varphi^\varepsilon, \quad (x,t) \in \mathbb{R}^d \times (t_1, t_2). \tag{54}$$

Indeed, for any $\psi \in C_c^\infty(\mathbb{R}^d \times (t_1, t_2))$ since $q_t$ solves (13) weakly we have

$$\iint \tilde{q} * \varphi^\varepsilon(x,t) \partial_t \psi(x,t) \mathrm{d}x\mathrm{d}t$$

$$= \iiiint \tilde{q}_{t-s}(x-y)\varphi^\varepsilon(y,s)\partial_t\psi(x,t)\mathrm{d}x\mathrm{d}y\mathrm{d}t\mathrm{d}s$$

$$= \iint \varphi^\varepsilon(y,s) \iint q_{t-s}(x-y)\partial_t\psi(x,t)\mathrm{d}x\mathrm{d}t\mathrm{d}y\mathrm{d}s$$

$$= \iint \varphi^\varepsilon(y,s) \iint \left(-q_{t-s}(x-y)\partial_{x_i}\log p_{\tau(t-s)}(x-y) + \partial_{x_i}q_{t-s}(x-y)\right)\partial_{x_i}\psi(x,t)\mathrm{d}x\mathrm{d}t\mathrm{d}y\mathrm{d}s \tag{55}$$

$$= \iint \partial_{x_i}\psi(x,t) \iint \left(-q_{t-s}(x-y)\partial_{x_i}\log p_{\tau(t-s)}(x-y) + \partial_{x_i}q_{t-s}(x-y)\right)\varphi^\varepsilon(y,s)\mathrm{d}y\mathrm{d}s\mathrm{d}x\mathrm{d}t$$

$$= \iint \partial_{x_i}\psi(x,t)\left(-q_t\partial_{x_i}\log p_{\tau(t)} + \partial_{x_i}q_t\right) * \varphi^\varepsilon(x)\mathrm{d}x\mathrm{d}t$$

$$= -\iint \psi(x,t)\left(-q_t\partial_{x_i}\log p_{\tau(t)} + \partial_{x_i}q_t\right) * \partial_{x_i}\varphi^\varepsilon(x)\mathrm{d}x\mathrm{d}t$$

where we used that $\varphi^\varepsilon(y,s) = 0$ for $|s| \geq \varepsilon$, $\psi(x,t) = 0$ for $t \notin [t_1, t_2]$ and consequently we can replace $\tilde{q}_{t-s}$ by $q_{t-s}$ since they coincide for $t - s \in [t_1 - \varepsilon, t_2 + \varepsilon]$. Moreover, by the exponential decay with respect to $x$ and the compact support in $t$, we have that $q_t^\varepsilon \to \tilde{q}_t$ in $L^p(\mathbb{R}^d \times \mathbb{R})$ as $\varepsilon \to 0$ for any $p \in [1, \infty)$ and pointwise a.e. by taking an appropriate subsequence. In addition $\partial_{x_i} q_t^\varepsilon \to \partial_{x_i} q_t$ in $L^2(\mathbb{R}^d \times (t_1, t_2))$ and, again by taking a subsequence, pointwise a.e. (see (Bogachev et al., 2002, Theorem 7.4.1) for a proof that $\nabla q_t \in L^2(\mathbb{R}^d \times (t_1, t_2))$). Since $q$ is moreover continuous on $\mathbb{R}^d \times (0, \infty)$ it follows that $q_t^\varepsilon \to q_t$ uniformly on any set of the form $K \times (t_1, t_2)$ with $K \subset \mathbb{R}^d$ compact. Since $q_t$ satisfies (53)[6] we find that

$$q_t^\varepsilon(x) \leq \max_{y \in B_\varepsilon(x)} q_t(y) \leq \max_{y \in B_\varepsilon(x)} \exp\left(-c_2|y|^2\right) = \mathbb{1}_{B_{2\varepsilon}(0)}(x) + \mathbb{1}_{B_{2\varepsilon}^c(0)}(x)\exp\left(-c_2\left|x - \frac{x}{|x|}\varepsilon\right|^2\right)$$

$$= \mathbb{1}_{B_{2\varepsilon}(0)}(x) + \mathbb{1}_{B_{2\varepsilon}^c(0)}(x)\exp\left(-c_2|x|^2\left(1 - \frac{\varepsilon}{|x|}\right)^2\right) \tag{56}$$

$$\leq \mathbb{1}_{B_{2\varepsilon}(0)}(x) + \mathbb{1}_{B_{2\varepsilon}^c(0)}(x)\exp\left(-c_2|x|^2/4\right)$$

$$\leq c\exp\left(-c_2|x|^2/4\right)$$

where the constants can be chosen uniformly with respect to $\varepsilon$ and $t \in [t_1, t_2]$. Similarly we have

$$q_t^\varepsilon(x) \geq \exp\left(-c_1(2|x|^2 + 1)\right) \tag{57}$$

which, in particular, implies that $q_t^\varepsilon$ has finite entropy for all $t$. For $R > 0$ let $\chi_R \in C_c^\infty(\mathbb{R}^d)$ be such that $0 \leq \chi_R \leq 1$, $\chi_R(x) = 1$ for $x \in B_R(0)$, $\chi_R(x) = 0$ for $x \notin B_{R+1}(0)$, and such that $\|\partial_{x_i}\chi_R\|_\infty$ is bounded uniformly with respect to $R$. Using the fundamental theorem of calculus, we have[7]

$$\int \chi_R q_{t_2}^\varepsilon \log \frac{q_{t_2}^\varepsilon}{p_{\tau(t_2)}}\mathrm{d}\lambda - \int \chi_R q_{t_1}^\varepsilon \log \frac{q_{t_1}^\varepsilon}{p_{\tau(t_1)}}\mathrm{d}\lambda = \iint \chi_R \frac{\mathrm{d}}{\mathrm{d}t}\left(q_t^\varepsilon \log \frac{q_t^\varepsilon}{p_{\tau(t)}}\right)\mathrm{d}t\mathrm{d}\lambda$$

$$= \iint \chi_R \frac{\mathrm{d}}{\mathrm{d}t}q_t^\varepsilon \log \frac{q_t^\varepsilon}{p_{\tau(t)}} + \chi_R q_t^\varepsilon \frac{\mathrm{d}}{\mathrm{d}t}\log \frac{q_t^\varepsilon}{p_{\tau(t)}}\mathrm{d}t\mathrm{d}\lambda \tag{58}$$

$$= A + B.$$

Our goal is in the following to take the limit on both sides of (58), first as $\varepsilon \to 0$ and afterward as $R \to \infty$.

---

[6]also for adapted $c_1, c_2$ on the slightly larger interval $[t_1 - \varepsilon_0, t_2 + \varepsilon_0]$

[7]The time integral is over $(t_1, t_2)$ which we omit for simpler notation.

**Convergence of $B$**   For the term $B$ we can estimate

$$
\begin{aligned}
B &= \iint \chi_R \frac{p_{\tau(t)}\partial_t q_t^\varepsilon - q_t^\varepsilon \partial_t p_{\tau(t)}}{p_{\tau(t)}} \mathrm{d}t\mathrm{d}\lambda \\
&= \iint \chi_R \left( \partial_t q_t^\varepsilon - q_t^\varepsilon \partial_t \log p_{\tau(t)} \right) \mathrm{d}t\mathrm{d}\lambda \\
&= \iint \chi_R \partial_t q_t^\varepsilon \mathrm{d}\lambda \mathrm{d}t - \iint \chi_R q_t^\varepsilon \partial_t \log p_{\tau(t)} \mathrm{d}t\mathrm{d}\lambda \\
&= -\iint ((-q_t \partial_{x_i} \log p_{\tau(t)}) * \varphi^\varepsilon + q_t * \partial_{x_i}\varphi^\varepsilon)\partial_{x_i}\chi_R \mathrm{d}\lambda \mathrm{d}t - \iint \chi_R q_t^\varepsilon \partial_t \log p_{\tau(t)} \mathrm{d}t\mathrm{d}\lambda \\
&= AA + BB.
\end{aligned}
\tag{59}
$$

As $\varepsilon \to 0$ we then find for $BB$

$$
\lim_{\varepsilon \to 0} BB = -\iint \chi_R q_t \partial_t \log p_{\tau(t)} \mathrm{d}t\mathrm{d}\lambda
\tag{60}
$$

due to convergence of $q_t^\varepsilon$ uniformly on compact sets and the fact that $\chi_R \partial_t \log p_{\tau(t)}$ is bounded and compactly supported with respect to $x$. For $AA$ we obtain using similar arguments that

$$
\lim_{\varepsilon \to 0} AA = -\iint ((-q_t \partial_{x_i} \log p_{\tau(t)}) + \partial_{x_i} q_t)\partial_{x_i}\chi_R \mathrm{d}\lambda \mathrm{d}t.
\tag{61}
$$

**Convergence of $A$**   Regarding the term $A$ in (58) we find

$$
\begin{aligned}
A &= \iint \chi_R((-q_t \partial_{x_i} \log p_{\tau(t)}) * \partial_{x_i}\varphi^\varepsilon + q_t * \partial_{x_i}\partial_{x_i}\varphi^\varepsilon) \log \frac{q_t^\varepsilon}{p_{\tau(t)}} \mathrm{d}\lambda \mathrm{d}t \\
&= -\iint ((-q_t \partial_{x_i} \log p_{\tau(t)}) * \varphi^\varepsilon + q_t * \partial_{x_i}\varphi^\varepsilon)\partial_{x_i} \left( \chi_R \log \frac{q_t^\varepsilon}{p_{\tau(t)}} \right) \mathrm{d}\lambda \mathrm{d}t \\
&= -\iint ((-q_t \partial_{x_i} \log p_{\tau(t)}) * \varphi^\varepsilon + q_t * \partial_{x_i}\varphi^\varepsilon)\partial_{x_i}\chi_R \log \frac{q_t^\varepsilon}{p_{\tau(t)}} \mathrm{d}\lambda \mathrm{d}t \\
&\quad -\iint ((-q_t \partial_{x_i} \log p_{\tau(t)}) * \varphi^\varepsilon + q_t * \partial_{x_i}\varphi^\varepsilon)\chi_R \partial_{x_i} \log \frac{q_t^\varepsilon}{p_{\tau(t)}} \mathrm{d}\lambda \mathrm{d}t = AA + BB
\end{aligned}
\tag{62}
$$

Again, we consider the limits as $\varepsilon \to 0$. Regarding $AA$ we first separate two terms

$$
\begin{aligned}
AA &= -\iint ((-q_t \partial_{x_i} \log p_{\tau(t)}) * \varphi^\varepsilon + q_t * \partial_{x_i}\varphi^\varepsilon)\partial_{x_i}\chi_R \log q_t^\varepsilon \mathrm{d}\lambda \\
&\quad + \iint ((-q_t \partial_{x_i} \log p_{\tau(t)}) * \varphi^\varepsilon + q_t * \partial_{x_i}\varphi^\varepsilon)\partial_{x_i}\chi_R \log p_{\tau(t)} \mathrm{d}\lambda.
\end{aligned}
\tag{63}
$$

We begin with the more difficult first term

$$
\begin{aligned}
&\left| \iint ((-q_t \partial_{x_i} \log p_{\tau(t)}) * \varphi^\varepsilon + q_t * \partial_{x_i}\varphi^\varepsilon)\partial_{x_i}\chi_R \log q_t^\varepsilon - (-q_t \partial_{x_i} \log p_{\tau(t)} + \partial_{x_i} q_t)\partial_{x_i}\chi_R \log q_t \mathrm{d}\lambda \mathrm{d}t \right| \\
&\leq \left| \iint ((-q_t \partial_{x_i} \log p_{\tau(t)}) * \varphi^\varepsilon + q_t * \partial_{x_i}\varphi^\varepsilon)\partial_{x_i}\chi_R \log q_t^\varepsilon - (-q_t \partial_{x_i} \log p_{\tau(t)} + \partial_{x_i} q_t)\partial_{x_i}\chi_R \log q_t^\varepsilon \mathrm{d}\lambda \mathrm{d}t \right| \\
&\quad + \left| \iint (-q_t \partial_{x_i} \log p_{\tau(t)} + \partial_{x_i} q_t)\partial_{x_i}\chi_R \log q_t^\varepsilon - (-q_t \partial_{x_i} \log p_{\tau(t)} + \partial_{x_i} q_t)\partial_{x_i}\chi_R \log q_t \mathrm{d}\lambda \mathrm{d}t \right| \\
&\leq \iint \left| ((-q_t \partial_{x_i} \log p_{\tau(t)}) * \varphi^\varepsilon + q_t * \partial_{x_i}\varphi^\varepsilon) - (-q_t \partial_{x_i} \log p_{\tau(t)} + \partial_{x_i} q_t) \right| \times \left| \partial_{x_i}\chi_R \log q_t^\varepsilon \right| \mathrm{d}\lambda \mathrm{d}t \\
&\quad + \iint \left| (-q_t \partial_{x_i} \log p_{\tau(t)} + \partial_{x_i} q_t)\partial_{x_i}\chi_R \right| \times \left| \log q_t^\varepsilon - \log q_t \right| \mathrm{d}\lambda \mathrm{d}t \to 0
\end{aligned}
\tag{64}
$$

as $\varepsilon \to 0$. For the first integral this follows from $L^1_{\mathrm{loc}}$ convergence[8] of $(-q_t \partial_{x_i} \log p_{\tau(t)}) * \varphi^\varepsilon + q_t * \partial_{x_i}\varphi^\varepsilon$ and uniform boundedness of $\partial_{x_i}\chi_R \log q_t^\varepsilon$ where we make use of the estimates (56) and (57). For the second term, to the contrary, we

---

[8]$L^1_{\mathrm{loc}}$ convergence, in turn, is implied by $L^2$ convergence.

use $L^1$ boundedness of $((-q_t\partial_{x_i}\log p_{\tau(t)}) + \partial_{x_i}q_t)\partial_{x_i}\chi_R$ and uniform convergence of $\log q_t^\varepsilon$ on compact sets. The latter uniform convergence follows from uniform convergence of $q_t^\varepsilon$ on sets of the form $K \times (t_1, t_2)$ for $K \subset \mathbb{R}^d$ compact together with the fact that by (56) and (57) the set

$$U := \{q_t^\varepsilon(x) \,|\, (x, t) \in K \times (t_1, t_2),\ 0 < \varepsilon < \varepsilon_0\} \tag{65}$$

satisfies $U \subset [a, b]$ for some $0 < a < b < \infty$ and the logarithm restricted to $[a, b]$ is Lipschitz. For the second term in $AA$ similar arguments can be repeated using uniform upper and lower bounds of $\log p_{\tau(t)}$ again on compact sets.

Next, we tackle $BB$. We find

$$\iint ((-q_t\partial_{x_i}\log p_{\tau(t)}) * \varphi^\varepsilon + q_t * \partial_{x_i}\varphi^\varepsilon)\chi_R\partial_{x_i}\log\frac{q_t^\varepsilon}{p_{\tau(t)}}\mathrm{d}\lambda\mathrm{d}t$$

$$= \iint ((-q_t\partial_{x_i}\log p_{\tau(t)}) * \varphi^\varepsilon + q_t * \partial_{x_i}\varphi^\varepsilon)\chi_R\partial_{x_i}\log q_t^\varepsilon\mathrm{d}\lambda\mathrm{d}t \tag{66}$$

$$- \iint ((-q_t\partial_{x_i}\log p_{\tau(t)}) * \varphi^\varepsilon + q_t * \partial_{x_i}\varphi^\varepsilon)\chi_R\partial_{x_i}\log p_{\tau(t)}\mathrm{d}\lambda\mathrm{d}t$$

and once again, start with the more difficult first term

$$\iint ((-q_t\partial_{x_i}\log p_{\tau(t)}) * \varphi^\varepsilon + q_t * \partial_{x_i}\varphi^\varepsilon)\chi_R\partial_{x_i}\log q_t^\varepsilon\mathrm{d}\lambda\mathrm{d}t$$

$$= \iint ((-q_t\partial_{x_i}\log p_{\tau(t)}) * \varphi^\varepsilon + q_t * \partial_{x_i}\varphi^\varepsilon)\chi_R\frac{\partial_{x_i}q_t^\varepsilon}{q_t^\varepsilon}\mathrm{d}\lambda\mathrm{d}t \tag{67}$$

$$= \iint \chi_R\frac{\partial_{x_i}q_t^\varepsilon}{q_t^\varepsilon}(-q_t\partial_{x_i}\log p_{\tau(t)}) * \varphi^\varepsilon + \frac{(\partial_{x_i}q_t^\varepsilon)^2}{q_t^\varepsilon}\chi_R\mathrm{d}\lambda\mathrm{d}t.$$

The second term can be understood as the squared $L^2$ norm of $\frac{\partial_{x_i}q_t^\varepsilon}{\sqrt{q_t^\varepsilon}}\sqrt{\chi_R}$ which converges if the respective function converges in $L^2$. We show the latter,

$$\left\|\frac{\partial_{x_i}q_t^\varepsilon}{\sqrt{q_t^\varepsilon}}\sqrt{\chi_R} - \frac{\partial_{x_i}q_t}{\sqrt{q_t}}\sqrt{\chi_R}\right\|_2 \leq \left\|(\partial_{x_i}q_t^\varepsilon - \partial_{x_i}q_t)\frac{\sqrt{\chi_R}}{\sqrt{q_t^\varepsilon}}\right\|_2 + \left\|\partial_{x_i}q_t\left(\frac{\sqrt{\chi_R}}{\sqrt{q_t^\varepsilon}} - \frac{\sqrt{\chi_R}}{\sqrt{q_t}}\right)\right\|_2. \tag{68}$$

Using uniform, strictly positive upper and lower bounds of $q_t^\varepsilon$ on $B_{R+1}(0) \times (t_1, t_2)$, the $L^2$ convergence of $\partial_{x_i}q_t^\varepsilon$ and uniform convergence on compact sets of $\frac{\sqrt{\chi_R}}{\sqrt{q_t^\varepsilon}}$ we find that the above tends to zero as $\varepsilon \to 0$. The second term in $BB$ is handled similarly.

**Left-hand side of (58)** Regarding the left-hand side of (58) it follows that

$$\lim_{\varepsilon \to 0}\int \chi_R q_{t_1}^\varepsilon\log\frac{q_{t_1}^\varepsilon}{p_{\tau(t_1)}}\mathrm{d}\lambda = \int \chi_R q_{t_1}\log\frac{q_{t_1}}{p_{\tau(t_1)}}\mathrm{d}\lambda \tag{69}$$

again by uniform convergence of $q^\varepsilon$ on compact sets and appropriate boundedness rendering the logarithm Lipschitz continuous. We therefore find

$$\int \chi_R q_{t_2}\log\frac{q_{t_2}}{p_{\tau(t_2)}}\mathrm{d}\lambda - \int \chi_R q_{t_1}\log\frac{q_{t_1}}{p_{\tau(t_1)}}\mathrm{d}\lambda$$

$$= -\iint (-q_t\partial_{x_i}\log p_{\tau(t)} + \partial_{x_i}q_t)\partial_{x_i}\chi_R\log\frac{q_t}{p_{\tau(t)}}\mathrm{d}\lambda\mathrm{d}t$$

$$- \iint (-q_t\partial_{x_i}\log p_{\tau(t)} + \partial_{x_i}q_t)\chi_R\partial_{x_i}\log\frac{q_t}{p_{\tau(t)}}\mathrm{d}\lambda\mathrm{d}t \tag{70}$$

$$- \iint (-q_t\partial_{x_i}\log p_{\tau(t)} + \partial_{x_i}q_t)\partial_{x_i}\chi_R\mathrm{d}\lambda\mathrm{d}t - \iint \chi_R q_t\partial_t\log p_{\tau(t)}\mathrm{d}t\mathrm{d}\lambda.$$

Next, we want to take the limit of the above as $R \to \infty$ using Lebesgue's dominated convergence. For this we note that $\chi_R \to 1$, $\partial_{x_i} \chi_R \to 0$ pointwise a.e. as $R \to \infty$. Moreover, in order to apply dominated convergence, we need to make sure that each function sequence admits an integrable upper bound. For most of the involved functions this follows directly from the exponential decay of $q_t$ with respect to $x$ and the fact that $\partial_{x_i} q_t \in L^2(\mathbb{R}^d \times (t_1, t_2))$. For the term $\partial_{x_i} q_t \log q_t$ we note that

$$\iint |\partial_{x_i} q_t \log q_t| \, \mathrm{d}\lambda \mathrm{d}t = \iint \left| \frac{\partial_{x_i} q_t}{\sqrt{q_t}} \sqrt{q_t} \log q_t \right| \mathrm{d}\lambda \mathrm{d}t \leq \left\| \frac{\partial_{x_i} q_t}{\sqrt{q_t}} \right\|_2 \| \sqrt{q_t} \log q_t \|_2 \tag{71}$$

where the second norm is finite by the bounds (56) and (57) and the first term is bounded as shown in (Bogachev et al., 2022, Theorem 7.4.1). Similar arguments apply to the term $\partial_{x_i} q_t \log p_{\tau(t)}$. Thus, taking the limit for $R \to \infty$ yields

$$\begin{aligned}
\mathrm{KL}(\mu_{t_2} | \pi_{\tau(t_2)}) - \mathrm{KL}(\mu_{t_1} | \pi_{\tau(t_1)}) &= -\iint ((-q_t \partial_{x_i} \log p_{\tau(t)}) + \partial_{x_i} q_t) \partial_{x_i} \log \frac{q_t}{p_{\tau(t)}} \mathrm{d}\lambda \mathrm{d}t - \iint q_t \partial_t \log p_{\tau(t)} \mathrm{d}\lambda \mathrm{d}t \\
&= -\iint \left( q_t \partial_{x_i} \log \frac{q_t}{p_{\tau(t)}} \right) \partial_{x_i} \log \frac{q_t}{p_{\tau(t)}} \mathrm{d}\lambda \mathrm{d}t - \iint q_t \partial_t \log p_{\tau(t)} \mathrm{d}\lambda \mathrm{d}t \\
&= -\iint \left| q_t \partial_{x_i} \log \frac{q_t}{p_{\tau(t)}} \right|^2 \mathrm{d}\lambda \mathrm{d}t - \iint q_t \partial_t \log p_{\tau(t)} \mathrm{d}\lambda \mathrm{d}t
\end{aligned} \tag{72}$$

where we used that, by Lemma B.11, $\partial_{x_i} \log q_t = \frac{\partial_{x_i} q_t}{q_t}$. From the above it is also apparent that

$$t \mapsto \mathrm{KL}(\mu_t | \pi_{\tau(t)}) \tag{73}$$

is absolutely continuous as the integral of an $L^1$ function. Consequently, $\mathrm{KL}(\mu_t | \pi_{\tau(t)})$ is a.e. differentiable with respect to $t$ and we have

$$\frac{\mathrm{d}}{\mathrm{d}t} \mathrm{KL}(\mu_t | \pi_{\tau(t)}) = -\int q_t \left| \nabla \log \frac{q_t}{p_{\tau(t)}} \right|^2 - q_t \partial_t \log p_{\tau(t)} \mathrm{d}\lambda. \tag{74}$$

concluding the proof. $\square$

Using the above lemma we can immediately derive the following lemma.

**Lemma B.1.** *Under Assumption 3.2, the diffusion $\mu_t$ satisfy for some $c > 0$*

$$\mathrm{KL}(\mu_t | \pi_{\tau(t)}) \leq \exp\left( -\int_0^t \frac{4}{C_{\mathrm{LSI}}(s)} \mathrm{d}s \right) \mathrm{KL}(\mu_0 | \pi_{\tau(0)}) + c \int_0^t \exp\left( -\int_s^t \frac{4}{C_{\mathrm{LSI}}(\sigma)} \mathrm{d}\sigma \right) |\dot{\tau}(s)| \mathrm{d}s \tag{75}$$

*Proof.* From Lemma 4.2 and LSI together with Lemma B.8 it follows that

$$\frac{\mathrm{d}}{\mathrm{d}t} \mathrm{KL}(\mu_t | \pi_{\tau(t)}) = -\int q_t \left| \nabla \log \frac{q_t}{p_{\tau(t)}} \right|^2 - q_\tau \partial_\tau \log p_{\tau(t)} \mathrm{d}\lambda \leq -\frac{4}{C_{\mathrm{LSI}}(t)} \mathrm{KL}(\mu_t | \pi_{\tau(t)}) - \int q_t \partial_t \log p_{\tau(t)} \mathrm{d}\lambda. \tag{76}$$

Regarding the last term, we find $\partial_\tau \log p_\tau = -\partial_\tau U_\tau - \frac{\partial_\tau Z_\tau}{Z_\tau}$. For $\partial_\tau Z_\tau$, in turn, we can estimate

$$\frac{1}{Z_\tau} \partial_t Z_\tau = \frac{1}{Z_\tau} \partial_\tau \int \exp(-U_\tau(x)) \mathrm{d}x = \frac{1}{Z_\tau} \int -\partial_\tau U_\tau(x) \exp(-U_\tau(x)) \mathrm{d}x = \int -\partial_\tau U_\tau p_\tau \mathrm{d}\lambda \tag{77}$$

where we used dominated convergence to swap the integral and the derivative. Thus,

$$\left| \int q_t \partial_t \log p_{\tau(t)} \mathrm{d}\lambda \right| = \left| \dot{\tau}(t) \int \partial_\tau U_{\tau(t)} (p_{\tau(t)} - q_t) \mathrm{d}\lambda \right| \leq |\dot{\tau}(t)| \int c(|x|^2 + 1)(p_{\tau(t)}(x) - q_t(x)) \mathrm{d}x \leq c |\dot{\tau}(t)| \tag{78}$$

with adapted $c$ using Lemmas B.2 and B.3. In total we find

$$\frac{\mathrm{d}}{\mathrm{d}t} \mathrm{KL}(\mu_t | \pi_{\tau(t)}) \leq -\frac{4}{C_{\mathrm{LSI}}(t)} \mathrm{KL}(\mu_t | \pi_{\tau(t)}) + c |\dot{\tau}(t)| \tag{79}$$

Then, by Lemma B.10 the assertion follows. $\square$

### B.1.3. PROOF OF LEMMA 4.3

*Proof.* We compute

$$\mathrm{KL}(\mu_t|\pi) = \mathrm{KL}(\mu_t|\pi) - \mathrm{KL}(\mu_t|\pi_{\tau(t)}) + \mathrm{KL}(\mu_t|\pi_{\tau(t)}) = \int q_t \log \frac{p_{\tau(t)}}{p} \mathrm{d}\lambda + \mathrm{KL}(\mu_t|\pi_{\tau(t)}). \tag{80}$$

By Lemma B.9, $\log \frac{p_{\tau(t)}(x)}{p(x)} \leq c_1 \tau(t) + c_2 \tau(t)|x|^2$. Therefore, using Lemmas B.1 and B.3 the result follows. □

### B.1.4. PROOF OF THEOREM 4.4

*Proof.* Let $\delta > 0$ be arbitrary. We can pick $t_1 > 0$ such that $\tau(t_1) < \delta$. Afterward, pick $t_2 > t_1$ sufficiently large such that $\exp\left(-\int_{t_1}^{t_2} \frac{4}{C_{\mathrm{LSI}}(s)}\mathrm{d}s\right) < \delta$ which is possible since by uniform dissipativity $C_{\mathrm{LSI}}(t) \leq \overline{C_{\mathrm{LSI}}}$ for some $\overline{C_{\mathrm{LSI}}} < \infty$ and all $t$. Moreover, pick $t_2$ sufficiently large such that $\tau(t) < \delta$ for $t > t_2$. We find for any $t > t_2$ by Lemma 4.3

$$
\begin{aligned}
\mathrm{KL}(\mu_t|\pi) &\leq c\tau(t) + \exp\left(-\int_0^t \frac{4}{C_{\mathrm{LSI}}(s)}\mathrm{d}s\right)\mathrm{KL}(\mu_0|\pi_{\tau(0)}) + c\int_0^t \exp\left(-\int_s^t \frac{4}{C_{\mathrm{LSI}}(\sigma)}\mathrm{d}\sigma\right)|\dot\tau(s)|\mathrm{d}s \\
&\leq c\tau(t) + \mathrm{KL}(\mu_0|\pi_{\tau(0)})\exp\left(-\int_0^t \frac{4}{C_{\mathrm{LSI}}(s)}\mathrm{d}s\right) + c\int_0^{t_1} \exp\left(-\int_s^t \frac{4}{C_{\mathrm{LSI}}(\sigma)}\mathrm{d}\sigma\right)|\dot\tau(s)|\mathrm{d}s \\
&\quad + c\int_{t_1}^t \exp\left(-\int_s^t \frac{4}{C_{\mathrm{LSI}}(\sigma)}\mathrm{d}\sigma\right)|\dot\tau(s)|\mathrm{d}s \\
&= c\tau(t) + \mathrm{KL}(\mu_0|\pi_{\tau(0)})\exp\left(-\int_0^t \frac{4}{C_{\mathrm{LSI}}(s)}\mathrm{d}s\right) - c\int_0^{t_1}\exp\left(-\int_{t_1}^t \frac{4}{C_{\mathrm{LSI}}(\sigma)}\mathrm{d}\sigma\right)\dot\tau(s)\mathrm{d}s - c\int_{t_1}^t \dot\tau(s)\mathrm{d}s \\
&= c\tau(t) + \mathrm{KL}(\mu_0|\pi_{\tau(0)})\exp\left(-\int_0^t \frac{4}{C_{\mathrm{LSI}}(s)}\mathrm{d}s\right) - c(\tau(t_1) - \tau(0))\exp\left(-\int_{t_1}^t \frac{4}{C_{\mathrm{LSI}}(\sigma)}\mathrm{d}\sigma\right) \\
&\quad - c(\tau(t) - \tau(t_1)) \\
&\leq c\tau(t) + \mathrm{KL}(\mu_0|\pi_{\tau(0)})\exp\left(-\int_0^t \frac{4}{C_{\mathrm{LSI}}(s)}\mathrm{d}s\right) + c\tau(0)\exp\left(-\int_{t_1}^t \frac{4}{C_{\mathrm{LSI}}(\sigma)}\mathrm{d}\sigma\right) + c\tau(t) + c\tau(t_1) \\
&\leq c\delta + \mathrm{KL}(\mu_0|\pi_{\tau(0)})\delta + c\tau(0)\delta + c\delta + c\delta
\end{aligned}
\tag{81}
$$

picking $\delta$ sufficiently small concludes the proof.

To obtain the worst case bound on the complexity we use the estimate $C_{\mathrm{LSI}} \leq \overline{C_{\mathrm{LSI}}}$ in (81) to obtain

$$
\begin{aligned}
\mathrm{KL}(\mu_t|\pi) &\leq c(\tau(t) + \tau(t_1)) + \mathrm{KL}(\mu_0|\pi_{\tau(0)})\exp\left(-\int_0^t \frac{4}{\overline{C_{\mathrm{LSI}}}}\mathrm{d}s\right) - c\int_0^{t_1}\exp\left(-\int_s^t \frac{4}{\overline{C_{\mathrm{LSI}}}}\mathrm{d}\sigma\right)\dot\tau(s)\mathrm{d}s \\
&\leq c(\tau(t) + \tau(t_1)) + \mathrm{KL}(\mu_0|\pi_{\tau(0)})\exp\left(-\frac{4t}{\overline{C_{\mathrm{LSI}}}}\right) - c\int_0^{t_1}\exp\left(-\frac{4(t-s)}{\overline{C_{\mathrm{LSI}}}}\mathrm{d}\sigma\right)\dot\tau(s)\mathrm{d}s \\
&\leq c(\tau(t) + \tau(t_1)) + \mathrm{KL}(\mu_0|\pi_{\tau(0)})\exp\left(-\frac{4t}{\overline{C_{\mathrm{LSI}}}}\right) + c\|\dot\tau\|_\infty \int_0^{t_1}\exp\left(-\frac{4(t-s)}{\overline{C_{\mathrm{LSI}}}}\right)\mathrm{d}s \\
&\leq c(\tau(t) + \tau(t_1)) + \mathrm{KL}(\mu_0|\pi_{\tau(0)})\exp\left(-\frac{4t}{\overline{C_{\mathrm{LSI}}}}\right) + c\left[\exp\left(-\frac{4(t-t_1)}{\overline{C_{\mathrm{LSI}}}}\right) - \exp\left(-\frac{4t}{\overline{C_{\mathrm{LSI}}}}\right)\right] \\
&\leq c(\tau(t) + \tau(t_1)) + c\exp\left(-\frac{4(t-t_1)}{\overline{C_{\mathrm{LSI}}}}\right).
\end{aligned}
\tag{82}
$$

Therefore, in order to reach accuracy $\mathrm{KL}(\mu_t|\pi) \leq \varepsilon$, ignoring constants, we require $\tau(t_1) = \mathcal{O}(\varepsilon)$ and $t - t_1 = \mathcal{O}(\log(\varepsilon^{-1}))$. We want to stress here again, that this estimate is very pessimistic as the benefits of annealing are not taken into account due to the uniform estimation of the LSI constant. □

### B.1.5. HELPER RESULTS

**Lemma B.2.** *Let $p \in \mathbb{N}$. If $\mu_0$ admits a finite $4p$-th moment, then the $2p$-th moments of $\mu_t$ are bounded uniformly in $t$.*

*Proof.* Any solution $(X_t)_t$ of (4) has the property that $\mathbb{E}[\sup_{0 \le t \le T} |X_t|^{2p}] < \infty$ for any $T > 0$ (Kloeden et al., 2012, Exercise 4.5.5) This bound together with a.e. continuity of the sample paths yields continuity of $t \mapsto \mathbb{E}[|X_t|^{2p}]$ by dominated convergence. By Itô's lemma and uniform dissipativity we have

$$\mathrm{d}|X_t|^{2p} = 2p|X_t|^{2(p-1)} \left\{ (X_t)_i \partial_{x_i} \log p_{\tau(t)}(X_t) + d + 2(p-1) \right\} \mathrm{d}t + \sqrt{8}p|X_t|^{2(p-1)} X_t^\top \mathrm{d}W_t$$
$$\le 2p|X_t|^{2(p-1)} \left\{ -a_{\tau(t)}|X_t|^2 + b_{\tau(t)} + d + 2(p-1) \right\} \mathrm{d}t + \sqrt{8}p|X_t|^{2(p-1)} X_t^\top \mathrm{d}W_t.$$

As a consequence it follows for any $t, h > 0$

$$\mathbb{E}[|X_{t+h}|^{2p} - |X_t|^{2p}] \le \mathbb{E}\left[ \int_t^{t+h} 2p|X_s|^{2(p-1)} \left\{ -a_{\tau(t)}|X_s|^2 + b_{\tau(t)} + d + 2(p-1) \right\} \mathrm{d}s \right]$$
$$\le \int_t^{t+h} -2pa_{\tau(t)} \mathbb{E}\left[|X_s|^{2p}\right] + 2p(b_{\tau(t)} + d + 2(p-1)) \mathbb{E}\left[|X_s|^{2(p-1)}\right] \mathrm{d}s$$
$$\le \int_t^{t+h} -C_1 \mathbb{E}\left[|X_s|^{2p}\right] + C_2 \mathbb{E}\left[\mathbb{1}_{|X_s| \le \alpha}|X_s|^{2(p-1)} + \mathbb{1}_{|X_s| > \alpha}|X_s|^{-2}|X_s|^{2p}\right] \mathrm{d}s \qquad (83)$$
$$\le \int_t^{t+h} -C_1 \mathbb{E}\left[|X_s|^{2p}\right] + C_2 \mathbb{E}\left[\alpha^{2(p-1)} + \alpha^{-2}|X_s|^{2p}\right] \mathrm{d}s$$
$$\le \int_t^{t+h} -(C_1 - \alpha^{-2}C_2) \mathbb{E}\left[|X_s|^{2p}\right] + C_2|\alpha|^{2(p-1)} \mathrm{d}s$$

where $C_1 = 2pa_{\tau(t)} \ge 2pa$, $C_2 = 2p(b_{\tau(t)} + d + 2(p-1)) \le 2p(b + d + 2(p-1))$, and $\alpha > 0$ above is chosen sufficiently large so that $C_1 - \alpha^{-2}C_2 > 0$. Since $t \mapsto \mathbb{E}[|X_t|^{2p}]$ is a.e. differentiable as an absolutely continuous function, dividing (83) by $h > 0$ and letting $h \to 0$ yields

$$\frac{\mathrm{d}}{\mathrm{d}t}\left(\mathbb{E}[|X_t|^{2p}] - \frac{C_2\alpha^{2(p-1)}}{C_1 - \alpha^{-2}C_2}\right) \le -(C_1 - \alpha^{-2}C_2)\left(\mathbb{E}\left[|X_t|^{2p}\right] - \frac{C_2\alpha^{2(p-1)}}{C_1 - \alpha^{-2}C_2}\right). \qquad (84)$$

Grönwall's inequality then yields

$$\mathbb{E}[|X_t|^{2p}] \le \left(\mathbb{E}[|X_0|^{2p}] - \frac{C_2|\alpha|^{2(p-1)}}{C_1 - \alpha^{-2}C_2}\right) \exp\left(-(C_1 - \alpha^{-2}C_2)t\right) + \frac{C_2|\alpha|^{2(p-1)}}{C_1 - \alpha^{-2}C_2}. \qquad (85)$$

which is bounded uniformly in $t$ by the bounds on $C_1$ and $C_2$. $\qquad \square$

**Lemma B.3.** *All moments of $\pi_\tau$ are bounded uniformly in $\tau$.*

*Proof.* The dissipativity and Lipschitz continuity of $\nabla U_\tau$ immediately implies that $\pi_\tau$ has finite $p$-th moments for all $p \in \mathbb{N}$ and it remains to show that this holds uniformly in $\tau$. To this end, let $\tau_0 > 0$ be arbitrary and fix $\tau(t) = \tau_0$, that is we consider now the Langevin diffusion for the fixed target $\pi_{\tau_0}$. Initializing $(X_t)_t$ with the stationary distribution $X_0 \sim \pi_{\tau_0}$, we find that $X_t \sim \pi_{\tau_0}$ for all $t$. Then (85) yields

$$\int |x|^{2p} \mathrm{d}\pi_{\tau_0}(x) \le \left(\mathbb{E}[|X_0|^{2p}] - \frac{C_2|\alpha|^{2(p-1)}}{C_1 - \alpha^{-2}C_2}\right) \exp\left(-(C_1 - \alpha^{-2}C_2)t\right) + \frac{C_2|\alpha|^{2(p-1)}}{C_1 - \alpha^{-2}C_2}. \qquad (86)$$

Since the above is true for all $t$, we may let $t \to \infty$ and find $\int |x|^{2p} \mathrm{d}\pi_{\tau_0}(x) \le \frac{C_2|\alpha|^{2(p-1)}}{C_1 - \alpha^{-2}C_2}$ which is bounded uniformly in $\tau$. $\qquad \square$

### B.2. Discretization Analysis

First we note that the discretization (18) may be interpreted as a solution of the SDE

$$\mathrm{d}\overline{X}_t = -\overline{\nabla U}_t(\overline{X})\mathrm{d}t + \sqrt{2}\mathrm{d}W_t \qquad (87)$$

where for any function $x : (0, \infty) \to \mathbb{R}^d$, $\overline{\nabla U}_t(x) = \sum_k \nabla U_{t_k}(x_{t_k})\mathbb{1}_{[t_k, t_{k+1})}(t)$. In the following we will denote $\hat\mu_t := \mathrm{law}(\overline{X}_t)$ and in the case $t = t_k$, in a slight abuse of notation we will write $\hat\mu_k = \hat\mu_{t_k}$. Moreover, we denote the density as $\hat q_t = \frac{\mathrm{d}\hat\mu_t}{\mathrm{d}\lambda}$.

### B.2.1. PROOF OF LEMMA 4.5

The proof is similar to Lemma 3 in (Vempala & Wibisono, 2019).

*Proof.* We consider for simplicity of notation the step from $k = 0$ to $k = 1$. We find that for $t \in (0, t_1]$ the density of $\overline{X}_t | \overline{X}_0 = x_0$ is $\hat{q}_{t|0}(x|x_0) = \frac{1}{(4\pi t)^{d/2}} \exp\left(-\frac{1}{4t}|x + t\nabla U_{t_0}(x_0) - x_0|^2\right)$. The unconditional density can therefore be expressed as $\hat{q}_t(x_t) = \int \hat{q}_{t|0}(x|x_0)\hat{q}_0(x_0)\mathrm{d}x_0$. It is well-known (*cf.* Lemma B.12), that $\hat{q}_{t|0}(x|x_0)$ is a classical solution of the diffusion equation

$$\partial_t \hat{q}_{t|0}(x|x_0) = \operatorname{div}(\nabla U_{\tau(t_0)}(x_0)\hat{q}_{t|0}(x|x_0)) + \Delta \hat{q}_{t|0}(x|x_0) \tag{88}$$

where the divergence and Laplacian are with respect to $x$. As a consequence it follows that $\hat{q}_t(x_t)$ satisfies in the classical sense the PDE

$$
\begin{aligned}
\partial_t \hat{q}_t(x) &= \partial_t \int \hat{q}_{t|0}(x|x_0)\hat{q}_0(x_0)\mathrm{d}x_0 \\
&= \int \left\{ \operatorname{div}(\nabla U_{\tau(t_0)}(x_0)\hat{q}_{t|0}(x|x_0)) + \Delta \hat{q}_{t|0}(x|x_0) \right\} \hat{q}_0(x_0)\mathrm{d}x_0 \\
&= \operatorname{div} \int \nabla U_{\tau(t_0)}(x_0)\hat{q}_{t|0}(x|x_0)\hat{q}_0(x_0)\mathrm{d}x_0 + \Delta \int \hat{q}_{t|0}(x|x_0)\hat{q}_0(x_0)\mathrm{d}x_0 \\
&= \operatorname{div} \left( \int \nabla U_{\tau(t_0)}(x_0)\hat{q}_{0|t}(x_0|x)\mathrm{d}x_0 \hat{q}_t(x) \right) + \Delta \hat{q}_t(x) \\
&= \operatorname{div} \left( \mathbb{E}\left[\nabla U_{\tau(t_0)}(X_0) \big| X_t = x\right] \hat{q}_t(x) \right) + \Delta \hat{q}_t(x)
\end{aligned}
\tag{89}
$$

where we applied dominated convergence multiple times, first to pull $\partial_t$ into the integral and afterward the divergence and the Laplacian out of the integral. This is allowed since the functions

$$\partial_t \hat{q}_{t|0}(x|x_0)\hat{q}_0(x_0), \ \nabla U_{\tau(t_0)}(x_0)\partial_{x_i}\hat{q}_{t|0}(x|x_0)\hat{q}_0(x_0), \ \partial_{x_i}^2 \hat{q}_{t|0}(x|x_0)\hat{q}_0(x_0)$$

all admit $x_0$-integrable upper bounds locally uniformly with respect to the respective variable of differentiation. Indeed, for the first term we note for every $t \in (0, t_1)$ due to the exponential nature of $q_{t|0}(x|x_0)$ and the fact that $x_0 \mapsto x_0 - t\nabla U_{\tau(t_0)}(x_0)$ grows linearly for $t < \frac{a_0}{L_0^2} \le \frac{1}{L_0}$, there exists $\varepsilon > 0$ such that

$$\sup_{s \in (t-\varepsilon, t+\varepsilon)} \sup_{x_0} |\partial_t \hat{q}_{t|0}(x|x_0)| < \infty \tag{90}$$

so that $x_0 \mapsto \partial_t \hat{q}_{t|0}(x|x_0)\hat{q}_0(x_0)$ admits an integrable upper bound. Similar arguments apply for the other terms. Let us for simplicity denote $b(x) = \mathbb{E}\left[\nabla U_{\tau(t_0)}(X_0) \big| X_t = x\right]$. Next we claim that, again by dominated convergence, similar to the continuous diffusion we have

$$\frac{\mathrm{d}}{\mathrm{d}t} \operatorname{KL}(\hat{\mu}_t | \pi_{\tau(t_0)}) = \int (\partial_t \hat{q}_t(x_t)) \log \frac{\hat{q}_t(x_t)}{p_{\tau(t_0)}(x_t)} + \hat{q}_t(x_t)\partial_t \log \frac{\hat{q}_t(x_t)}{p_{\tau(t_0)}(x_t)}\mathrm{d}x_t. \tag{91}$$

To this end, take $\chi_R \in C_c^\infty(\mathbb{R}^d)$ such that $0 \le \chi_R \le 1$, $\chi_R(x) = 1$ for $x \in B_R(0)$, $\chi_R(x) = 0$ for $x \notin B_R(0)$ and $\partial_{x_i}\chi_R(x)$ is bounded uniformly with respect to $x$ and $R$. As in the proof of Lemma 4.2 we find for $0 < s_1 < s_2 < t_1$ using the fundamental theorem of calculus

$$
\begin{aligned}
&\int \chi_R(x)\hat{q}_{s_2}(x) \log \frac{\hat{q}_{s_2}(x)}{p_{\tau(t_0)}}\mathrm{d}x - \int \chi_R(x)\hat{q}_{s_1}(x) \log \frac{\hat{q}_{s_1}(x)}{p_{\tau(t_0)}}\mathrm{d}x \\
&= \iint_{s_1}^{s_2} \chi_R(x)\partial_t \hat{q}_t(x) \log \frac{\hat{q}_t(x)}{p_{\tau(t_0)}(x)} + \chi_R(x)\partial_t \hat{q}_t(x)\mathrm{d}t\mathrm{d}x \\
&= \iint \chi_R(x) \left\{ \partial_{x_i}(b_i(x)\hat{q}_t(x)) + \partial_{x_i}\partial_{x_i}\hat{q}_t(x) \right\} \log \frac{\hat{q}_t(x)}{p_{\tau(t_0)}(x)} + \chi_R(x)\partial_t \hat{q}_t(x)\mathrm{d}t\mathrm{d}x \\
&= -\iint \left\{ b(x)\hat{q}_t(x) + \partial_{x_i}\hat{q}_t(x) \right\} \partial_{x_i} \left( \chi_R(x) \log \frac{\hat{q}_t(x)}{p_{\tau(t_0)}(x)} \right) + \chi_R(x)\partial_t \hat{q}_t(x)\mathrm{d}t\mathrm{d}x \\
&= -\iint \left\{ b(x)\hat{q}_t(x) + \partial_{x_i}\hat{q}_t(x) \right\} \left\{ \partial_{x_i}\chi_R(x) \log \frac{\hat{q}_t(x)}{p_{\tau(t_0)}(x)} + \chi_R(x)\partial_{x_i} \log \frac{\hat{q}_t(x)}{p_{\tau(t_0)}(x)} \right\} \mathrm{d}t\mathrm{d}x \\
&\quad + \iint \chi_R(x)\partial_t \hat{q}_t(x)\mathrm{d}t\mathrm{d}x.
\end{aligned}
\tag{92}
$$

We want to let $R \to \infty$. For the third term we find

$$\iint \chi_R(x) \partial_t \hat{q}_t(x) \mathrm{d}t \mathrm{d}x = \int \chi_R(x) \int \partial_t \hat{q}_t(x) \mathrm{d}t \mathrm{d}x = \int \chi_R(x) \left( \hat{q}_{s_2}(x) - \hat{q}_{s_1}(x) \right) \mathrm{d}x, \tag{93}$$

which tends to zero as $R \to \infty$ by dominated or monotone convergence. To apply dominated convergence to the remaining terms, it suffices to show that the expression

$$\{|b(x)\hat{q}_t(x)| + |\partial_{x_i}\hat{q}_t(x)|\} \left\{ \left| \log \frac{\hat{q}_t(x)}{p_{\tau(t_0)}(x)} \right| + \left| \partial_{x_i} \log \frac{\hat{q}_t(x)}{p_{\tau(t_0)}(x)} \right| \right\} \tag{94}$$

is integrable with respect to $(x, t)$. Since locally in $t$ the right term can be bounded by $c(|x|^2 + 1)$ for some $c > 0$ we estimate

$$\begin{aligned}
\iint &\{|b(x)\hat{q}_t(x)| + |\partial_{x_i}\hat{q}_t(x)|\} \left\{ \left| \log \frac{\hat{q}_t(x)}{p_{\tau(t_0)}(x)} \right| + \left| \chi_R(x) \partial_{x_i} \log \frac{\hat{q}_t(x)}{p_{\tau(t_0)}(x)} \right| \right\} \mathrm{d}x \mathrm{d}t \\
&\leq \iint \{|b(x)\hat{q}_t(x)| + |\partial_{x_i}\hat{q}_t(x)|\} c(|x|^2 + 1) \mathrm{d}x \mathrm{d}t \\
&\leq \iint \left\{ \int |\nabla U_{\tau(t_0)}(x_0)| \hat{q}_{t|0}(x|x_0) \hat{q}_0(x_0) \mathrm{d}x_0 + |\partial_{x_i} \hat{q}_t(x)| \right\} c(|x|^2 + 1) \mathrm{d}x \mathrm{d}t.
\end{aligned} \tag{95}$$

Regarding the first term we find with the transformation $x = (x_0 - t\nabla U_{\tau(t_0)}(x_0)) + z$ the bound

$$\begin{aligned}
\iiint &|\nabla U_{\tau(t_0)}(x_0)| \hat{q}_{t|0}(x|x_0) c(|x|^2 + 1) \hat{q}_0(x_0) \mathrm{d}x_0 \mathrm{d}x \mathrm{d}t \\
&= \iiint |\nabla U_{\tau(t_0)}(x_0)| \mathcal{N}(x; x_0 - t\nabla U_{\tau(t_0)}(x_0), 2t) c(|x|^2 + 1) \hat{q}_0(x_0) \mathrm{d}x_0 \mathrm{d}x \mathrm{d}t \\
&= \iiint |\nabla U_{\tau(t_0)}(x_0)| \mathcal{N}(z; 0, 2t) c(|x_0 - t\nabla U_{\tau(t_0)}(x_0) + z|^2 + 1) \hat{q}_0(x_0) \mathrm{d}x_0 \mathrm{d}z \mathrm{d}t \\
&\leq \iiint |\nabla U_{\tau(t_0)}(x_0)| \mathcal{N}(z; 0, 2t) c(2|x_0 - t\nabla U_{\tau(t_0)}(x_0)|^2 + 2|z|^2 + 1) \hat{q}_0(x_0) \mathrm{d}x_0 \mathrm{d}z \mathrm{d}t \\
&\leq \iint |\nabla U_{\tau(t_0)}(x_0)| c(2|x_0 - t\nabla U_{\tau(t_0)}(x_0)|^2 + 4t + 1) \hat{q}_0(x_0) \mathrm{d}x_0 \mathrm{d}t < \infty
\end{aligned} \tag{96}$$

where finiteness follows under the assumption that $\hat{q}_0$ admits a finite third moment and Lipschitz continuity of $\nabla U_{\tau(t_0)}$. Regarding the second term in (95), on the other hand, we compute

$$\begin{aligned}
\iint &|\partial_{x_i} \hat{q}_t(x)| c(|x|^2 + 1) \mathrm{d}x \mathrm{d}t \\
&\leq \iiint \frac{1}{2t} |x_i - ((x_0)_i - t\partial_{x_i} U_{\tau(t_0)}(x_0)| \mathcal{N}(x; x_0 - t\nabla U_{\tau(t_0)}(x_0), 2t) c(|x|^2 + 1) \hat{q}_0(x_0) \mathrm{d}x_0 \mathrm{d}x \mathrm{d}t \\
&\leq \iiint \frac{1}{t} \mathcal{N}(x; x_0 - t\nabla U_{\tau(t_0)}(x_0), 2t) c(|x|^3 + |x|^2|x_0| + |x_0| + 1) \hat{q}_0(x_0) \mathrm{d}x_0 \mathrm{d}x \mathrm{d}t \\
&\leq c \iiint \frac{1}{t} \mathcal{N}(z; 0, 2t)(|x_0 - t\nabla U_{\tau(t_0)}(x_0) + z|^3 + |x_0 - t\nabla U_{\tau(t_0)}(x_0) + z|^2|x_0| + |x_0| + 1) \hat{q}_0(x_0) \mathrm{d}x_0 \mathrm{d}x \mathrm{d}t < \infty
\end{aligned} \tag{97}$$

again since the third moment of $q_0$ is finite. Thus, letting $R \to \infty$ in (92) yields

$$\begin{aligned}
\int \hat{q}_{s_2}(x) \log \frac{\hat{q}_{s_2}(x)}{p_{\tau(t_0)}} \mathrm{d}x - \int \hat{q}_{s_1}(x) \log \frac{\hat{q}_{s_1}(x)}{p_{\tau(t_0)}} \mathrm{d}x &= -\iint \{b(x)\hat{q}_t(x) + \partial_{x_i}\hat{q}_t(x)\} \partial_{x_i} \log \frac{\hat{q}_t(x)}{p_{\tau(t_0)}(x)} \mathrm{d}t \mathrm{d}x \\
&= -\iint \hat{q}_t(x) \{b(x) + \partial_{x_i} \log \hat{q}_t(x)\} \partial_{x_i} \log \frac{\hat{q}_t(x)}{p_{\tau(t_0)}(x)} \mathrm{d}t \mathrm{d}x.
\end{aligned} \tag{98}$$

This, in turn, implies that for a.e. $t$

$$
\begin{aligned}
\frac{\mathrm{d}}{\mathrm{d}t} \mathrm{KL}(\hat{\mu}_t | \pi_{\tau(t_0)}) &= -\int \hat{q}_t(x) \left\{ b(x) + \partial_{x_i} \log \hat{q}_t(x) \right\} \partial_{x_i} \log \frac{\hat{q}_t(x)}{p_{\tau(t_0)}(x)} \mathrm{d}x \\
&= -\int \hat{q}_t(x) \left\{ b(x) + \partial_{x_i} \log \hat{p}_{\tau(t_0)}(x) - \partial_{x_i} \log \hat{p}_{\tau(t_0)}(x) + \partial_{x_i} \log \hat{q}_t(x) \right\} \partial_{x_i} \log \frac{\hat{q}_t(x)}{p_{\tau(t_0)}(x)} \mathrm{d}x \quad (99) \\
&= -\int \hat{q}_t(x) \left\{ b(x) - \partial_{x_i} U_{\tau(t_0)}(x) \right\} \partial_{x_i} \log \frac{\hat{q}_t(x)}{p_{\tau(t_0)}(x)} \mathrm{d}x - \int \left| \partial_{x_i} \log \frac{\hat{q}_t(x)}{p_{\tau(t_0)}(x)} \right|^2 \mathrm{d}x
\end{aligned}
$$

The first integral can be estimated as

$$
\begin{aligned}
&\left| \int \hat{q}_t(x) \left\{ b_i(x) - \partial_{x_i} U_{\tau(t_0)}(x) \right\} \partial_{x_i} \log \frac{\hat{q}_t(x)}{p_{\tau(t_0)}(x)} \mathrm{d}x \right| \\
&= \left| \int \hat{q}_t(x) \left\{ \int \partial_{x_i} U_{\tau(t_0)}(x_0) \hat{q}_{0|t}(x_0|x) \mathrm{d}x_0 - \partial_{x_i} U_{\tau(t_0)}(x) \right\} \partial_{x_i} \log \frac{\hat{q}_t(x)}{p_{\tau(t_0)}(x)} \mathrm{d}x \right| \\
&\leq \iint \left| \partial_{x_i} U_{\tau(t_0)}(x_0) - \partial_{x_i} U_{\tau(t_0)}(x) \right| \left| \partial_{x_i} \log \frac{\hat{q}_t(x)}{p_{\tau(t_0)}(x)} \right| \hat{q}_t(x) \hat{q}_{0|t}(x_0|x) \mathrm{d}x_0 \mathrm{d}x \\
&\leq \iint L_{\tau(t_0)} |x_0 - x| \left| \nabla \log \frac{\hat{q}_t(x)}{p_{\tau(t_0)}(x)} \right| \hat{q}_{0,t}(x_0, x) \mathrm{d}x_0 \mathrm{d}x \\
&\leq \iint \frac{1}{2\varepsilon} L_{\tau(t_0)}^2 |x_0 - x|^2 \hat{q}_{0,t}(x_0, x) \mathrm{d}x_0 \mathrm{d}x + \iint \frac{\varepsilon}{2} \left| \nabla \log \frac{\hat{q}_t(x)}{p_{\tau(t_0)}(x)} \right|^2 \hat{q}_{0,t}(x_0, x) \mathrm{d}x_0 \mathrm{d}x \\
&\leq \iint \frac{1}{2\varepsilon} L_{\tau(t_0)}^2 |x_0 - x|^2 \hat{q}_{0,t}(x_0, x) \mathrm{d}x_0 \mathrm{d}x + \frac{\varepsilon}{2} \int \left| \nabla \log \frac{\hat{q}_t(x)}{p_{\tau(t_0)}(x)} \right|^2 \hat{q}_t(x) \mathrm{d}x.
\end{aligned}
\quad (100)
$$

The first term can be expressed using $\mathbb{E}[|X_t - X_0|^2] = \mathbb{E}[|-t\nabla U_{\tau(t_0)}(X_0) + \sqrt{2}W_t|^2] = t^2 \mathbb{E}[|\nabla U_{\tau(t_0)}(X_0)|^2] + 2td$. Denoting for any $\tau$, $x_\tau^*$ a minimizer of $U_\tau$ it follows, since $|x_\tau^*|^2 \leq b_\tau / a_\tau$ by dissipativity

$$
\begin{aligned}
\mathbb{E}[|\nabla U_{\tau(t_0)}(X_0)|^2] = \mathbb{E}[|\nabla U_{\tau(t_0)}(X_0) - \nabla U_{\tau(t_0)}(x_{\tau(t_0)}^*)|^2] &\leq L_{\tau(t_0)}^2 \mathbb{E}[|X_0 - x_{\tau(t_0)}^*|^2] \\
&\leq L_{\tau(t_0)}^2 \mathbb{E}[|X_0|^2] + 2|x_{\tau(t_0)}^*|^2 \\
&\leq L_{\tau(t_0)}^2 \mathbb{E}[|X_0|^2] + 2\frac{b_{\tau(t_0)}}{a_{\tau(t_0)}}
\end{aligned}
\quad (101)
$$

and in the general case for $k \in \mathbb{N}$, $\mathbb{E}[|X_t - X_{t_k}|^2] \leq 2d(t - t_k) + (t - t_k)^2 (L_{\tau(t_k)}^2 \mathbb{E}[|X_{t_k}|^2] + 2b_{\tau(t_k)} / a_{\tau(t_k)})$. Since by Lemma B.4, $\sup_k \mathbb{E}[|X_{t_k}|^2] < \infty$, Using also Lemma B.8 it follows for $\varepsilon = 1$ that

$$
\frac{\mathrm{d}}{\mathrm{d}t} \mathrm{KL}(\hat{\mu}_t | \pi_{\tau(t_0)}) \leq -\frac{2}{C_{\mathrm{LSI}}(t_0)} \mathrm{KL}(\hat{\mu}_t | \pi_{\tau(t_0)}) + ct. \quad (102)
$$

By Lemma B.10 we find

$$
\begin{aligned}
\mathrm{KL}(\hat{\mu}_{t_1} | \pi_{\tau(t_0)}) &\leq \mathrm{KL}(\hat{\mu}_0 | \pi_{\tau(t_0)}) \exp\left( -\frac{2h_0}{C_{\mathrm{LSI}}(t_0)} \right) + c \int_{t_0}^{t_1} \exp\left( -\frac{2(t_1 - s)}{C_{\mathrm{LSI}}(t_0)} \right) s \mathrm{d}s \\
&\leq \mathrm{KL}(\hat{\mu}_0 | \pi_{\tau(t_0)}) \exp\left( -\frac{2h_0}{C_{\mathrm{LSI}}(t_0)} \right) + ch_0^2.
\end{aligned}
\quad (103)
$$

concluding the proof. $\qquad\square$

### B.2.2. PROOF OF LEMMA 4.6

*Proof.* By Lemma 4.5 we find

$$
\begin{aligned}
\mathrm{KL}(\hat{\mu}_{k+1}|\pi_{\tau(t_k)}) &\leq \mathrm{KL}(\hat{\mu}_{t_k}|\pi_{\tau(t_k)}) \exp\left(-\frac{2h_k}{C_{\mathrm{LSI}}(t_k)}\right) + ch_k^2 \\
&\leq \left(\mathrm{KL}(\hat{\mu}_{t_k}|\pi_{\tau(t_k)}) - \mathrm{KL}(\hat{\mu}_{t_k}|\pi_{\tau(t_{k-1})}) + \mathrm{KL}(\hat{\mu}_{t_k}|\pi_{\tau(t_{k-1})})\right) \exp\left(-\frac{h_k}{2C_{\mathrm{LSI}}(t_k)}\right) + ch_k^2 \\
&\leq \mathrm{KL}(\hat{\mu}_{t_k}|\pi_{\tau(t_{k-1})}) \exp\left(-\frac{2h_k}{C_{\mathrm{LSI}}(t_k)}\right) + \int \hat{q}_k \left|\log\frac{p_{\tau(t_{k-1})}}{p_{\tau(t_k)}}\right| \mathrm{d}x \exp\left(-\frac{2h_k}{C_{\mathrm{LSI}}(t_k)}\right) + ch_k^2.
\end{aligned}
\tag{104}
$$

Using Lemmas B.4 and B.9 we find that $\int \hat{q}_k(x) \log\frac{p_{\tau(t_{k-1})}(x)}{p_{\tau(t_k)}(x)}\mathrm{d}x \leq c|\tau(t_k) - \tau(t_{k-1})|$. Inserting into (104) yields

$$
\mathrm{KL}(\hat{\mu}_{k+1}|\pi_{\tau(t_k)}) \leq \mathrm{KL}(\hat{\mu}_{t_k}|\pi_{\tau(t_{k-1})}) \exp\left(-\frac{2h_k}{C_{\mathrm{LSI}}(t_k)}\right) + c|\tau(t_k) - \tau(t_{k-1})| \exp\left(-\frac{2h_k}{C_{\mathrm{LSI}}(t_k)}\right) + ch_k^2.
\tag{105}
$$

Solving the recursion, we find

$$
\begin{aligned}
\mathrm{KL}(\hat{\mu}_{k+1}|\pi_{\tau(t_k)}) \leq{}& \mathrm{KL}(\hat{\mu}_1|\pi_{\tau(t_0)}) \exp\left(-\sum_{i=1}^{k}\frac{2h_i}{C_{\mathrm{LSI}}(t_i)}\right) + c\sum_{i=0}^{k-1}\exp\left(-\sum_{j=0}^{i}\frac{2h_{k-j}}{C_{\mathrm{LSI}}(t_{k-j})}\right)|\tau(t_{k-i}) - \tau(t_{k-1-i})| \\
&+ c\sum_{i=0}^{k-1}h_{k-i}^2 \exp\left(-\sum_{j=0}^{i-1}\frac{2h_{k-j}}{C_{\mathrm{LSI}}(t_{k-j})}\right).
\end{aligned}
\tag{106}
$$

The decomposition $\mathrm{KL}(\hat{\mu}_{k+1}|\pi) = \mathrm{KL}(\hat{\mu}_{k+1}|\pi_{\tau(t_k)}) + \mathrm{KL}(\hat{\mu}_{k+1}|\pi) - \mathrm{KL}(\hat{\mu}_{k+1}|\pi_{\tau(t_k)})$ then leads to

$$
\begin{aligned}
\mathrm{KL}(\hat{\mu}_{k+1}|\pi) \leq{}& \mathrm{KL}(\hat{\mu}_1|\pi_{\tau(t_0)}) \exp\left(-\sum_{i=1}^{k}\frac{2h_i}{C_{\mathrm{LSI}}(t_i)}\right) + c\sum_{i=0}^{k-1}\exp\left(-\sum_{j=0}^{i}\frac{2h_{k-j}}{C_{\mathrm{LSI}}(t_{k-j})}\right)|\tau(t_{k-i}) - \tau(t_{k-1-i})| \\
&+ c\sum_{i=0}^{k-1}h_{k-i}^2 \exp\left(-\sum_{j=0}^{i-1}\frac{2h_{k-j}}{C_{\mathrm{LSI}}(t_{k-j})}\right) + \int \hat{q}_{k+1} \log\frac{p}{p_{\tau(t_k)}}\mathrm{d}x \\
\leq{}& \mathrm{KL}(\hat{\mu}_1|\pi_{\tau(t_0)}) \exp\left(-\sum_{i=1}^{k}\frac{2h_i}{C_{\mathrm{LSI}}(t_i)}\right) + c\sum_{i=0}^{k-1}\exp\left(-\sum_{j=0}^{i}\frac{2h_{k-j}}{C_{\mathrm{LSI}}(t_{k-j})}\right)|\tau(t_{k-i}) - \tau(t_{k-1-i})| \\
&+ c\sum_{i=0}^{k-1}h_{k-i}^2 \exp\left(-\sum_{j=0}^{i-1}\frac{2h_{k-j}}{C_{\mathrm{LSI}}(t_{k-j})}\right) + c\tau(t_k)
\end{aligned}
\tag{107}
$$

where we used Lemma B.9 once more for the last inequality. $\qquad\square$

### B.2.3. PROOF OF THEOREM 4.7

*Proof.* By changing the summation indices, Lemma 4.6 leads to

$$
\begin{aligned}
\mathrm{KL}(\hat{\mu}_{k+1}|\pi) \leq{}& \mathrm{KL}(\hat{\mu}_1|\pi_{\tau(t_0)}) \exp\left(-\sum_{i=1}^{k}\frac{2h_i}{C_{\mathrm{LSI}}(t_i)}\right) + c\sum_{i=1}^{k}\exp\left(-\sum_{j=i}^{k}\frac{2h_j}{C_{\mathrm{LSI}}(t_j)}\right)|\tau(t_i) - \tau(t_{i-1})| \\
&+ c\sum_{i=1}^{k}h_i^2 \exp\left(-\sum_{j=i+1}^{k}\frac{2h_j}{C_{\mathrm{LSI}}(t_j)}\right) + c\tau(t_k)
\end{aligned}
\tag{108}
$$

It follows for any $1 \leq k' \leq k$

$$
\begin{aligned}
\mathrm{KL}(\hat{\mu}_{k+1}|\pi) \leq{}& \mathrm{KL}(\hat{\mu}_1|\pi_{\tau(t_0)}) \exp\left(-\sum_{i=1}^{k} \frac{2h_i}{C_{\mathrm{LSI}}(t_i)}\right) + c\sum_{i=1}^{k'} \exp\left(-\sum_{j=i}^{k} \frac{2h_j}{C_{\mathrm{LSI}}(t_j)}\right)|\tau(t_i)-\tau(t_{i-1})| \\
&+ c\sum_{i=k'+1}^{k} \exp\left(-\sum_{j=i}^{k} \frac{2h_j}{C_{\mathrm{LSI}}(t_j)}\right)|\tau(t_i)-\tau(t_{i-1})| \\
&+ c\sum_{i=1}^{k'} h_i^2 \exp\left(-\sum_{j=i+1}^{k} \frac{2h_j}{C_{\mathrm{LSI}}(t_j)}\right) + c\sum_{i=k'+1}^{k} h_i^2 \exp\left(-\sum_{j=i+1}^{k} \frac{2h_j}{C_{\mathrm{LSI}}(t_j)}\right) + c\tau(t_k) \\
\leq{}& \mathrm{KL}(\hat{\mu}_1|\pi_{\tau(t_0)}) \exp\left(-\sum_{i=1}^{k} \frac{2h_i}{C_{\mathrm{LSI}}(t_i)}\right) + c\exp\left(-\sum_{j=k'}^{k} \frac{2h_j}{C_{\mathrm{LSI}}(t_j)}\right) \\
&+ c|\tau(t_k)-\tau(t_{k'})| \\
&+ c\exp\left(-\sum_{j=k'+1}^{k} \frac{2h_j}{C_{\mathrm{LSI}}(t_j)}\right)\sum_{i=1}^{k'} h_i^2 + c\sum_{i=k'+1}^{k} h_i^2 + c\tau(t_k)
\end{aligned}
\tag{109}
$$

By assumption, $h_k$ is square summable but not summable. Let $\delta > 0$ be arbitrary. Choose first $k'$ sufficiently large so that $\tau(k') < \delta$ and $\sum_{i=k'+1}^{\infty} h_i^2 < \delta$. Then choose $k > k'$ such that $\exp\left(-\sum_{j=k'}^{k} \frac{2h_j}{C_{\mathrm{LSI}}(t_j)}\right) < \delta$. I follows that for such $k$, $\mathrm{KL}(\hat{\mu}_{k+1}|\pi) < c\delta$ for some $c$ concluding the proof.

To obtain the worst case complexity bound for constant step sizes, similarly as above, but inserting $h_i = h$ for all $i$ we find for any $1 \leq k' \leq k$

$$
\begin{aligned}
\mathrm{KL}(\hat{\mu}_{k+1}|\pi) \leq{}& \mathrm{KL}(\hat{\mu}_1|\pi_{\tau(t_0)}) \exp\left(-\sum_{i=1}^{k} \frac{2kh}{C_{\mathrm{LSI}}}\right) + c\sum_{i=1}^{k} \exp\left(-\frac{2(k-i+1)h}{C_{\mathrm{LSI}}}\right)|\tau(t_i)-\tau(t_{i-1})| \\
&+ c\sum_{i=1}^{k} h^2 \exp\left(-\frac{2(k-i)h}{C_{\mathrm{LSI}}}\right) + c\tau(t_k) \\
\leq{}& \mathrm{KL}(\hat{\mu}_1|\pi_{\tau(t_0)}) \exp\left(-\sum_{i=1}^{k} \frac{2kh}{C_{\mathrm{LSI}}}\right) + c\sum_{i=1}^{k'} \exp\left(-\frac{2(k-i+1)h}{C_{\mathrm{LSI}}}\right)|\tau(t_i)-\tau(t_{i-1})| \\
&+ c\sum_{i=k'+1}^{k} \exp\left(-\frac{2(k-i+1)h}{C_{\mathrm{LSI}}}\right)|\tau(t_i)-\tau(t_{i-1})| + c\sum_{i=1}^{k} h^2 \exp\left(-\frac{2(k-i)h}{C_{\mathrm{LSI}}}\right) + c\tau(t_k)
\end{aligned}
\tag{110}
$$

Solving the geometric sums then leads to

$$
\begin{aligned}
\mathrm{KL}(\hat{\mu}_{k+1}|\pi) \leq{}& \mathrm{KL}(\hat{\mu}_1|\pi_{\tau(t_0)}) \exp\left(-\frac{2kh}{C_{\mathrm{LSI}}}\right) + c\exp\left(-\frac{2(k-k'+1)h}{C_{\mathrm{LSI}}}\right)|\tau(t_{k'})-\tau(t_0)| \\
&+ c|\tau(t_k)-\tau(t_{k'})| \\
&+ ch^2 \frac{\exp\left(\frac{2h}{C_{\mathrm{LSI}}}\right) - \exp\left(-\frac{2kh}{C_{\mathrm{LSI}}}\right)}{\exp\left(\frac{2h}{C_{\mathrm{LSI}}}\right) - 1} + c\tau(t_k) \\
\leq{}& c\exp\left(-\frac{2(k-k')h}{C_{\mathrm{LSI}}}\right) + c\tau(t_{k'}) + \frac{ch^2}{\exp\left(\frac{2h}{C_{\mathrm{LSI}}}\right) - 1}
\end{aligned}
\tag{111}
$$

Noting that $\frac{h}{\exp(\frac{2h}{C_{\mathrm{LSI}}})-1}$ is bounded for $h \to 0$ we may conclude that, in order to obtain $\mathrm{KL}(\hat{\mu}_{k+1}|\pi)$ we have to proceed as follows (ignoring constants): First pick $h = \mathcal{O}(\varepsilon)$. Then pick $T_1 > 0$ sufficiently large such that $\tau(T_1) = \mathcal{O}(\varepsilon)$ and define $k' = \lfloor T_1/\varepsilon \rfloor$. Finally, choose $k = \mathcal{O}(T_1\varepsilon^{-1} + \log(\varepsilon^{-1}h^{-1}))$. $\qquad\square$

B.2.4. HELPER RESULTS

**Lemma B.4.** *Assume the step sizes $h_k$ satisfy $h_k < a_{\tau(t_k)}/L^2_{\tau(t_k)}$ and*

$$\limsup_{k \to \infty} \sum_{i=0}^{k-1} h_i \prod_{j=i+1}^{k-1} \left( 1 - 2h_j a_{\tau(t_j)} + 2L^2_{\tau(t_j)} h_j^2 \right) < \infty. \tag{112}$$

*Then, if $\int |x|^2 \hat{q}_0(x) \mathrm{d}x < \infty$, the second moments of the discrete scheme* (18) *are bounded. That is,*

$$\sup_{k \in \mathbb{N}} \int |x|^2 \hat{q}_k(x) \mathrm{d}x < \infty. \tag{113}$$

*Proof.* The proof is similar to (Habring et al., 2026; Fruehwirth & Habring, 2024). Let us denote $x^*_\tau$ for the minimizer of $U_\tau$. Now let $(h_k)_k$ be any sequence of step sizes such that $0 \le h_k < \frac{a_{\tau(t_k)}}{L^2_{\tau(t_k)}}$ for all $k$. We find

$$
\begin{aligned}
\mathbb{E}\left[ |X_{k+1}|^2 \right] &= \mathbb{E}\left[ |X_k - h_k \nabla U_{\tau(t_k)}(X_k) + \sqrt{2h_k} Z_k|^2 \right] \\
&= \mathbb{E}\left[ |X_k|^2 - 2h_k X_k \cdot \nabla U_{\tau(t_k)}(X_k) + h_k^2 |\nabla U_{\tau(t_k)}(X_k)|^2 \right] + 2h_k \\
&\le \mathbb{E}\left[ |X_k|^2 - 2h_k a_{\tau(t_k)} |X_k|^2 + h_k^2 (L_{\tau(t_k)} |X_k - x^*_{\tau(t_k)}|)^2 \right] + 2(1 + b_{\tau(t_k)}) h_k \\
&\le \mathbb{E}\left[ |X_k|^2 \right] \left( 1 - 2h_k a_{\tau(t_k)} + 2L^2_{\tau(t_k)} h_k^2 \right) + 2(1 + b_{\tau(t_k)}) h_k + 2h_k^2 L^2_{\tau(t_k)} |x^*_{\tau(t_k)}|^2.
\end{aligned}
\tag{114}
$$

Using also that by dissipativity $|x^*_{\tau(t_k)}|^2 \le b/a$ we find for any $h_k < a_{\tau(t_k)}/L^2_{\tau(t_k)}$

$$
\begin{aligned}
\mathbb{E}\left[ |X_k|^2 \right] &\le \prod_{i=0}^{k-1} \left( 1 - 2h_i a_{\tau(t_i)} + 2L^2_{\tau(t_i)} h_i^2 \right) \mathbb{E}\left[ |X_0|^2 \right] + 2(1 + 2b) \sum_{i=0}^{k-1} h_i \prod_{j=i+1}^{k-1} \left( 1 - 2h_j a_{\tau(t_j)} + 2L^2_{\tau(t_j)} h_j^2 \right) \\
&\le \mathbb{E}\left[ |X_0|^2 \right] + 2(1 + 2b) \sum_{i=0}^{k-1} h_i \prod_{j=i+1}^{k-1} \left( 1 - 2h_j a_{\tau(t_j)} + 2L^2_{\tau(t_j)} h_j^2 \right)
\end{aligned}
\tag{115}
$$

concluding the proof. $\qquad \square$

Since the step size condition (112) might seem pathological, let us investigate it in a little more detail. First, if $h_k = h \in \mathbb{R}$ is constant for all $k$, we find

$$
\begin{aligned}
\sum_{i=0}^{k-1} h_i \prod_{j=i+1}^{k-1} \left( 1 - 2h_j a_{\tau(t_j)} + 2L^2_{\tau(t_j)} h_j^2 \right) &\le \sum_{i=0}^{k-1} h \left( 1 - 2ha + 2L^2 h^2 \right)^{k-i-1} \\
&\le h \frac{1}{2h(a - L^2 h)} = \frac{1}{2(a - L^2 h)} < \infty.
\end{aligned}
\tag{116}
$$

In this case, we even find that the second moments are bounded uniformly with respect to $h$. Note, however, that it is not advised to choose a constant step size. In particular, for many popular choices of the family $(p_\tau)_\tau$, for large $\tau$ the density is better behaved allowing for larger step sizes initially, leading to faster convergence. On the contrary, when the step size is chosen constant, it has to satisfy $h < a/L^2$ with the worst case constants $a$, $L$ (*cf.* Assumption 3.2). However, an analogous result holds if the step size is lower-bounded, that is, there exists $h > 0$ such that $h_k \ge h$ for all $k$. Moreover, we can derive the following general result.

**Lemma B.5.** *Let $(h_k)_k$ be bounded and satisfy $h_k < a_{\tau(t_k)}/L^2_{\tau(t_k)}$. Then it holds*

$$\limsup_{k \to \infty} \sum_{i=0}^{k-1} h_i \prod_{j=i+1}^{k-1} \left( 1 - 2h_j a_{\tau(t_j)} + 2L^2_{\tau(t_j)} h_j^2 \right) < \infty. \tag{117}$$

*Proof.* Using that for any $x > 0$, $\log(x) \leq x - 1$ we find

$$
\sum_{i=0}^{k-1} h_i \prod_{j=i+1}^{k-1} \left(1 - 2h_j a_{\tau(t_j)} + 2L_{\tau(t_j)}^2 h_j^2\right) = \sum_{i=0}^{k} h_i \exp\left(\sum_{j=i+1}^{k} \log(1 - 2h_j a_{\tau(t_j)} + 2L_{\tau(t_j)}^2 h_j^2)\right)
$$

$$
\leq \sum_{i=0}^{k} h_i \exp\left(-\sum_{j=i+1}^{k} 2h_j a_{\tau(t_j)} - 2L_{\tau(t_j)}^2 h_j^2\right) \tag{118}
$$

$$
=: S_k.
$$

It holds true that $S_{k+1} = \exp\left(-2h_{k+1} a_{\tau(t_j)} + 2L_{\tau(t_j)}^2 h_{k+1}^2\right) S_k + h_{k+1}$. Let us denote

$$
\varphi_k(s) = \exp\left(-2h_k a_{\tau(t_j)} + 2L_{\tau(t_j)}^2 h_k^2\right) s + h_k \tag{119}
$$

so that $S_{k+1} = \varphi_k(S_k)$. Each function $\varphi_k$ is a contraction with unique fixed point

$$
\hat{s}_k = \frac{h_k}{1 - \exp\left(-2h_k a_{\tau(t_j)} + 2L_{\tau(t_j)}^2 h_k^2\right)} \leq \frac{h_k}{1 - \exp\left(-2h_k a + 2L^2 h_k^2\right)}. \tag{120}
$$

It holds for any $k$ that if $s > \hat{s}_k$, then $\varphi_k(\hat{s}_k) < s$ and if $s \leq \hat{s}_k$, then $\varphi_k(s) \leq \hat{s}_k$. Therefore, we have $\varphi_k(s) \leq \max\{\hat{s}_k, s\}$. Since the right-hand side of (120) is bounded for $h_k$ bounded (note that it converges to $1/(2a)$ as $h_k \to 0$), it follows that $\sup_k \hat{s}_k \leq M < \infty$ so that $\varphi_k(s) \leq \max\{M, s\}$ implying that $S_k \leq \max\{S_1, M\}$ and concluding the proof. $\square$

## B.3. Properties of the Convolutional Path

### B.3.1. PROOF OF LEMMA 4.14

*Proof.* It is well-known that for $\alpha, \sigma : [0, \infty) \to \mathbb{R}$ the Ornstein-Uhlenbeck process

$$
\mathrm{d}X_t = \alpha_t X_t \mathrm{d}t + \sigma_t \mathrm{d}W_t \tag{121}
$$

admits the solution $X_t = \exp\left(\int_0^t \alpha_s \mathrm{d}s\right) X_0 + \int_0^t \sigma_\tau \exp\left(\int_\tau^t \alpha_s \mathrm{d}s\right) \mathrm{d}W_\tau$. Indeed, for the solution of (121) by Itô's lemma it follows

$$
\mathrm{d}\left(X_t \exp\left(-\int_0^t \alpha_s \mathrm{d}s\right)\right) = \left\{-\alpha_t \exp\left(-\int_0^t \alpha_s \mathrm{d}s\right) X_t + \alpha_t \exp\left(-\int_0^t \alpha_s \mathrm{d}s\right) X_t\right\} \mathrm{d}t + \exp\left(-\int_0^t \alpha_s \mathrm{d}s\right) \sigma_t \mathrm{d}W_t
$$

$$
= \exp\left(-\int_0^t \alpha_s \mathrm{d}s\right) \sigma_t \mathrm{d}W_t \tag{122}
$$

which, by integrating over $t$ yields the desired result. Moreover, since $\int_0^t \sigma_\tau \exp\left(\int_\tau^t \alpha_s \mathrm{d}s\right) \mathrm{d}W_\tau$ is a Gaussian with zero mean and variance $\int_0^t \sigma_\tau^2 \exp\left(2 \int_\tau^t \alpha_s \mathrm{d}s\right) \mathrm{d}\tau$ it follows that

$$
X_t | X_0 \sim \mathcal{N}\left(\exp\left(\int_0^t \alpha_s \mathrm{d}s\right) X_0, \int_0^t \sigma_\tau^2 \exp\left(2 \int_\tau^t \alpha_s \mathrm{d}s\right) \mathrm{d}\tau\right). \tag{123}
$$

By choosing $\alpha_t = -\frac{1}{2(1-t)}, \sigma_t^2 = \frac{1}{1-t}$ we recover (28). $\square$

### B.3.2. PROOF OF LEMMA 4.15

*Proof.* By Lemma 4.14 $p_\tau$ satisfies the Fokker–Planck equation

$$
\partial_\tau p_\tau(x) = \mathrm{div}\left(\frac{x}{2(1-\tau)} p_\tau(x)\right) + \frac{1}{2(1-\tau)} \Delta p_\tau(x). \tag{124}
$$

Note also, that in this case it is trivial to see that the PDE is satisfied in the classical sense as $p_\tau$ can be written as a convolution between the initial distribution and a Gaussian which is sufficiently smooth. Since, moreover, it always holds

true that $\Delta \log p = \Delta p / p - |\nabla \log p|^2$ we find

$$
\begin{aligned}
\partial_\tau \log p_\tau(x) = \frac{\partial_\tau p_\tau}{p_\tau} &= \frac{\operatorname{div}\left(\frac{x}{2(1-\tau)} p_\tau(x)\right) + \frac{1}{2(1-\tau)} \Delta p_\tau(x)}{p_\tau(x)} \\
&= \frac{\operatorname{div}\left(\frac{x}{2(1-\tau)} p_\tau(x)\right) + \frac{1}{2(1-\tau)}(p_\tau(x)\Delta \log p_\tau(x) + p_\tau(x)|\nabla \log p_\tau(x)|^2)}{p_\tau(x)} \\
&= \frac{d}{2(1-\tau)} + \frac{x \cdot \nabla \log p_\tau(x)}{2(1-\tau)} + \frac{1}{2(1-\tau)}(\Delta \log p_\tau(x) + |\nabla \log p_\tau(x)|^2).
\end{aligned}
\tag{125}
$$

As shown in (Cordero-Encinar et al., 2025, Lemma 3.2), the Eigenvalues of $\nabla^2 \log p_\tau(x)$ are bounded uniformly in $\tau$ and $x$ and, thus, the result follows. $\qquad\square$

### B.4. About the Log-Sobolev Inequality

#### B.4.1. EXPLICIT DERIVATION OF $C_{\mathrm{LSI}}(\tau)$

**Lemma B.6.** *Let $\nu(\mathrm{d}x) = \exp(-U(x))\mathrm{d}x$ with $U$ twice continuously differentiable, $L$-smooth and dissipative with parameters $a, b, R$. Then $\nu$ satisfies LSI with constant $C_{\mathrm{LSI}}$ which can explicitly be estimated from $a, b, R, L$ and the dimension $d$ according to the proof below.*

*Proof.* As shown in (Raginsky et al., 2017, Propositions 13, 15) (*cf.*also (Cattiaux et al., 2010; Bakry et al., 2008)), if $U$ is twice continuously differentiable and $L$-smooth and, in addition, there exist $C^2$ functions $V, W : \mathbb{R}^d \to [1, \infty)$ such that with the operator $\mathcal{L}f = \Delta f - \langle \nabla f, \nabla U \rangle$, $U$ in addition satisfies the Lyapunov drift conditions[9]

$$
\begin{cases}
\mathcal{L}V(x) &\leq (-\lambda_0 + \kappa_0 \mathbb{1}_{\overline{B}_\rho(0)}(x))V(x), \quad \text{and} \\
\mathcal{L}W(x) &\leq (\kappa - \gamma|x|^2)W(x)
\end{cases}
\tag{126}
$$

for some $\lambda_0, \kappa_0, \rho, \kappa, \gamma > 0$, then $\nu$ satisfies LSI with $C_{\mathrm{LSI}} = C_1 + (C_2 + 2)C_P$ where the constants are computed as

$$
C_1 = \frac{2}{\gamma}\left(\frac{1}{\varepsilon} + \frac{L}{2}\right) + \varepsilon, \quad C_2 = \frac{2}{\gamma}\left(\frac{1}{\varepsilon} + \frac{L}{2}\right)\left(\kappa + \gamma \int |x|^2 \mathrm{d}\nu(x)\right), \quad C_p = \frac{1}{\lambda_0}\left(1 + 1C\kappa_0\rho^2 \mathrm{Osc}_\rho(U)\right)
\tag{127}
$$

where $\varepsilon > 0$ can be chosen arbitrarily, $C$ is a universal constant and $\mathrm{Osc}_\rho(U) = \max_{|x|\leq\rho} U(x) - \min_{|x|\leq\rho} U(x)$. We will apply this result and bound all appearing constants in terms of $L, a, b, R$: First of all, we choose $V(x) = 1 + |x|^2/2$ which yields

$$
\begin{aligned}
\mathcal{L}V(x) = d - \langle \nabla U(x), x \rangle &\leq d - (a|x|^2 - b\mathbb{1}_{B_R(0)}) \leq (1 + |x|^2)\left(-a + \frac{a + d + b\mathbb{1}_{B_R(0)}}{(1 + |x|^2)}\right) \\
&\leq V(x)\left(-\frac{a}{2} + \frac{a + d + b\mathbb{1}_{B_R(0)}}{(1 + |x|^2)} - \frac{a}{2}\right)
\end{aligned}
\tag{128}
$$

which implies

$$
\mathcal{L}V(x) \leq (-\lambda_0 + \kappa_0 \mathbb{1}_{\overline{B}_\rho(0)}(x))V(x)
\tag{129}
$$

with $\lambda_0 = a/2$, $\rho^2 = \frac{2(a+b+d)}{a} - 1$, and $\kappa_0 = a/2 + b + d$. For the second drift condition we may choose $W(x) = \exp(-\alpha|x|^2)$. Then one can check that $\nabla W(x) = 2\alpha x \exp(-\alpha|x|^2)$ and $\Delta W(x) = (2\alpha d + 4\alpha^2|x|^2)\exp(-\alpha|x|^2)$ implying

$$
\begin{aligned}
\mathcal{L}W(x) &= W(x)\left(2\alpha d + 4\alpha^2|x|^2 - 2\alpha\langle x, \nabla U(x)\rangle\right) \\
&\leq W(x)\left(2\alpha d + 4\alpha^2|x|^2 - 2\alpha(a|x|^2 - b\mathbb{1}_{B_R(0)})\right) \\
&\leq W(x)\left(-2\alpha(a - 2\alpha)|x|^2 + 2\alpha d + b\mathbb{1}_{B_R(0)}\right).
\end{aligned}
\tag{130}
$$

---

[9]Note that, if $V$ is coercive, the second implies the first condition. However, choosing different functions for $V$ and $W$ might lead to different constants.

Choosing, *e.g.*, $\alpha = a/4$, we obtain the desired second drift condition with $\gamma = a^2/4$, $\kappa = da/4 + b$. What is left to proof is that $\mathrm{Osc}_\rho(U)$ and $\int |x|^2 \mathrm{d}\nu(x)$ are bounded. A bound of $\int |x|^2 \mathrm{d}\nu(x)$ in terms of $L, a, b, R$ is established in Lemma B.3. Regarding $\mathrm{Osc}_\rho(U)$, let $x^*$ be a global minimizer of $U$ which exists by continuity and coercivity. It follows by dissipativity

$$0 = \langle \nabla U(x^*), x^* \rangle \geq a|x^*|^2 - b\mathbb{1}_{B_R(0)}$$

and, thus, $|x^*|^2 \leq b/a$. By the fundamental theorem of calculus and the fact that $\nabla U(x^*) = 0$ we find for any $x$

$$
\begin{aligned}
U(x) - U(x^*) &= \int_0^1 \langle \nabla U(x^* + t(x - x^*)), x - x^* \rangle \mathrm{d}t \\
&= \int_0^1 \langle \nabla U(x^* + t(x - x^*)) - \nabla U(x^*), x - x^* \rangle \mathrm{d}t \\
&\leq \int_0^1 Lt|x - x^*|^2 \mathrm{d}t \\
&\leq \frac{L}{2}|x - x^*|^2
\end{aligned}
\tag{131}
$$

which implies

$$\mathrm{Osc}_\rho(U) \leq \frac{L}{2} \max_{\substack{|x| \leq \rho \\ |x^*|^2 \leq b/a}} |x - x^*|^2 = \frac{L}{2}\left|\rho + \sqrt{\frac{b}{a}}\right|^2.$$

$\square$

### B.4.2. LOG-SOBOLEV INEQUALITES FOR NON-SMOOTH TEST FUNCTIONS

**Lemma B.7.** *Assume $\nu(\mathrm{d}x) = q(x)\mathrm{d}x$ with $q \in L^1(\mathbb{R}^d)$ and bounded from above. Then LSI holds for every $f \in W^{2,1}_{\mathrm{loc}}(\mathbb{R}^d) \cap L^2(\mathbb{R}^d, \nu)$ where, in the case that $\nabla f \notin L^2(\mathbb{R}^d, \nu)$, we set $\int |\nabla f|^2 \mathrm{d}\nu = \infty$.*

*Proof.* Let for $R > 0$, $\chi_R \in C_c^\infty(\mathbb{R}^d)$ such that $\chi_R(x) = 1$ for $x \in B_R(0)$, $\mathrm{supp}(\chi_R) \subset B_{R+1}$ and $\nabla \chi_R$ bounded independently of $R$. Let $f \in W^{2,1}_{\mathrm{loc}}(\mathbb{R}^d)$ be arbitrary. We consider first LSI for $h := f\chi_R$ and apply dominated convergence afterward. Indeed, since $f \in W^{2,1}_{\mathrm{loc}}(\mathbb{R}^d)$, we have $h \in W^{2,1}(\mathbb{R}^d)$. Let $h_n \in C_c^\infty(\mathbb{R}^d)$ be an approximating sequence for $h$ in $W^{2,1}(\mathbb{R}^d)$. Moreover, by taking a subsequence we can ensure that $h_n$ also converges pointwise a.e. For each $h_n$ LSI holds. We want to pass to the limit first with respect to $n$, then with respect to $R$. Using Fatou's lemma and the fact that $h^2 \log(h^2) \geq \min_x x^2 \log(x^2) > -\infty$ and constants are integrable with respect to the probability measure $\nu$ we find

$$\int h^2 \log h^2 \mathrm{d}\nu = \int \lim_n h_n^2 \log h_n^2 \mathrm{d}\nu \leq \liminf_n \int h_n^2 \log h_n^2 \mathrm{d}\nu \tag{132}$$

for which we use the pointwise a.e. convergence of $h_n$. Since $h_n \to h$ in $L^2(\mathbb{R}^d)$ we have that $\lim_n \overline{h_n^2} = \overline{h^2}$. Indeed, as $\overline{h_n^2} = \|h_n\sqrt{q}\|_{L^2(\mathbb{R}^d)}^2$, it follows that $\overline{h_n^2} \to \overline{h^2}$ if $h_n\sqrt{q} \to h\sqrt{q}$ in $L^2(\mathbb{R}^d)$ which follows from boundedness of $q$ and $L^2(\mathbb{R}^d)$ convergence $h_n \to h$. Next, we distinguish two cases: First, if $\overline{h^2} = 0$, then $h = 0$ $\nu$-a.e. so that the entire left-hand side of (LSI) is zero and LSI is satisfied for $h$. Therefore, let us assume $\overline{h^2} \neq 0$. In this case with $L^2(\mathbb{R}^d)$ convergence of $h_n$ it follows that

$$
\begin{aligned}
&\left| \int h_n^2 \log \overline{h_n^2} \mathrm{d}\nu - \int h^2 \log \overline{h^2} \mathrm{d}\nu \right| \\
&\leq \int h_n^2 \left| \log \overline{h_n^2} - \log \overline{h^2} \right| q \mathrm{d}\lambda + \int \left| h_n^2 - h^2 \right| \left| \log \overline{h^2} \right| q \mathrm{d}\lambda \\
&\leq \|h_n\|_{L^2(\mathbb{R}^d)}^2 \|q\|_{L^\infty(\mathbb{R}^d)} \left| \log \overline{h_n^2} - \log \overline{h^2} \right| + \|q\|_{L^\infty(\mathbb{R}^d)} \left| \log \overline{h^2} \right| \int |h_n - h| |h_n + h| \mathrm{d}\lambda \\
&\leq \|h_n\|_{L^2(\mathbb{R}^d)}^2 \|q\|_{L^\infty(\mathbb{R}^d)} \left| \log \overline{h_n^2} - \log \overline{h^2} \right| + \|q\|_{L^\infty(\mathbb{R}^d)} \left| \log \overline{h^2} \right| \|h_n - h\|_{L^2(\mathbb{R}^d)} \|h_n + h\|_{L^2(\mathbb{R}^d)} \to 0
\end{aligned}
\tag{133}
$$

as $n \to \infty$. Combining (132) and (133) leads to

$$\int h^2 \log \frac{h^2}{\overline{h^2}} \mathrm{d}\nu \leq \liminf_n \int h_n^2 \log \frac{h_n^2}{\overline{h_n^2}} \mathrm{d}\nu. \tag{134}$$

Convergence of the right-hand side of LSI follows directly by the approximation in $W^{2,1}(\mathbb{R}^d)$ together with boundedness of $q$. In total we obtain

$$\int h^2 \log \frac{h^2}{\overline{h^2}} \mathrm{d}\nu \leq \liminf_n \int h_n^2 \log \frac{h_n^2}{\overline{h_n^2}} \mathrm{d}\nu \leq C_{\mathrm{LSI}} \liminf_n \int |\nabla h_n|^2 \mathrm{d}\nu = C_{\mathrm{LSI}} \int |\nabla h|^2 \mathrm{d}\nu. \tag{135}$$

Lastly, assume $\nabla f \in L^2(\mathbb{R}^d, \nu)$. Let us denote $h$ as $h_R$ to make the dependence on $R$ explicit. By the same argument as above we can assume without loss of generality $\overline{f^2}, \overline{h_R^2} > 0$. Again, by Fatou's lemma

$$\int f^2 \log \overline{f^2} \mathrm{d}\nu \leq \liminf_R \int h_R^2 \log \overline{h_R^2} \mathrm{d}\nu. \tag{136}$$

Moreover, note that by choosing $\chi_R$ appropriately, for every $x$, $h_R(x)$ is monotonically increasing with respect to $R$, implying that also the sequence $\overline{h_R^2}$ is increasing and so is $h_R^2 \log \overline{h_R^2}$. Assuming without loss of generality $R > R_{\min}$ for some $R_{\min} > 0$ we also note that $h_R^2 \log \overline{h_R^2} \geq h_{R_{\min}}^2 \log \overline{h_{R_{\min}}^2}$ the latter being an element of $L^1(\mathbb{R}^d, \nu)$. Then by monotone convergence we also have

$$\int f^2 \log \overline{f^2} \mathrm{d}\nu = \lim_R \int h_R^2 \log \overline{h_R^2} \mathrm{d}\nu. \tag{137}$$

Combining (136) and (137) we find

$$\begin{aligned}
\int f^2 \log \frac{f^2}{\overline{f^2}} \mathrm{d}\nu &\leq \liminf_R \int h_R^2 \log \frac{h_R^2}{\overline{h_R^2}} \mathrm{d}\nu \\
&\leq C_{\mathrm{LSI}} \liminf_R \int |\nabla h_R|^2 \mathrm{d}\nu \\
&\leq C_{\mathrm{LSI}} \liminf_R \int |\nabla f \chi_R + f \nabla \chi_R|^2 \mathrm{d}\nu \\
&\leq C_{\mathrm{LSI}} \int |\nabla f|^2 \mathrm{d}\nu
\end{aligned} \tag{138}$$

where the last limit follows from dominated convergence. $\qquad\square$

**Lemma B.8.** *Assume $\nu$ satisfies LSI, $\nu(\mathrm{d}x) = q(x)\mathrm{d}x$ with $q > 0$ and $q$ is continuously differentiable. Let $\mu \ll \nu$ and assume the density of $\mu$ with respect to the Lebesgue measure[10] denoted as $h$ is continuous, $h > 0$, and $h \in W^{2,1}(\mathbb{R}^d)$. Then it holds*

$$\mathrm{KL}(\mu|\nu) \leq \frac{C_{\mathrm{LSI}}}{4} \int \left| \nabla \log \frac{\mathrm{d}\mu}{\mathrm{d}\nu} \right|^2 \mathrm{d}\mu \tag{139}$$

*Proof.* Consider $f = \left(\frac{h}{q}\right)^{\frac{1}{2}} \in L^2(\mathbb{R}^d, \nu)$. By the assumptions on $q$ we have for any compact set $K \subset \mathbb{R}^d$

$$\int_K f^2 \mathrm{d}x = \int_K f^2 q q^{-1} \mathrm{d}x \leq \|q^{-1}\|_{L^\infty(K)} \|f\|_{L^2(\mathbb{R}^d, \nu)}^2 < \infty. \tag{140}$$

Moreover, $f$ admits a weak derivative, $\nabla f = \frac{1}{2}\left(\frac{q}{h}\right)^{1/2} \frac{q\nabla h - h\nabla q}{q^2}$. Indeed, by weak differentiability of $h$ and continuous differentiability and positivity of $q$ we obtain $\frac{h}{q} \in W^{2,1}_{\mathrm{loc}}(\mathbb{R}^d)$. Continuity and positivity of $\frac{h}{q}$ then imply that $f = \left(\frac{h}{q}\right)^{1/2} \in W^{2,1}_{\mathrm{loc}}(\mathbb{R}^d)$ according to Lemma B.11. Using Lemma B.7 we find

$$\int f^2 \log(f^2) \mathrm{d}\nu - \int f^2 \log \overline{f^2} \mathrm{d}\nu \leq C_{\mathrm{LSI}} \int |\nabla f|^2 \mathrm{d}\nu(x). \tag{141}$$

For the right-hand side of the above equation we can compute

$$\int |\nabla f|^2 \mathrm{d}\nu(x) = \frac{1}{4} \int \frac{q}{h} \left| \frac{q\nabla h - h\nabla q}{q^2} \right|^2 q \mathrm{d}x = \frac{1}{4} \int h \left| \frac{\nabla h}{h} - \frac{\nabla q}{q} \right|^2 \mathrm{d}x = \frac{1}{4} \int h \left| \nabla \log \frac{h}{q} \right|^2 \mathrm{d}x \tag{142}$$

---

[10]Since $\mu \ll \nu \ll$ Leb, $\mu$ admits a Lebesgue density.

where for the last equality we used again Lemma B.11. For the left-hand side in (141) we can compute

$$\int f^2 \log(f^2) \mathrm{d}\nu - \int f^2 \log \overline{f^2} \mathrm{d}\nu = \int \frac{h}{q} \log\left(\frac{h}{q}\right) q \mathrm{d}\lambda - \int \frac{h}{q} q \mathrm{d}\lambda \underbrace{\log \int \frac{h}{q} q \mathrm{d}\lambda}_{=\log(1)=0} = \mathrm{KL}(\mu|\nu) \tag{143}$$

concluding the proof. □

### B.5. Miscellaneous Results

**Lemma B.9.** *Under Assumption 3.2 we have for some constants $c_1, c_2 > 0$, (i) $|U_{\tau_1}(x) - U_{\tau_2}(x)| \leq c_1 |x|^2 |\tau_1 - \tau_2|$ and (ii) $|Z_{\tau_1} - Z_{\tau_2}| \leq c_2 |\tau_1 - \tau_2|$.*

*Proof.* (i) The first assertion follows from the simple computation for $0 < \tau_1 < \tau_2$

$$|U_{\tau_1}(x) - U_{\tau_2}(x)| = \left|\int_{\tau_1}^{\tau_2} \partial_s U_s(x) \mathrm{d}s\right| \leq \int_{\tau_1}^{\tau_2} c|x|^2 \mathrm{d}s = c|x|^2 |\tau_1 - \tau_2|. \tag{144}$$

(ii) Regarding the second assertion, using that for any $z \geq 0$, $|1 - \exp(-z)| = 1 - \exp(-z) \leq z = |z|$ and distinguishing between the cases $U_{\tau_1}(x) \geq U_{\tau_2}(x)$ and $U_{\tau_1}(x) < U_{\tau_2}(x)$

$$\begin{aligned}
|Z_{\tau_1} - Z_{\tau_2}| &= \left|\int \exp\bigl(-U_{\tau_1}(x)\bigr) - \exp\bigl(-U_{\tau_2}(x)\bigr) \mathrm{d}x\right| \\
&\leq \int \bigl(\mathbb{1}_{\{U_{\tau_1} \geq U_{\tau_2}\}}(x) - \mathbb{1}_{\{U_{\tau_1} < U_{\tau_2}\}}(x)\bigr)\left[\exp\bigl(-U_{\tau_2}(x)\bigr) - \exp\bigl(-U_{\tau_1}(x)\bigr)\right] \mathrm{d}x \\
&= \int \mathbb{1}_{\{U_{\tau_1} \geq U_{\tau_2}\}}(x)\left[1 - \exp\bigl(-(U_{\tau_1}(x) - U_{\tau_2}(x))\bigr)\right] \exp\bigl(-U_{\tau_2}(x)\bigr) \mathrm{d}x \\
&\quad + \int \mathbb{1}_{\{U_{\tau_1} < U_{\tau_2}\}}(x)\left[1 - \exp\bigl(-(U_{\tau_2}(x) - U_{\tau_1}(x))\bigr)\right] \exp\bigl(-U_{\tau_1}(x)\bigr) \mathrm{d}x \\
&\leq \int \mathbb{1}_{\{U_{\tau_1} \geq U_{\tau_2}\}}(x)|U_{\tau_2}(x) - U_{\tau_1}(x)| \exp\bigl(-U_{\tau_2}(x)\bigr) \mathrm{d}x \\
&\quad + \int \mathbb{1}_{\{U_{\tau_1} < U_{\tau_2}\}}(x)|U_{\tau_2}(x) - U_{\tau_1}(x)| \exp\bigl(-U_{\tau_1}(x)\bigr) \mathrm{d}x \\
&\leq c_1 |\tau_1 - \tau_2| \int |x|^2 (Z_{\tau_1} p_{\tau_1}(x) + Z_{\tau_2} p_{\tau_2}(x)) \mathrm{d}x \\
&\leq c_2 |\tau_1 - \tau_2|
\end{aligned} \tag{145}$$

where the last inequality follows from the uniform boundedness of the second moments of $(\pi_\tau)_\tau$, Lemma B.3 and the uniform boundedness of the partition function also with respect to $\tau$ due to Assumption 3.2. □

**Lemma B.10** (Grönwall's inequality (Evans, 2010, Section B.2)). *Let $0 \leq t_0 \leq t_1$, $u, \varphi, \psi : [t_0, t_1] \to \mathbb{R}$ such that $u$ is absolutely continuous and $\varphi, \psi \in L^1([t_0, t_1])$. Moreover, assume*

$$u'(t) \leq \varphi(t) u(t) + \psi(t), \quad t \in [t_0, t_1]. \tag{146}$$

*Then it holds for all $t \in [t_0, t_1]$,*

$$u(t) \leq \exp\left(\int_{t_0}^t \varphi(s) \mathrm{d}s\right) u(t_0) + \int_{t_0}^t \exp\left(\int_\tau^t \varphi(s) \mathrm{d}s\right) \psi(\tau) \mathrm{d}\tau. \tag{147}$$

*Proof.* By (146) we have

$$\frac{\mathrm{d}}{\mathrm{d}t}\left(\exp\left(-\int_{t_0}^t \varphi(s) \mathrm{d}s\right) u(t)\right) = \exp\left(-\int_{t_0}^t \varphi(s) \mathrm{d}s\right)(u'(t) - u(t)\varphi(t)) \leq \exp\left(-\int_{t_0}^t \varphi(s) \mathrm{d}s\right) \psi(t). \tag{148}$$

Thus,

$$\exp\left(-\int_{t_0}^{t}\varphi(s)\mathrm{d}s\right)u(t) - u(t_0) \leq \int_{t_0}^{t}\exp\left(-\int_{t_0}^{\tau}\varphi(s)\mathrm{d}s\right)\psi(\tau)\mathrm{d}\tau \tag{149}$$

and, consequently,

$$u(t) \leq \exp\left(\int_{t_0}^{t}\varphi(s)\mathrm{d}s\right)u(t_0) + \int_{t_0}^{t}\exp\left(\int_{\tau}^{t}\varphi(s)\mathrm{d}s\right)\psi(\tau)\mathrm{d}\tau \tag{150}$$

$\square$

**Lemma B.11.** *Let $u \in W^{p,1}_{\mathrm{loc}}(\mathbb{R}^d)$, $p \in [1,\infty)$ such that $u > 0$ and $u$ is continuous. If $f$ is continuously differentiable on $(0,\infty)$, then it holds in the weak sense $\partial_{x_i}f(u) = f'(u)\partial_{x_i}u$ and $f(u) \in W^{p,1}_{\mathrm{loc}}(\mathbb{R}^d)$. If in addition $f(u), f'(u)\partial_{x_i}u \in L^2(\mathbb{R}^d)$ then $f(u) \in W^{p,1}(\mathbb{R}^d)$.*

*Proof.* Let $(u_n)_n \subset C^\infty(\mathbb{R}^d)$, $u_n = u * \psi_n$ with $\psi$ a mollifier with $\mathrm{supp}(\psi) \subset B_1(0)$, $\psi_n = n^d\psi(x/n)$ so that $u_n \to u$ in $W^{p,1}(K)$ for any compact $K \subset \mathbb{R}^d$ and pointwise a.e. by continuity of $u$. Let $\varphi \in C^\infty_c(\mathbb{R}^d)$ arbitrary. We have by integration by parts for all $n$

$$\int \partial_{x_i}\varphi f(u_n)\mathrm{d}x = -\int \varphi f'(u_n)\partial_{x_i}u_n\mathrm{d}x. \tag{151}$$

Let $K = \mathrm{supp}(\varphi)$. By definition of $u_n$ and continuity of $u$ we have that $u_n(K) \subset u(K + B_{\frac{1}{n}}(0)) \subset u(K_{\frac{1}{n}}) \subset u(K_1)$ where we denote $K_{\frac{1}{n}} = \overline{K + B_{\frac{1}{n}}(0)}$. By continuity $u(K_1)$ is a compact set and a subset of $(0,\infty)$. We can find $0 < a < b < \infty$ such that $u(K_1) \subset [a,b]$. On the set $[a,b]$, $f$ is Lipschitz continuous and $f'$ is bounded. As a consequence of Lipschitz continuity $f(u_n) \to f(u)$ in $L^p$. We can conclude

$$\begin{aligned}
\int \partial_{x_i}\varphi f(u)\mathrm{d}x &= \int_K \partial_{x_i}\varphi f(u)\mathrm{d}x \\
&= \lim_{n\to\infty}\int_K \partial_{x_i}\varphi f(u_n)\mathrm{d}x \\
&= -\lim_{n\to\infty}\int_K \varphi f'(u_n)\partial_{x_i}u_n\mathrm{d}x \\
&= -\int_K \varphi f'(u)\partial_{x_i}u\mathrm{d}x \\
&= -\int \varphi f'(u)\partial_{x_i}u\mathrm{d}x
\end{aligned} \tag{152}$$

where the second limit follows from boundedness of $f'(u_n)$, $L^p$ convergence of $\partial_{x_i}u_n$ and pointwise a.e. convergence of $f'(u_n)$. This concludes the weak differentiability. The fact that $f(u) \in W^{p,1}_{\mathrm{loc}}(\mathbb{R}^d)$ then is easily seen since $f(u)$, $f'(u)\partial_{x_i}u \in L^p(U)$ for any bounded set $U \subset \mathbb{R}^d$. The rest of the theorem is trivial. $\square$

**Lemma B.12.** *For some fixed $v \in \mathbb{R}^d$, the PDE*

$$\begin{cases} \partial_t p = \mathrm{div}(v \cdot p) + \Delta p, & (x,t) \in \mathbb{R}^d \times (0,\infty) \\ p(x,0) = p_0(x), & x \in \mathbb{R}^d \end{cases} \tag{153}$$

*admits the unique solution $p(x,t) = (p_0 * G_t)(x + tv)$ with $G_t(x) = \frac{1}{(4\pi t)^{d/2}}\exp\left(-\frac{1}{4t}|x|^2\right)$.*

*Proof.* Consider the transformation $q(x,t) := p(x - tv, t)$. Indeed, we find

$$\begin{aligned}
\partial_t q(x,t) &= \partial_t p(x - tv, t) - \nabla p(x - tv, t) \cdot v \\
&= \mathrm{div}(v \cdot p)(x - tv, t) + \Delta p(x - tv, t) - \nabla p(x - tv, t) \cdot v \\
&= \Delta p(x - tv, t) = \Delta q(x,t).
\end{aligned} \tag{154}$$

For the latter the unique solution is well-known to be $q(x,t) = (p_0 * G_t)(x)$ with $G_t(x) = \frac{1}{(4\pi t)^{d/2}}\exp\left(-\frac{1}{4t}|x|^2\right)$. $\square$

