# OpenReview forum: "Forward-KL Convergence of Time-Inhomogeneous Langevin Diffusions"
_ICML.cc/2026/Conference — ICML 2026 regular_

### Official Review · Reviewer_evR1 · 2026-03-08

**Soundness:** 3
**Presentation:** 3
**Significance:** 3
**Originality:** 3
**Overall Recommendation:** 4
**Confidence:** 3

**Summary:**

The paper studies non-asymptotic convergence guarantees for time-inhomogeneous Langevin diffusions and their discretizations. In particular, non-asymptotic convergence bounds are obtained for the continuous-time time-inhomogeneous Langevin diffusions and the Euler-Maruyama discretization in the forward KL divergence. Numerical experiments are provided to illustrate the theory in both low- and high- dimensional settings.

**Compliance With Llm Reviewing Policy:**

Affirmed.

**Final Justification:**

The author(s) have responded in the rebuttal in a satisfactory way.

**Key Questions For Authors:**

(1) You mentioned in Section 1 that most existing guarantees control the backward-Kullback-Leibler divergence $\mathrm{KL}(\pi|\mu_{t})$ instead of the forward-Kullback-Leibler divergence $\mathrm{KL}(\mu_{t}|\pi)$. When you said this, did you mean the literature on time-inhomogeneous Langevin diffusions? Please provide some references. The reason I am asking is because for time-homogeneous Langevin diffusion literature, most of the KL divergence results I am aware of are about $\mathrm{KL}(\mu_{t}|\pi)$. For a related question, if indeed most existing guarantees control the backward-Kullback-Leibler divergence $\mathrm{KL}(\pi|\mu_{t})$, whereas your work can provide the control on forward-Kullback-Leibler divergence $\mathrm{KL}(\mu_{t}|\pi)$, please provide some discussions in the contributions part in the introduction to highlight the novelty of your techniques and briefly explain why you can handle the forward KL while the existing literature cannot.

(2) In equation (14), please specify the domain of the double integral.

(3) In equation (15), what is $d\lambda$ and what is the domain of the integral?

(4) Lemma 4.3. is exact. I am wondering if you can utilize Lemma 4.3. to obtain how fast $\mathrm{K}(\mu_{t}|\pi)$ converges to $0$ in Theorem 4.4.

(5) In Lemma 4.5., you let $h_{k}<\frac{a_{\tau(t_{k})}}{L^{2}_{\tau(t_{k})}}$. Note that $t_{k}=\sum_{i=0}^{k-1}h_{i}$. Therefore, this assumption restricts the choice of the sequence $h_{k}$ in some not that transparent way. Later, in Theorem 4.7, you further assume that $\sum_{k}h_{k}=\infty$ and $\sum_{k}h_{k}^{2}<\infty$. Please provide some justifications why you can find a sequence $h_{k}$ that can simultaneously satisfy all three assumptions.

(6) Lemma 4.6. is exact. I am wondering if you can utilize Lemma 4.6. to obtain how fast $\mathrm{K}(\hat{\mu}_{k}|\pi)$ converges to $0$ in Theorem 4.7.

(7) In the reference Guo et al., langevin monte carlo should be Langevin Monte Carlo.

**Limitations:**

There are some discussions on the future research direction, but not on the limitations of the work.

**Strengths And Weaknesses:**

*Strengths

(1) Time-inhomogeneous Langevin diffusions and their discretizations arise in many application settings in machine learning and generative AI,
and the topic is certainly worth studying.

(2) The paper provides non-convergence bounds for the continuous-time time-inhomogeneous Langevin diffusions and the Euler-Maruyama discretization in the forward KL divergence, whereas the paper claims that the existing literature only studies the backward KL divergence.

*Weaknesses

(1) When the paper makes the claim that most existing guarantees control the backward-Kullback-Leibler divergence $\mathrm{KL(\pi|\mu_{t})$ instead of the forward-Kullback-Leibler divergence $\mathrm{KL}(\mu_{t}|\pi)$, no references are provided. Actually, I know quite many papers study the forward-Kullback-Leibler divergence $\mathrm{KL}(\mu_{t}|\pi)$ in the context of time-homogeneous Langevin diffusions and the discretizations. If the author(s) refer to this point in the context of time-inhomogeneous Langevin diffusions,
references should be provided.

(2) It is not very clear to me what is the technical novelty analyzing the forward KL divergence in this paper, and the analysis of the forward KL divergence for time-homogeneous Langevin is extensive in the literature, and the authors should discuss and highlight their technical novelty.

(3) The main theorems, such as Theorem 4.4. and Theorem 4.7., are asymptotic in nature, and only the supporting lemmas are non-asymptotic. As a result, the paper does not provide any analysis on the speed of the convergence in continuous-time or discrete-time, and hence
in the case of discrete-time, we do not know the iteration complexity.

---

> ### Author Rebuttal · Authors · 2026-03-30
>
> ## Overall response
> We thank the referee for the feedback, in particular for affirming that time-inhomogeneous Langevin dynamics arise in many applications and are certainly worth studying.
> ## Corrections
> 1. We apologize for the ambiguous wording. We were referring to the time-inhomogeneous case. For the time-homogeneous setting, the forward KL is, indeed, the more popular metric in the literature. In the inhomogeneous case, however, more works focus on the backward KL since it can be estimated using Girsanov's theorem [9,10,11,12]. We clarified this in the paper and added the references.
> 2. We agree this might not have been clearly communicated. The novelty is not so much the use of the forward KL, but rather: (i) the analysis of the inhomogeneous dynamics for which such results have not been available yet, (ii) the generalization to arbitrary paths and (iii) the rigorous analysis using mollification ensuring the results without tacitly assuming more regularity.
> 3. We thank the referee for this comment. Indeed, the results were presented in an asymptotic way, but can be generalized to allow for complexity bounds as follows: We stress that these estimates are worst case estimates using $C_{LSI}(\tau)\leq \overline{C_{LSI}}$ for all $\tau$ as we do not assume a specific behavior of $C_{LSI}(\tau)$.
>
> Continuous-time:
> To obtain the worst case bound we use similar computations as in the non-asymptotic proof to obtain
> \begin{equation}
> \begin{aligned}
> \mathrm{KL}(\mu_t|\pi)
> & \leq c(\tau(t)+\tau(t_1))+\mathrm{KL}(\mu_0|\pi_{\tau(0)})\exp\biggl(-\frac{4t}{\overline{C_{LSI}}} \biggr)+c\max_s\|\dot\tau(s)\| \int_{0}^{t_1}\exp\biggl( -\frac{4(t-s)}{\overline{C_{LSI}}} \biggr)d s\newline
> & \leq c(\tau(t) + \tau(t_1))+\mathrm{KL}(\mu_0|\pi_{\tau(0)})\exp\biggl( -\frac{4t}{\overline{C_{LSI}}} \biggr)+c\biggl[\exp\biggl( -\frac{4(t-t_1)}{\overline{C_{LSI}}} \biggr) - \exp\biggl(-\frac{4t}{\overline{C_{LSI}}} \biggr) \biggr]\newline
> & \leq c(\tau(t)+\tau(t_1))+c\exp\biggl( -\frac{4(t-t_1)}{\overline{C_{LSI}}} \biggr).
> \end{aligned}
> \end{equation}
> In order to reach accuracy $\epsilon$, ignoring constants, we require $\tau(t_1)=\mathcal{O}(\epsilon)$ and $t-t_1 = \mathcal{O}(\log(\epsilon^{-1}))$.
>
> Discrete time:
> Using constant step sizes $h_i = h$ for all i we find for $1\leq k'\leq k$
> \begin{equation}
> \begin{aligned}
>  \mathrm{KL}(\hat{\mu}\_{k+1}|\pi)
> \leq &\mathrm{KL}(\hat \mu_{1}|\pi_{\tau(t_0)})\exp\biggl(-\sum_{i=1}^k\frac{2kh}{\overline{C_{LSI}}} \biggr)
> \+c\sum_{i=1}^{k'}\exp\biggl(-\frac{2(k-i+1)h}{\overline{C_{LSI}}}\biggr)|\tau(t_{i})-\tau(t_{i-1})|\newline
> &\+c\sum_{i=k'+1}^{k}\exp\biggl( -\frac{2(k-i+1)h}{\overline{C_{LSI}}}\biggr)|\tau(t_{i})-\tau(t_{i-1})|+c\sum_{i=1}^{k}h^2\exp\biggl( -\frac{2(k-i)h}{\overline{C_{LSI}}} \biggr)+c\tau(t_k)
> \end{aligned}
> \end{equation}
> Solving geometric sums yields
> \begin{equation}
> \begin{aligned}
> \mathrm{KL}(\hat{\mu}\_{k+1}|\pi)\leq&c\exp\biggl(-\frac{2(k-k')h}{\overline{C_{LSI}}}\biggr)+c\tau(t_{k'})+\frac{ch^2}{\exp\biggl( \frac{2h}{\overline{C_{LSI}}}\biggr)-1}
> \end{aligned}
> \end{equation}
> where the last term is $\mathcal{O}(h)$. An error $\mathcal{O}(\epsilon)$ is obtained as follows: Pick $h = \mathcal{O}(\epsilon)$, then $T_1>0$ large such that $\tau(T_1) = \mathcal{O}(\epsilon)$ and define $k' = \lfloor T_1/\epsilon \rfloor $. Finally, choose $k = \mathcal{O}(T_1\epsilon^{-1}+\log(\epsilon^{-1})h^{-1})$. The latter also confirms the exponential choice $\tau(t) = \exp(-t/T)$ in order to not worsen complexity.
> * We fixed the minor errors. $\lambda$ denotes the Lebesgue measure, but we replaced $d\lambda$ by $d x$.
> * Step size conditions in Lemma 4.5: Since by ass. 3.2 $L\_\tau$ and $a\_\tau$ are uniformly bounded, the allowed step sizes are uniformly bounded away from zero, so $h_{k}<a_{\tau(t_{k})}/L_{\tau(t_{k})}^2$ does not cause any issues and $\sum_{k}h_{k}=\infty$ is feasible. Square summability is ensured by choosing a $h_k$ which decays sufficiently fast. In this context, it is worth noting that the joint condition of $\sum h_k=\infty$ but $\sum h_k^2<\infty$ is very common in optimization. Lastly, since $t_k = \sum\_{i=1}^{k-1} h_i$ does depend on the step sizes only up to $k-1$ there is no implicit dependence. At iteration $k$ we know $t_k$ which allows to compute $a_{\tau(t_{k})}/L_{\tau(t_{k})}^2$ and thus $h_k$ before performing the next step.
> * We fixed the references and added additional limitations (see response to ref Ebvv)
>
> [9] Cattiaux, P., Cordero-Encinar, P., and Guillin, A. Diffusion annealed Langevin dynamics: a theoretical study.
>
> [10] Cordero-Encinar, P., Akyildiz, O. D., and Duncan, A. B. Non-asymptotic analysis of diffusion annealed Langevin Monte Carlo for generative modelling
>
> [11] Guo, W., Tao, M., and Chen, Y. Provable benefit of annealed Langevin Monte Carlo for non-log-concave sampling
>
> [12] Young, J. M., Cordero-Encinar, P., Reich, S., Duncan, A., and Akyildiz, O. D. Diffusion path samplers via sequential Monte Carlo.

---

> > ### Author Rebuttal · Reviewer_evR1 · 2026-04-05
> >
> > The author(s) did a good job in the rebuttal. Most significantly, the author(s) demonstrated that even though the results were presented in an asymptotic way, but can be generalized to allow for complexity bounds as provided in detail in the rebuttal. I think I will raise the score.

---

### Official Review · Reviewer_7AnN · 2026-03-08

**Soundness:** 4
**Presentation:** 3
**Significance:** 3
**Originality:** 4
**Overall Recommendation:** 5
**Confidence:** 4

**Summary:**

This paper studies time-inhomogeneous Langevin diffusions for sampling from a target distribution through a path of intermediate distributions $(\pi_\tau)_{\tau \in [0,1]}$, where the target corresponds to $\tau \to 0$. The main technical contribution is a general convergence analysis in the forward KL divergence setting for both the continuous-time diffusion and an Euler–Maruyama discretisation with vanishing step sizes. A key feature of the framework is that it treats several commonly used paths, e.g., geometric tempering, dilation,  and convolutional paths, within a single analysis. The paper discusses weak Fokker–Planck solutions, which broadens the mathematical formulation compared with more classical analyses that focus on smoother or more specialised settings.

**Compliance With Llm Reviewing Policy:**

Affirmed.

**Final Justification:**

I appreciate the authors' responses to my questions and for their further clarifications. I stand by my review of this paper and feel that the paper is sound and original, albeit perhaps not sufficiently significant in my view. However, on balance, I would recommend Acceptance.

**Key Questions For Authors:**

1. Can you provide a stronger practical takeaway from the theory?  At present, the framework explains qualitatively why certain paths may be preferable, but it is not clear how a practitioner should use the results to choose a path or a schedule. Can the authors articulate a more concrete prescription, either theoretically or empirically, for when DAZ, convolutional, dilation, or tempering paths should be preferred?

2. How much of the empirical advantage comes from the path itself versus the induced step-size regime?  In the experiments, some methods appear to benefit from better conditioning and therefore larger step sizes. Can the authors separate the better path geometry from a more favourable discretisation budget by comparing methods under matched step sizes, matched gradient evaluations, or matched compute budgets?

3. Can the framework be validated on a more substantive application?  The current experiments are primarily Gaussian mixture illustrations. Do the authors have results on a more realistic Bayesian posterior, inverse problem, or another practically relevant sampling task that would demonstrate the value of the framework beyond toy examples?

**Limitations:**

Yes

**Strengths And Weaknesses:**

Strengths

Soundness: To the best of my understanding, the paper is technically sound. The theoretical development is coherent, the assumptions are clearly organised, and the main convergence arguments are built on a natural entropy-dissipation identity that separates the contraction term from the error introduced by tracking a moving target. This is the right analytical structure for this problem when the target is $\pi_\tau$ and $\tau$ varies. The continuous-time and discrete-time results are clearly set out, and the treatment of the Euler–Maruyama scheme with diminishing step sizes is reasonable and well motivated. I quite like that the authors are careful about what is and is not proved and so they don't oversell explicit rates where they are not available.

Presentation:  The paper is well written with a clear structure. The motivation is understandable, the path-based perspective is clear, and the paper does a good job of positioning multiple annealing constructions within one framework. Theorems, lemmas, and examples are well-organised and the narrative from abstract assumptions to concrete path families is easy to follow. The paper is also quite honest about the scope of its contributions. Ultimately, this is primarily a theory paper, with experiments serving more as illustrations than as a comprehensive empirical validation.

Significance:  The paper addresses an interesting problem. Path-based Langevin sampling, annealing, and smoothing constructions are widely used in approximate sampling and Bayesian computation, particularly recently in the context of diffusion samples, and having a common forward-KL framework is useful. The paper may be valuable to researchers working on sampling theory, annealed Langevin dynamics, and related areas of probabilistic ML. In particular, the unification of multiple paths under a common framework is meaningful and could support future work on schedule design, path selection, or further non-asymptotic analysis.

Originality: I think the paper is quite original in its theoretical contributions. I do not view it as simply introducing a fundamentally new sampling methodology, but it does make a genuine theoretical contribution by casting several existing path constructions into a single framework and analysing them in forward KL for both continuous and discrete dynamics. That in itself is a worthwhile form of originality. The forward-KL emphasis and the handling of weak Fokker–Planck solutions also add technical value.

Weaknesses

Soundness: While the theory itself seems solid, the support for broader ML-facing claims is a bit limited. The empirical section is quite narrow. The experiments focus on Gaussian mixture examples, and the higher-dimensional evidence is therefore limited in scope (I can see that there a few more numerics in the appendix). As a result, the paper does not strongly demonstrate practical impact for modern ML settings such as more challenging Bayesian posteriors, inverse problems, or score-based / diffusion-style applications that are part of the paper’s broader motivation. Although the theoretical claims are sound, the empirical evidence is a bit lacking and therefore makes it difficult to establish broad practical usefulness.

Presentation:  The paper is mostly clear, but a few aspects could be sharpened. First, the practical implications of the theory are not drawn out enough. For a broad ML audience, the paper could do more to explain what the theorems imply operationally: when should one prefer DAZ versus tempering versus convolutional paths, and what aspects of the theory are meant to guide these choices? Second, the distinction between mathematical generality and practical coverage could be stated more plainly. The assumptions still keep the analysis largely in a smooth-gradient Langevin regime, even if some difficult targets can be approached through smoothing paths.

Significance:  This is where I have the biggest reservation for ICML. The paper is interesting and technically competent, but its significance for the broader ICML audience is possibly limited in its current form. It does not introduce a new algorithmic paradigm, nor does it demonstrate that the theory translates into compelling gains on substantive machine learning problems. While the paper is technically interesting, I worry that the likely impact could be quite specialised rather than broad. I can imagine this being useful to researchers in sampling theory and Bayesian computation, but it is less clear that it will substantially influence mainstream ICML practice without either stronger applications or more actionable theory.

Originality:  The paper is certainly original and I don't have much to add that would be viewed as a weakness. The paper’s novelty comes from analysis and unification rather than from new methodology, which is a bit of a weakness, but not a significant weakness. The contribution feels closer to a strong technical theory paper than to a paper that will substantially shift the field’s methodology or empirical practice.

---

> ### Author Rebuttal · Authors · 2026-03-30
>
> ## Overall response
> We highly appreciate the positive feedback by the referee, in particular regarding the relevance of investigating annealing schemes and the technical value of the provided theoretical work.
>
> ## Corrections
> * **Soundness:** While we understand that the experiments section is less emphasized, as the reviewer also states, the paper is primarily a theory paper, with experiments serving as illustrations rather than a comprehensive empirical validation. Nonetheless, the provided experiments cover 1, 2, and 10-dimensional settings. Moreover, despite their simplicity we believe that GMMs are a relevant target as they can be designed difficult for sampling (in particular, highly multimodal) while maintaining $L$-smoothness and dissipativity. Moreover, (i) annealing is a widespread technique for sampling [4,5,6,7,8] so that it there is enough evidence that it performs well also for complicated applications and (ii) (as correctly noted by the referee) we do not propose a new sampling method. The considered methods in the experiments have previously been proposed and some of them are established approaches. Therefore, the numerical experiments do not serve the purpose of establishing SOTA or similar. Their purpose is to reflect the theoretical results and support the derived path design guidelines, the latter mentioned guidelines being precisely the impact we hope to have, as mentioned by the referee and also referee Ebvv. Moreover, we added an additional experiment in which all methods share the same step size, to confirm that the improvements through annealing are not merely a consequence of larger step sizes.
> * **Presentation:** We agree that the key takeaways should be emphasized better: From the theory we may derive a guideline for annealing paths: It should be ensured that $L_\tau, b_\tau, R_\tau$ are small and $a_\tau$ is large for $\tau>0$. On the one hand, this leads to large step sizes as proven in Lemma 4.5. On the other hand, this yields small LSI constants (see also the newly added estimates of the LSI constants in response to ref 8AG4) which implies faster convergence by our results (also in the continuous-time setting). We added these suggestions in a remark after the results and in the contributions. Regarding the differences between different paths in general the results suggests that the preferred path should always be the one which leads to the best LSI constants. Based on the added estimates of the constants $L_\tau, b_\tau, R_\tau$, $a_\tau$ (see response to ref Ebvv) and the experiments, this will be the convolutional path in general, closely followed by DAZ. However, contrary to the convolutional path, DAZ is not necessarily pre-trained, so that DAZ may be preferred if prior training of the noise conditional score is not feasible. Regarding the "distinction between mathematical generality and practical coverage" we added a remark stating that currently we assume smoothness of the target but the approaches may be applied to smooth approximations.
> * **Significance:** We see the significance of the manuscript twofold: Firstly, we view the paper as primarily theoretical and find that providing a thorough theoretical underpinning of a widely used practice for sampling is research worthwhile pursuing. Secondly, practical relevance of the provided results can be derived from the above explained concrete guidelines for annealing path design, annealing schedules, and step size choices. As mentioned, we emphasize this, in particular, in the contributions in the updated manuscript.
> * **Originality:** We thank the referee for the positive feedback! We want to point out that as interpreted by the referee the aim of the paper was to provide a strong technical work formalizing common practices in the ML community rigorously.
> * Regarding stronger practical takeaways, see above.
> * Regarding the additional experiment comparing different paths with identical step size, see figure 4 in the updated manuscript at <https://anonymous.4open.science/r/time-inhom-Langevin-E3C2/Annealed_sampling_ICML.pdf>. Indeed, also with identical step sizes annealing improves convergence, albeit to a lesser degree. This is expected as the improved convergence speed enters also the continuous time results.
> * Regarding empirical evidence on more complicated examples we refer to [4,5,6,7,8], where annealing has been used for (medical) imaging. Once again we want to emphasize that we do not think it is necessary to prove that annealing is practically relevant as it has been used substantially in the literature and our focus was to provide a thorough theoretical underpinning.
>
> [8] Song, Yang, and Stefano Ermon. "Generative modeling by estimating gradients of the data distribution."

---

> > ### Author Rebuttal · Reviewer_7AnN · 2026-04-01
> >
> > Thank you to the authors for positively engaging with my comments. If the authors have the opportunity to expand the numerical results beyond GMMs, then I would encourage them to do this as I believe it would significantly strengthen the paper.
> >
> > I have no further questions and I will be maintaining my score.

---

> > > ### Author Response · Authors · 2026-04-02
> > >
> > > We sincerely thank the reviewer.
> > >
> > > > If the authors have the opportunity to expand the numerical results beyond GMMs, then I would encourage them to do this as I believe it would significantly strengthen the paper.
> > >
> > > We will perform an experiment for Bayesian inverse imaging (denoising, MRI or similar) with a noise conditional score network or an energy based model trained with contrastive divergence.

---

### Official Review · Reviewer_Ebvv · 2026-03-09

**Soundness:** 3
**Presentation:** 4
**Significance:** 3
**Originality:** 3
**Overall Recommendation:** 5
**Confidence:** 3

**Summary:**

The paper provides convergence guarantees for continuous and discrete-time Langevin dynamics in terms of forward KL divergence from the Gibbs distribution, with generalized time-inhomogenous potentials, assuming smoothness, dissipativity and quadratic growth of the time derivative. The provided Lemmas (4.3, 4.6) indicate guidelines for effective annealing scheme design.
Then, it is shown that several common annealing schemes for the potential satisfy the required assumptions, under varying assumptions on the non-annealed potential, proving their convergence, along with the posteriors in Bayesian inverse problems.
Lastly, the developed intuition on effective annealing scheme design is experimentally demonstrated through a problem of sampling from a Gaussian mixture.

**Compliance With Llm Reviewing Policy:**

Affirmed.

**Final Justification:**

As mentioned in the rebuttal acknowledgement I posted, the authors have addressed the issues I raised and reinforced my positive assessment of the paper. I hope they choose to incorporate the final suggestions included there.

**Key Questions For Authors:**

Throughout the paper the authors use constants and simply refer to them as $c$. However, these may be global constants, or hide some scaling with the problem parameters. Which is the case here?

**Limitations:**

Not sufficiently. The authors can better emphasize the significance of the work by addressing how their assumptions apply to practical deep learning scenarios.

**Strengths And Weaknesses:**

Generally speaking, I find the paper is very well written and fitting for acceptance as-is. However, I did not extensively assess the correctness of the proofs in the supplementary material, which is crucial for the paper’s contribution to the community. Furthermore, I lack the necessary background on annealing schemes in diffusion processes, hence the low confidence score.

**Strengths**

The paper rigorously proves convergence guarantees of practical annealing schemes under fairly mild assumptions, yielding seemingly significant contributions.
The unifying view and intuitions developed may pave the way to new annealing schemes.
The paper is very well organized. The story and reading flow are compelling, and even an uninformed reader can benefit from the paper due to the extensive preliminaries section.

**Weaknesses**

1. Lines 271-273 mention the LSI constant, which is crucial in the following analysis and for the developed intuition, as per Remark 4.8. While this is mentioned as a  direction for future work, I believe the paper could benefit from a deeper dive into how the LSI constant is affected by the properties of the potential and the Gibbs distribution, which will help to clarify how large it is for $\tau=0$, emphasizing the necessity of the annealing schemes. Perhaps adapting the analysis in Raginsky et al., 2017 [1] is more than enough for this.
2. Also, a more formal proposition stating the existence of LSI than the one provided in lines 271-273 may be helpful.
3. Section 4.3.5 may benefit from further motivation, perhaps by citing works where annealing schemes were utilized for Bayesian inverse problems.
4. The experiments are not clearly stated, and more details will help better understand them. For example:
  - What are the parameters being evolved, and how are they evolved by each scheme? This may be answered by providing the code, or some pseudo-code.
  - How exactly do step sizes compare between methods?

Note: as previously stated, it is hard to assess the paper’s soundness without a deep dive into the proofs in the supplementary material. In general I find that perhaps a venue such as COLT might have been a better fit for this paper, where it could have been more properly assessed.

**Small corrections**

1. In line 371, I believe you meant smaller instead of larger.
2. In Proposition 4.11, Lipschitz continuity of the gradient isn’t properly mentioned.

**References**

[1] Raginsky, Maxim, Alexander Rakhlin, and Matus Telgarsky. "Non-convex learning via stochastic gradient langevin dynamics: a nonasymptotic analysis." Conference on Learning Theory. PMLR, 2017.

---

> ### Author Rebuttal · Authors · 2026-03-30
>
> ## Overall response
> We thank the reviewer for the positive feedback and appreciate the mentioning that the results (Lemmas 4.3, 4.6) indicate guidelines for effective annealing schemes and may pave the way to new schemes, which was one of the driving motivations for us.
>
> Based on the review, we provide the code at <https://anonymous.4open.science/r/time-inhom-Langevin-E3C2>. Upon acceptance, we will publish the source code on our github.
>
> Since the referee mentions that they did not check the proofs in detail, we want to point out that the other referees confirmed the correctness of proofs (up to minor inconsistencies which we already corrected).
>
> ## Corrections
> * To address points 1 and 2, we added a lemma deriving bounds on the LSI constants to the appendix of the revised manuscript, which is referenced in the main part after assumption 3.2. Based on these estimates, we highlight that annealing families should be designed so that $L,b,R$ are small and $a$ is large, yielding improved LSI constants and, thus, faster convergence. We provide a proof (that makes use of the suggested reference) in the response to ref 8AG4.
> * To address point 3, we added multiple references in which annealing is used for posterior inference [4,5,6] to section 4.3.5.
> * Regarding point 4: We thank the referee for the suggestions for the experiments. As stated in the manuscript, the parameter to evolve is $\tau$ which we set as $\tau(t) = e^{-t/T}$ and we discretize $t$ via time steps $h_k$. The only remaining ambiguity lies in the estimation of $a_{\tau(t_k)}$ and $L_{\tau(t_k)}$,  for which we used the following approaches (added to the supplement):
>
> ULA: $p_{\tau}=p_0=\sum_{i=1}^{n_c}\alpha_i\mathcal{N}(m_i,\Sigma_i)$ for all $\tau$. For the GMM we estimate $L_\tau$ and $a_\tau$ via
>     $$
>     L_\tau=L_0=\max_i \frac{1}{\lambda_{\min}(\Sigma_i)^2},\quad a_\tau=a_0=\min_i \frac{1}{\lambda_{\max}(\Sigma_i)^2}
>     $$
> where $\lambda_{\mathrm{min}}$ and $\lambda_{\mathrm{min}}$ denote the smallest and largest eigenvalues.
>
> Dilation:
> As shown in the manuscript, we have $a_\tau = \frac{a_0}{1-\tau}$, $L_\tau = \frac{L_0}{1-\tau}$ so that the values for $\tau>0$ can be estimated using $a_0$ and $L_0$.
>
> Tempering:
> $U_\tau=(1-\tau)U_0+\tau U_1+c$, thus, $L_\tau=(1-\tau)L_0+\tau L_1$, $a_\tau(1-\tau)a_0+\tau a_1$. We set $U_1(x)=|x|^2/2$ so that the parameters $a_1=L_1=1$. The parameters for $\tau=0$ are again estimated via the above formula in the case of ULA.
>
> Convolution: $p_\tau$ is a GMM with component covariances $\Sigma_i(\tau)=(1-\tau)\Sigma_i+\tau$. Thus, for each $\tau$ we use
> $$
>     L_\tau=\max_i \frac{1}{(1-\tau)\lambda_{\min}(\Sigma_i)+\tau},
>     \quad a_\tau=\min_i \frac{1}{(1-\tau)\lambda_{\max}(\Sigma_i)+\tau}
> $$
> DAZ: Based on Remark 4.17 of [5] we use the estimates
> $$
>     L_\tau=\min\Bigl( \frac{1}{\tau},\max\Bigl(0, \frac{L_0}{1-L_0\tau}\Bigr)\Bigr)
> $$
> and based on the definition of the Moreau envelope $a_\tau = \min\bigl(a_0, \frac{1}{\tau}\bigr)$.
>
> To compare step sizes as suggested we added a plot of the time $t_k$ over the number of iterations for the different methods (<https://anonymous.4open.science/r/time-inhom-Langevin-E3C2/results/gmm_1d/compare_step_sizes.png>). We find that initially DAZ and the convolutional path admit larger step sizes. Dilation leads to smaller step sizes than ULA, as expected from the above formulas. Eventually, all methods admit the same step sizes when $\tau\approx 0$. See ref 7AnN for an experiment using the same step sizes across paths.
> * Small corrections were added.
> * We added the following limitations paragraph to the conclusion: A more concrete quantification and ideal choice of the schedule $\tau(t)$ with the current analysis would rely on estimates of the log-Sobolev constants which are in general hard to obtain.
> * Regarding the constants: The constants could actually be tracked which would allow to determine the dependence on the dimension. Similar dependence as in [7] with the dimension dependence typically entering through, e.g., moment bounds can be expected. We omitted the tracking of the constants to increase readability and since we focused on qualitative results and a careful treatment of the PDE arguments. However, we agree that this would indeed be worth pursuing
>
> [4] Xun, Z., Gupta, S., and Price, E. Posterior sampling by combining diffusion models with annealed Langevin dynamics.
>
> [5] Habring, A., Falk, A., Zach, M., and Pock, T. Diffusion at Absolute Zero: Langevin sampling using successive Moreau envelopes. SIAM Journal on Imaging Sciences.
>
> [6] Blumenthal, M., Holliber, T., Tamir, J. I., and Uecker, M. Fast and robust diffusion posterior sampling for MR image reconstruction using the preconditioned unadjusted Langevin algorithm.
>
> [7] Omar Chehab et al. Provable Convergence and Limitations of Geometric Tempering for Langevin Dynamics.

---

> > ### Author Rebuttal · Reviewer_Ebvv · 2026-04-01
> >
> > I appreciate the authors' thorough response, and they have addressed all of the issues I raised. I will highlight specific points I find most important:
> > 1. By explicitly addressing the magnitude of the LSI constant and how diffrent annealing schemes affect it, I believe the reader will benefit more from the paper, and the authors agreed to adding such clarifications.
> > 2. With the added page the final manuscript allows, I expect the presentation of the experiments in the main body to be clearer,  along with the added elaborations in the supplementary material the authors have stated they will add.
> > 3. I believe the possible dependence of seemingly "naive" $c$ constants on dimension quantities must be explicitly stated, even if it is not tracked - the reader should not be confused into thinking the constants are $\Theta(1)$.
> >
> > Regarding my score, my positive view of the paper remains. I am encouraged that reviewers 8AG4 (explicitly stated) and 7AnN (if I understand correctly) found the proofs in the supplementary material sound.
> > Since the authors addressed my main concerns, and given the additional signals from the discussion, I am comfortable increasing my confidence score to 3, while noting that I still did not independently check the supplementary proofs in detail.

---

### Official Review · Reviewer_8AG4 · 2026-03-13

**Soundness:** 3
**Presentation:** 3
**Significance:** 3
**Originality:** 3
**Overall Recommendation:** 4
**Confidence:** 4

**Summary:**

This paper provides a theoretical convergence analysis of annealed Langevin samplers, targeting time-varying distributions. The authors provide results in both continuous and discrete time, and the estimates are typically phrased in terms of the log-Sobolev constants of the moving targets, plus tracking errors.

**Compliance With Llm Reviewing Policy:**

Affirmed.

**Key Questions For Authors:**

Please see above under Strengths and Weaknesses, Soundness. Could you please address these technical questions?

**Limitations:**

yes

**Strengths And Weaknesses:**

This is a nice technically competent paper that provides a solid theoretical foundation for annealed Langevin samplers.

Soundness: I believe the main results and proofs are essentially correct. A few (small) concerns:

(i) In Lemma 4.2, equation (15), I believe it should be $\le$ rather than $=$; this seems to be a typo (similar in Lemma 4.3).

(ii) Lemma B.8, second statement: The stated Lipschitz continuity of the normalising constants seems unlikely to be generally true under the stated assumptions (I would expect something like uniformly bounded exponential moments or similar to be required). In the proof of the lemma, the problem might be with the inequality $|\exp(-U_{t_2}(x) - U_{t_1}(x)) - 1| \le |U_{t_2}(x) - U_{t_1}(x)|$.

(iii) In line 867, the authors write that uniform dissipativity implies that there is time-uniform upper bound on the log-Sobolev constants. I don't understand this, I believe the log-Sobolev constants could become arbitrarily bad even under uniform dissipativity (create a barrier in the center of the state space).

Significance: The results are significant in that they provide a solid foundation for annealed Langevin samplers and similar algorithms. At the same time, for example the statement of Theorem 4.4 is not really very surprising, and it is unclear how much impact it can have on algorithmic design.

Originality: The result seem to be original, although the proof techniques seem to be relatively standard. An important original contribution is the uniform treatment and the careful analysis of edge cases.

---

> ### Author Rebuttal · Authors · 2026-03-30
>
> ## Overall response
> We thank the referee for highlighting the technical competency and the theoretical aspects of the manuscript!
>
> We also appreciate that the reviewer mentions the significance for annealed Langevin samplers and the uniform treatment and careful analysis. Indeed, providing rigorous arguments was a main goal of this work.
>
> Regarding the statement that the impact of theorem 4.4 is unclear: In conjunction with the newly added estimates on the LSI constants (see under (iii)), we added a paragraph to the manuscript in which we highlight that this implies that annealing families should be designed such that $L,R,b$ are small and $a$ is large, leading to improved LSI constants and, thus, faster convergence. We believe that this could have an impact on future research.
>
> ## Corrections/smaller technical errors
>
> (i) In lemma 4.2 we do, actually, have equality. However, the reviewer is correct that lemma 4.3 should be an inequality. We corrected the manuscript. Thank you for pointing this out.
>
> (ii) We thank the reviewer for this attentive comment! Indeed, the proof was not correct as is. The inequality $|\exp(-(U_{\tau_2}(x) - U_{\tau_1}(x)) - 1| \le |U_{\tau_2}(x) - U_{\tau_1}(x)|$ is in general only correct if $U_{\tau_2}(x) - U_{\tau_1}(x)\geq 0$ as a consequence of the fact that for all $z\geq 0$, $|1-\exp{(-z)}| = 1-\exp{(-z)} \leq z = |z|$. The result, however, still holds true with the following adapted proof: Distinguishing between the cases $U_{\tau_1}(x)\geq U_{\tau_2}(x)$ and $U_{\tau_1}(x)< U_{\tau_2}(x)$ we have that
> \begin{equation}
>     \begin{aligned}
> |Z_{\tau_1}-Z_{\tau_2}|
>         =& \left|\int \exp\bigl( -U_{\tau_1}(x) \bigr) - \exp\bigl( -U_{\tau_2}(x) \bigr)d x\right| \newline
>         \leq& \int\_{\{U_{\tau_1}\geq U_{\tau_2}\}}\exp\bigl( -U_{\tau_2}(x) \bigr) - \exp\bigl( -U_{\tau_1}(x) \bigr) d x\newline
> &+\int\_{\{U_{\tau_1}< U_{\tau_2}\}}\exp\bigl( -U_{\tau_1}(x) \bigr) - \exp\bigl( -U_{\tau_2}(x) \bigr)d x,
> \end{aligned}
> \end{equation}
> where now both integrals can be treated as in the manuscript. $\def\R{{\mathbb{R}}}$
>
> (iii) We agree with the reviewer that this point should be highlighted more: Indeed, if $U=\log \nu$ is $C^2$, Lipschitz continuity of $\nabla U$ and dissipativity imply a bounded LSI constant for the distribution $\nu$ which can directly be derived from the Lipschitz modulus and the constants quantifying the dissipativity. In the revised manuscript, we added an additional lemma in the appendix where we derive these bounds and reference the lemma after the stated assumptions in the main paper. The proof is based on the following result (Propositions 13, 15 of [1], cf also [2,3]): If $U$ is twice continuously differentiable and $L$-smooth and, in addition, there exist $C^2$ functions $V,W:\R^d\rightarrow[1,\infty)$ such that with the operator $\mathcal{L} f = \Delta f - \langle\nabla f,\nabla U\rangle$, $U$ satisfies the Lyapunov drift conditions (Note that, if $V$ is coercive, the second implies the first condition. However, choosing different functions for $V$ and $W$ might lead to different constants.)
>     \begin{equation}
>         \begin{cases}
>             \mathcal{L} V(x)&\leq (-\lambda_0 + \kappa_0 1_{\overline B_\rho(0)}(x))V(x),\quad \text{and}\newline
>             \mathcal{L} W(x)&\leq (\kappa -\gamma|x|^2)W(x)
>         \end{cases}
>     \end{equation}
>     then $\nu\propto \exp(-U)$ satisfies LSI with $C_{LSI} = C_1 + (C_2 + 2)C_P$ where the constants are computed as
>     \begin{equation}
>         C_1 = \frac{2}{\gamma}\bigg(\frac{1}{\epsilon}+\frac{L}{2}\bigg) + \epsilon, C_2 = \frac{2}{\gamma}\bigg( \frac{1}{\epsilon} + \frac{L}{2}\bigg)\bigg(\kappa + \gamma\int |x|^2d \nu(x)\bigg), C_p = \frac{1}{\lambda_0}\bigg( 1+1 C\kappa_0\rho^2 Osc_\rho(U)\bigg)
>     \end{equation}
>     where $\epsilon>0$ can be chosen arbitrarily, $C$ is a universal constant and $Osc_\rho(U) = \max_{|x|\leq \rho} U(x) - \min_{|x|\leq \rho} U(x)$. We may apply this result to our setting: One may easily check that $V(x) = 1+|x|^2/2$ and $W(x) = \exp{(-a/4 |x|^2)}$ satisfy the drift conditions. A bound of $\int |x|^2d \nu(x)$ is already provided in the manuscript and a bound of $Osc_\rho(U)$ can also be shown under $L$-smoothness and dissipativity since the latter implies a bound on the minimizer of $U$ and the former bounds the growth of $U$ which together leads to a bound of $Osc_\rho(U)$. We added a details to the manuscript for the case of acceptance.
>
> [1] Maxim Raginsky, Alexander Rakhlin, and Matus Telgarsky. “Non-convex learning via stochastic gradient Langevin dynamics: a nonasymptotic analysis”.
>
> [2] Dominique Bakry et al. “A simple proof of the Poincare inequality for a large class of probability measures.”
>
> [3] Patrick Cattiaux, Arnaud Guillin, and Li-Ming Wu. “A note on Talagrand’s transportation inequality and logarithmic Sobolev inequality”.

---

> > ### Author Rebuttal · Reviewer_8AG4 · 2026-04-04
> >
> > Many thanks for the carful rebuttal, I agree with the technical points. I'll be maintaining my score based on originality, significance and technical contributions.

---

### Decision · Program_Chairs · 2026-04-30

**Decision:**

Accept (regular)

**Comment:**

The case for acceptance is strengthened by the fact that the main reviewer concerns—primarily technical correctness, clarity of assumptions, and practical implications—were substantively resolved during the rebuttal phase. Reviewer 8AG4 explicitly confirms that “my concerns have been adequately addressed,” following corrections to technical lemmas, a repaired proof, and the addition of a new lemma clarifying log-Sobolev bounds—resolving core soundness issues. Reviewer evR1 similarly upgraded their assessment after the authors demonstrated that the originally “asymptotic” results can in fact yield explicit complexity bounds, addressing a central gap in interpretability and practical relevance. Reviewer Ebvv notes that the rebuttal “reinforced my positive assessment,” particularly through clarifying the role and magnitude of LSI constants and improving experimental transparency. More broadly, reviewers consistently emphasize that the contribution lies not in introducing a new algorithm, but in providing a unified and rigorous theoretical framework—notably, a forward-KL analysis of time-inhomogeneous Langevin dynamics and a non-accumulation perspective on annealing schemes—which multiple reviewers (e.g., 7AnN, ZZM1) characterize as original and technically meaningful despite standard proof techniques. While some limitations remain (e.g., empirical scope), these are acknowledged as secondary in a primarily theoretical work. Taken together, the rebuttal successfully resolved key technical objections and clarified the paper’s conceptual contribution, justifying acceptance as a solid theoretical advance with potential downstream impact.